



# Improved parameterization of the weathering kinetics module in the PROFILE and ForSAFE models

Harald Ulrik Sverdrup[1,8*], Eric H. Oelkers[2,3] Martin Erlandsson Lampa[4],
Salim Belyazid[5], Daniel Kurz[6], Cecilia Akselsson[7]

1- System Dynamics, Department of Gamification, Inland Norway University of Applied Sciences, NO- 2318, Hamar, Norway, 2-Earth Sciences, University College London, WC1E 6BT, London, UK, 3-CNRS, UMR-5563, Toulouse, France, 4-Institute of Hydrology, University of Uppsala, SE-751 05 Uppsala, Sweden, 5-Physical Geography, Stockholm University, SE-106 91 Stockholm, Sweden, 6-EKG Geoscience, CH-3012 Bern, Switzerland, 7-Earth Sciences, University of Lund, SE-221 00 Lund, Sweden. 8-Industrial Engineering, University of Iceland, IS-107 Reykjavik, Iceland. *corresponding author (harald.sverdrup@inn.no)

**Abstract**

Although the PROFILE and ForSAFE model can accurately reproduce the chemical and mineralogical evolution of the soil unsaturated zone, it overestimates weathering rates in deeper soil layers and in groundwater systems. This overestimation has been corrected by improving the kinetic expression describing mineral dissolution by adding or upgrading 'braking functions'. The base cation and aluminium brakes have been strengthened, and an additional silicate brake has been developed, improving the ability to describe mineral-water reactions in deeper soils. These brakes are developed from a molecular-level model of the dissolution mechanisms. Equations, parameters and constants describing mineral dissolution kinetics have now been obtained for 113 minerals from 12 major structural groups, comprising all types of minerals encountered in most soils. The PROFILE and ForSAFE weathering sub-model was extended to cover two-dimensional catchments, both in the vertical and the horizontal direction, including the hydrology. Comparisons between this improved model and field observations are available in Erlandsson Lampa et al. (2019, This special issue). The results showed that the incorporation of a braking effect of silica concentrations was necessary and helps obtain more accurate descriptions of soil evolution rates at greater depths and within the saturated zone.

## 1. Introduction

This manuscript reviews the chemical weathering approach adopted by the PROFILE and ForSAFE models and describes continuing efforts to upgrade the kinetic databases of these models for improved model calculations. The application of mineral dissolution kinetics to natural systems requires a large amount of field input including information on mineral surface areas, mineral abundances over time and their spatial distribution, fluid flow and biotic activity. As such this manuscript will by design describe both the weathering models and the evaluation of laboratory mineral dissolution rate used in the development of the upgraded kinetic database.

Chemical weathering of silicate minerals, and notably the dissolution rates of these minerals are one of the most important factors shaping soil chemistry. The quality of the kinetic database in most cases determines the quality of its simulations of soil evolution. In the 1980's, the need arose to mitigate acid deposition, to set critical loads for acid deposition, and to set limits for sustainable forest growth and nitrogen critical loads. This need led to a re-evaluation of the weathering observations available in scientific publications and books (Sverdrup 1990, Sverdrup and Warfvinge 1992, 1993, 1995, Drever et al., 1994, Drever and Clow 1995, Ganor et al., 2005, Svoboda-Colberg and Drever 1993, Crundwell 2013). These observations led to a model that accurately reproduced weathering rates under field conditions. The early history of these efforts was reported by Sverdrup and Warfvinge (1988a,b, 1992, 1993, 1995) and Sverdrup (1990). By 1990, we had a set of equations that described the dissolution rates of 14 minerals (K-feldpar, albite, plagioclase, pyroxene, hornblende, garnet, epidote, chlorite, biotite, muscovite, vermiculite, apatite, kaolinite, and calcite). Later more silicate minerals were added, including illite, smectite, montmorillonite, sericite and volcanic glass. Eventually we amassed kinetic data for 45 additional silicate minerals and 25 different carbonates[1] at the time.

---

[1]**Calcite** (The calcites are all slightly different; $CaCO_3$ with 0-3% $MgCO_3$ and 0.05%-0.5% apatite, from Sweden, Norway, Denmark,and the United States. In addition, kinetics on **aragonite** ($CaCO_3$), **slavsonite** ($SrCO_3$), **dolomite** ($CaMg(CO_3)_2$, **magnesite** ($MgCO_3$), **brucite** ($MgOH$), **siderite** ($FeCO_3$), **witherite** ($BaCO_3$), and **rhodochroisite** ($MnCO_3$) is available.





By the middle of the 1980's, it became clear that we did not have a standard procedure for building a
weathering rate model based on molecular level mechanisms. There are many reasons for this, the most
important was the lack of a mechanistically oriented approach for guiding experimental studies. The lack of a
mechanistic understanding resulted in important factors being overlooked. Many essential variables required
for a weathering model were missing in the older experimental studies, sample preparation was often
inadequate or not done, and/or the material was inadequately characterized (Sverdrup et al., 1981, 1984,
Sverdrup, 1990). Often the experimental design had significant flaws and many experiments ran for too short
a time; see Sverdrup (1990) for a full description. As such there needed to be a sorting of the data, to avoid the
confusion brought by misleading data. This effort led to the creation of the original PROFILE mineral kinetic
weathering model (Sverdrup, 1990) to estimate the rate at which mineral dissolution provided essential cations
to soil waters. Although this model provides accurate estimates for shallow soils, it became less accurate for
deeper soils (e.g. > 1.5 meter soil depth).
This report outlines our efforts to update these early mineral weathering kinetics models for accurate
predictions of watershed water and deeper groundwater chemistry. This effort builds upon the weathering book
by Sverdrup (1990) and the articles Sverdrup and Warfvinge (1988a,b, 1992, 1995) and Warfvinge and
Sverdrup (1993). There is an advisory chapter on how to estimate weathering rates in soils on a regional scale
in Europe in the United Nations Economic Commission for Europe, Long Range Transboundary Convention
Mapping Manual for Critical loads (Sverdrup, 1996). The weathering rate mapping methodology based on
PROFILE model predictions was tested and used throughout 26 different European countries, and peer
reviewed at annual workshops from 1988 to 2017.
The revision of the original PROFILE weathering rate models described in this report was motivated
by several observations:
1. The PROFILE model was found to work satisfactorily in the unsaturated zone (0-1 meter), on thin
soils, on rock surfaces, and in low concentration systems (Sverdrup and Warfvinge 1988a,b, 1991,
1992, 1993, 1995, 1998, Sverdrup 1990, Sverdrup et al., 1998, Hettelingh et al., 1992, Alveteg et al.,
1996, 1998, 2000, Alveteg and Sverdrup 2000).
2. However, the chemical weathering rate for minerals is overestimated by this model in deeper soils,
and at depths of more than 1.5 meters. The original PROFILE model was used down to this depth
(Sverdrup et al., 1988a,b, 1992, 1996, Sverdrup 1990, Janicki et al., 1993, Holmqvist et al., 2003) for
critical loads for streams (Sverdrup et al., 1996) and groundwater (Warfvinge et al., 1987), and may
have possibly resulted in overestimates of the critical load.
3. The weathering rate is overestimated in the deeper soils and in ground water (Sverdrup 1990,
Warfvinge and Sverdrup 1987, 1992a,b,c, Sverdrup et al., 1996).
4. New experimental data published in the literature after 1995 is of far better quality and consistency,
with better experimental designs, better characterized materials and more complete observations than
previous studies. For example, the reader is encouraged to read two studies published by Holmqvist
et al., (2002, 2003) on the weathering rates of clay minerals under soil conditions and the concept of
mineral alteration sequences (Holmqvist 2004, PhD thesis from Chemical Engineering, Lund
University). The minerals used in the weathering rate experiments in those studies were extracted and
separated from in-situ soils at experimental field sites near Uppsala, Sweden.
This study describes the updated mineral kinetics database used in the PROFILE and ForSAFE
models,  Notably this update includes revised 'brake functions' in the kinetic rate equations to better fit the
observed field data down to the groundwater table and below. This was necessitated when the ForSAFE model
(thus also the PROFILE model) was reconfigured for a sloping catchment, expanding the model structure from
a 1-dimensional to a 2-dimensional model accounting for vertical and horizontal solute transport in a
catchment, including the ecosystem. In total 102 minerals are considered in the updated and expanded kinetics
parameter databases. An exhaustive description of the parameterization of the rate equations for all of the 102
minerals will require a text far beyond what is possible in this manuscript, so that only a summary and several
examples are provided here.





The arrow shows a causality. A variable at the tail causes a change to the variable at the head of the arrow.

(arrow)

(tail)          (head)

A plus sign near the arrowhead indicates that the variable at the tail of the arrow and the variable at the head of the arrow change in the *same* direction. If the tail *increases*, the head *increases*; if the tail *decreases*, the head *decreases*.

+

A minus sign near the arrowhead indicates that the variable at the tail of the arrow and the variable at the head of the arrow change in the *opposite* direction. If the tail *increases*, the head *decreases*; if the tail *decreases*, the head *increases*.

−

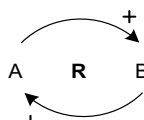

A   **R**   B

+

The letter **R** in the middle of a loop indicates that the loop is *reinforcing* a behavior in the same direction, causing either a systematic growth or decline. It is a behavior that is moving *away* from equilibrium point.

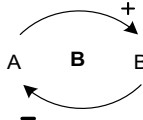

A   **B**   B

+

−

The letter **B** in the middle of a loop indicates that the loop is *balancing* and moves the system in the direction *towards* equilibrium or a fluctuation around equilibrium point.

*Figure 1. Weathering processes were mapped using systems analysis and by drawing causal loop diagrams (CLD) for the process and the whole system of the weathering process. This is a standard procedure in model building (Sverdrup and Stiernquist 2002, Sverdrup et al., 2018). B is a balancing loop (sometimes referred to as a negative feedback) and R is a reinforcing loop (sometimes referred to as a positive feedback) as explained in the figure.*

## 2. Methodology

The methods used in this study have their basis in terrestrial ecosystems system analysis and ecosystems system dynamics as described by Sverdrup and Stiernquist (2002) and Sverdrup et al. (2018). The main tools employed are the standard methods of system analysis and integrated system dynamics modelling (Forrester 1961, 1969, 1971, Meadows et al., 1972, 1974, 1992, 2005, Roberts et al., 1982, Senge 1990, Bossel 1998, Haraldsson and Sverdrup 2005, Haraldsson et al., 2006, Sverdrup and Stiernquist 2002, Sverdrup et al., 2018). The overall system is analysed using stock-and-flow charts and causal loop diagrams (Sverdrup et al., 2002). The learning loop was used as the adaptive learning procedure in past studies (Senge 1990, Kim 1992, Senge et al., 2008, Sverdrup et al., 2018). The conceptrual model must be clearly defined and constructed before any computational work can be undertaken. It is fundamental to understand that the causal understanding is the model. Systems analysis produces a causal loop diagram (CLD) linking causes, effects, and feedbacks among the processes in terms of causalities and flows (Albin 1997, Sverdrup et al., 2018, Kim 1992). These CLD need to be internally consistent. A summary of this approach is provided in Figure 1. A causal loop diagram is thus a map of the differential equations describing the evolution of the system. Mass- or energy flow charts and the causal loop diagram uniquely define the system. The ForSAFE model is not calibrated on large amounts of system output data (Sverdrup and Warfvinge 1992, Sverdrup et al., 2018). Instead, the system's causal linkages and the mass balances lead to equations that are parameterized using independent system properties, initial states and boundary conditions (Sverdrup et al., 2018).

## 2.1 Earlier development work and background

Critical to developing a database describing mineral dissolution rates is that it is coupled into a comprehensive model that can account for the large number of processes that affect rates in the field. From the beginning, weathering kinetics was developed and incorporated into the PROFILE model. The kinetics were parameterized using laboratory measurements and applied to field conditions on a plot scale and on a regional scale for Sweden (Sverdrup 1990, Sverdrup and Wafvinge 1988a,b, 1992, 1995, Warfvinge and Sverdrup 1992, 1993). The resulting kinetics sub-model was subsequently coupled into a biogeochemical ecosystem model, linking solute transport, soil chemistry, weathering, ion exchange, hydrology and biological





interactions with microbiology and forest plants, called the SAFE model (Sverdrup et al, 1995). The steady-
state model PROFILE and its dynamic variant SAFE, was further developed into other models as described in
the Appendix.
**2.2. Weathering under field conditions**
The dissolution of primary minerals at ambient temperature and pressure is irreversible with the
exceptions of a few simple chloride and sulphate salts and a few carbonates (Sverdrup 1990). Such irreversible
reactions do not attain equilibrium in near to ambient temperate systems as the chemical species released to
natural waters combine to form secondary solid phases far before the waters attain close to equilibrium
conditions with respect to primary minerals. Nevertheless, the dissolution rates of the primary minerals have
been observed to slow at far from equilibrium conditions in response to the increased concentration of
dissolved metals including Al and Si. A formulation based on transition state theory for the formation of
activated surface complexes that decay irreversibly was developed by (Sverdrup 1985, Sverdrup and
Warfvinge 1987, 1988a,b, 1992, Sverdrup 1990) to describe the effect of dissolved metals on primary mineral
dissolution at far from equilibrium conditions. Taking account of this approach as well as their coupling to
solute transport, ion exchange, plant nutrient uptake, organic matter decomposition and nitrogen
transformations detailed modelling of chemical weathering rates have been made (Sverdrup and Warfvinge
1988a,b, Sverdrup 1990, Akselsson et al., 2006, 2005, 2004, Sverdrup et al., 1990, 1995, 2017). A comparison
of calculated and observed weathering rates shown in Figure 2, demonstrates this approach can reproduce the
observed rates within ±5% across 4 orders of magnitude for the upper unsaturated parts of a soil (Sverdrup
and Warfvinge 1992, Barkman et al., 1999, Jönsson et al., 1995, Belyazid 2005, Kurz et al., 1998a,b). Further
comparisons of computed and calculated rates made with these models for field tests at Gårdsjön, Sweden and
at various sites were published by Sverdrup et al. (1988a,b, 1993, 1995, 1996, 1998, 2010), Sverdrup (1990,
2009), Sverdrup and Alveteg (1998), Rietz (1995) and Warfvinge et al., (1996), and Holmqvist et al., (2003,
2002). In addition, several other authors tested this approach independently (In the United States; Kolka et al
1996, Phelan et al., 2014, in Scotland; Langan et al. 2006b, in Germany; Becker 2002, in New Zealand:
Zabowski et al., 2007; tests on controlled experiments with granite slabs in the Swedish nuclear waste storage
assessment research programme at Göteborg by Claesson-Nyström and Andersson 1996, in Swedish soil
profiles; Lång 1998). Gunnar Jacks in KTH, Stockholm put these models to several blind test of the alteration
of blank granite surfaces used for ancient rock carvings and controlled mini-catchments (Jacks, unpublished
1990). In each case a close correspondence was observed in calculated as compared to the field weathering
rates.  The current manuscript reports on our efforts to extend these accurate calculations to deeper in the soil
column.
**3. Theory**
The kinetic weathering model presented in this manuscript originates from that of Sverdrup and
Warfvinge (1987a,b, 1988a,b, 1992a, 1995) and Sverdrup (1990), but numerous features have been added
since. Some of the updates have been described in later studies (Akselsson et al., 2005, 2005, 2006, 2007,
Alveteg et al., 2000, Kurz et al., 1998a,b, Sverdrup et al., 1997, 2002, 2008). Further updates are described in
this study. New weathering rate data published over the past 25 years have been regressed and new temperature
dependencies and modifications of some rate coefficients has resulted (Sverdrup 2010, Sverdrup et al., 1998,
Rizzetto et al., 2016, Holmqvist et al., 2002, 2003). The mineralogy and surface area inputs to the models are
based on site measurements, and in general are not adjustable parameters. Some of parameters can be
challenging to measure, such as some primary minerals with low soil content (apatite, epidote, pyroxene,
amphiboles, garnets accurate to 0.1%), or mineral surface area. However, getting accurate field estimates of
the weathering rates is also challenging, as it requires making many assumptions, so may be of limited
accuracy. Thus, we are comparing uncertain model estimates with equally or more uncertain field estimates at
the best (Sverdrup et al., 1998).  Nevertheless such comparisons are essential to validate model results. Of all
the parameters needed for calculating mineral dissolution rates in natural systems using laboratory measured
rates among the most challenging are mineral surface areas.  Whereas in laboratory studies of the dissolution
rates of individual minerals it is possible to measure directly the areas of cleaned mineral surfaces using gas
adsorption techniques, field samples are more complex as they many contain the surfaces of several minerals
and these surfaces can be covered by both organic substances or secondary minerals.  Assuming that the surface
area of each mineral in a soil is proportional to its mass or volume fraction may not be appropriate due to the
differing typical shapes of distinct minerals.  The protocols used to estimate the surface areas of natural
minerals in soils within the PROFILE and the ForSAFE models have been reviewed in detail by Sverdrup
(1990).

Biogeosciences Open Access
Discussions
EGU

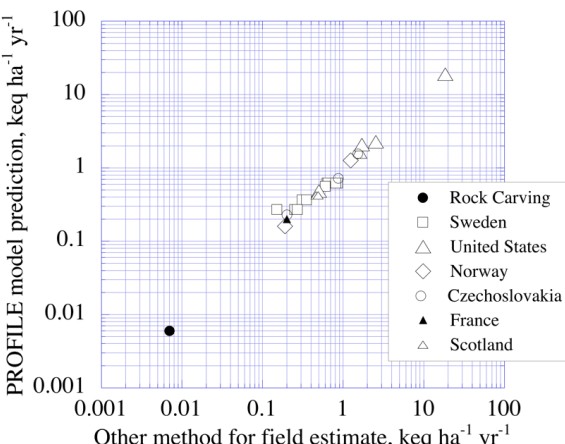

Figure 2. Comparison of weathering rates calculated using the original PROFILE model with corresponding rates obtained from field observations of the upper undersaturated parts of soils. Rates shown were reported or compiled by Sverdrup and Warfvinge (1988a,b, 1991, 1992, 1993, 1995, 1998), Sverdrup (1990), Sverdrup et al. (1990, 1998), Hettelingh et al. (1992), Barkmann et al. (1999), Holmqvist et al. (2003). The model test was performed on shallow soil profiles, no deeper than 0.6 meter.

### 3.1. Defining chemical weathering

Weathering neutralizes acids (neutralizing all or part of acid rain) and provides nutrients for vegetation (e.g. $Ca^{2+}$, $Mg^{2+}$, $K^+$, $PO_4$) (Sverdrup 1990, Sverdrup and Warfvinge 1995, Sverdrup et al., 2002). Thus weathering rates are defined as "the base cation release rates from the chemical weathering of minerals", "plant nutrient base cation release rates from the chemical weathering of minerals" or "the rate of acid neutralization by chemical weathering of soil minerals". Only secondarily are we interested in loss of minerals and soil profile development (Rietz 1995, Warfvinge et al., 1996, Sverdrup et al., 1996, 2002). Thus, the weathering rates in this study have been expressed as the sum of the release rates of base cations ($Ca^{2+}$, $Mg^{2+}$, $K^+$, $Na^+$) from the process. This is linked to the destruction of minerals, though results are generally expressed in these terms.

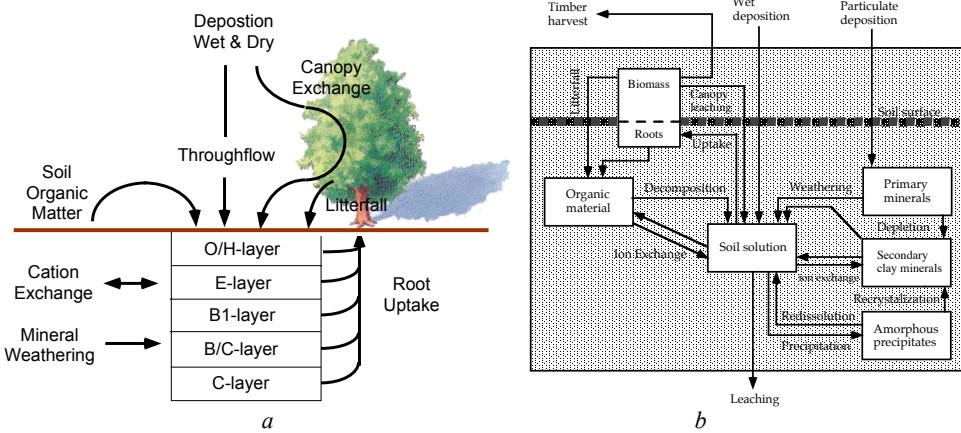

*a*             *b*

Figure 3. Overview of the PROFILE model. The original PROFILE model operates with a number of layers, and a vertical percolation of water. A set of processes take place in every layer. (b) A look inside PROFILE, showing how weathering is connected with other ecosystem processes (Sverdrup and Warfvinge 1995).



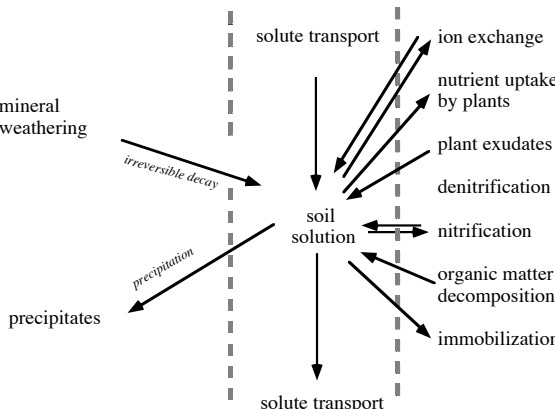

*Figure 4. Different soil processes communicate with the weathering processes via the soil solution. (Sverdrup*
*et al., 2002).*

**3.2. Mineral weathering rates**
The weathering rate of a mineral, r, defined here as its dissolution rate, is assumed to stem from the
sum of 5 simultaneous chemical reactions, involving the mineral surface and either aqueous $H^+$, $H_2O$, $OH^-$,
organic acid ligands, or $CO_2$. Assuming that the reactions occur at distinct active mineral surface sites, they
can be summed linearly in accord with (Sverdrup 1990, Sverdrup and Warfvinge 1995):
$$R_W = \sum_{j=1}^{Minerals} A_j * \sum_{i=1}^{\substack{Dissolution \\ reactions}} r_i \qquad (1a)$$

where $R_W$ stands for the soil weathering rate in a single soil layer. $A_j$ refers to the soil mineral surface available
for dissolution for each mineral j considered, $r_i$ designates the rate of the individual chemical reactions i. If
some reactions occupy the same active mineral surface sites, the expression given above would change to a
quadratic sum. Note that the results of the two equations are quite similar, so that the importance of knowing
if several reactions operate of the same surface site is relatively small. For the whole soil profile the rates are
summed over the different soil layers with depth and we get:

$$R_{Soil} = \sum_{s=1}^{Layers} R_{W,s} \qquad (2)$$

where $R_{Soil}$ denotes the weathering rate in the whole soil profile, and s represents the layer number. Evidence
that the $H^+$, $H_2O$ and $OH^-$ reactions take place at distinct surface sites has been reviewed by Sverdrup (1990)
and again by Holmqvist et al., (2003). The $H_2O$, the organic reaction and the $CO_2$ reactions may occur at the
same sites, but considering the available data, we have assumed that they occur at distinct sites and thus favour
a linear sum of rates. More on these assumptions have been reported by Sverdrup (1990), Sverdrup and
Warfvinge (1995), and Holmqvist et al. (2002, 2003).

**3.3. Field weathering rates**
To estimate field weathering rates using laboratory determined kinetic coefficients, an ecosystem
model is required to scale the process to field conditions. This ecosystem model includes effects of climate,
soil morphology, plants, trees, microbiology in the soil and fungi (Lin et al., 2017, Smits and Wallander 2016,
Smits et al., 2014). An ecosystem model is incorporated within PROFILE and ForSAFE (Sverdrup and
Warfvinge 1988a,b, 1991, 1992, 1993, 1995, 1998, Sverdrup 1990, Sverdrup et al., 1998, Hettelingh et al.,
1992, Barkmann et al., 1999, Holmqvist et al., 2003, Barkman et al., 1999). Figure 3 shows how the steady-
state PROFILE model was configured (Sverdrup and Warfvinge 1988a,b, 1992, 1993, Sverdrup and Alveteg
1998). In the dynamic integrated terrestrial ecosystem assessment model ForSAFE-VEG, the system evolution
takes account of interactions with a living biosphere, organic matter turnover and ion exchange (c.f. Figure 4).
Further details of these models can be found in the appendix and the literature (Sverdrup et al., 1987, 1995,





1996a,b, 1998, 2007, 2014, 2016, 2017, 2019, Wallman et al., 2002, 2003, Zancchi et al., 2014, 2016a,b,
Belyazid et al., 2017, 2018).
To estimate field weathering rates, each reaction i for every mineral j is corrected for the field site
temperature and for the partial wetting of the soil (Sverdrup 1990, Sverdrup and Warfvinge 1995, Sverdrup
and Alveteg 1998) in accord with:

$$R_W = h(\theta) * \sum_{j=1}^{\substack{\text{Minerals}}} A_j * \sum_{i=1}^{\substack{\text{Dissolution} \\ \text{reactions}}} \left( r_i * g_{i,j}(T) \right) \qquad (3)$$


where $\theta$ stands for the fraction of the soil mineral surfaces wetted, $A_j$ designates the surface area of the mineral
j, $h(\theta)$ refers to a wetting function for the mineral material and T signifies the soil temperature in centigrade.
$g_{ij}(T)$ corresponds to the temperature adjustment function for reaction i of mineral j. $r_i$ denotes the reaction rate
of dissolution reaction i. This adjustment is based on the Arrhenius equation and takes account of the
difference in rates between the temperature of the field site and that of the parameter database, which was set
at 8°C (Sverdrup 1990). Figure 6 shows the reaction causal loop diagram for silicate minerals in the soil
(Sverdrup 1990, Sverdrup and Warfvinge 1995). This diagram shows how the mineral weathering process
communicates with other biogeochemical processes in a terrestrial ecosystem. The causal loop diagram is a
graphical display of the differential balances in the system. Together with the flow charts, they define the
system. The process has several intermediate equilibrium steps, but pass an irreversible dissolution threshold
(Figure 7). The single irreversible step makes the whole process irreversible. The reaction products exert a
negative effect on the amount of activated complex that can decay, thus they slow the dissolution reaction. But
once the activated complex has formed, it has a constant decay rate, set by quantum mechanics (Sverdrup
1990, Sverdrup and Warfvinge 1995). The full derivation of the rate equations, starting from the elementary
chemical reactions and the decay of the surface complexes according to transition state theory has been
reviewed by Sverdrup (1990) and Sverdrup and Warfvinge (1995).
**3.4 Mineral reaction kinetics**
As stated above, five reactions are assumed to contribute to the total chemical weathering rate of a silicate
mineral in soils (Sverdrup 1990, 2009, Sverdrup and Warfvinge 1995):
1. The reaction between the mineral surface and the aqueous hydrogen ion
2. The reaction between the mineral surface and the water molecule
3. The reaction between the mineral surface and aqueous carbon dioxide
4. The reaction between the mineral surface and aqueous organic acid ligands
5. The reaction between the mineral surface and the aqueous hydroxy ion
Reactions 1-4 in the list above were included in earlier versions of the PROFILE and ForSAFE mineral
dissolution rate equations (Sverdrup 1990, Sverdrup and Warfvinge 1995). This original model has been
enlarged to include reaction 5.
The reaction of the mineral surface with the aqueous $H^+$ ion, reaction 1, is considered part of the
reaction with the $H^+$ reaction regardless of the source of $H^+$ (Figures 5 and 7). Both $CO_2$ and organic acids can
change the fluid pH, and this is accounted for in the $H^+$ reaction. Figure 5 shows the reaction pathway through
the $H^+$ reaction, adapted after Sverdrup (1990). Some of the reaction products form secondary minerals.
Amorphous phases may also precipitate from solution. These can slowly recrystallize to secondary minerals.
This has been generalized in Figure 6.
Reaction number 4 between organic acid ligands and the mineral surface contains at least two distinct
contributions: one from fast and one from slower reacting organic acid ligands (Sverdrup 1990). We have
simplified this to one generic rate equation that could be parameterized for some minerals (feldspar, olivine,
pyroxenes, hornblende, apatite; Sverdrup et al., 1990, later literature has extended the list somewhat). The
importance of organic acids for weathering has been frequently over estimated in the literature, and several
claims of strong effects of organic acids have been made (For a review see Smits and Wallander 2016, Smits
et al., 2014, Sverdrup 1990, 2009 but also Keegan and Laskow-Lehey 2014 on why these claims have been so
persistent). The highest concentration of organic acids occur in the upper soil layers, where the mineral content
is relatively low. As the mineral contents increase with depth, the concentrations of organic acids are lower
and have only a marginal effect on the overall weathering rate (Sverdrup 2009).



Organic acids in soils are mostly sourced from soil organic matter decomposition. Trees, soil fungi
and mycorrhiza do not have the ability to increase the weathering rate significantly (See Sverdrup 1990, 2009,
Sverdrup and Warfvinge 1992, Warfvinge and Sverdrup 1993 for details, kinetic expressions and data
underpinning this, see Smits and Wallander 2016 and Smits et al., 2014 on the subject concerning apatite).
Trees and vegetation can indirectly affect weathering rates when they take up Ca, Mg, K as nutrients, and
thereby removing weathering rate products that can slow mineral dissolution. Decomposition of plant debris
and soil organic matter produce organic acids that may react with the minerals. This effect is passive, and does
not occur not by design of the plants (See Smits and Wallander 2016 and Smits et al., 2014 for measurements,
Keegan and Laskow-Lehey 2014 for some social aspects and Sverdrup 2009 for a further analysis from a
systemic perspective).

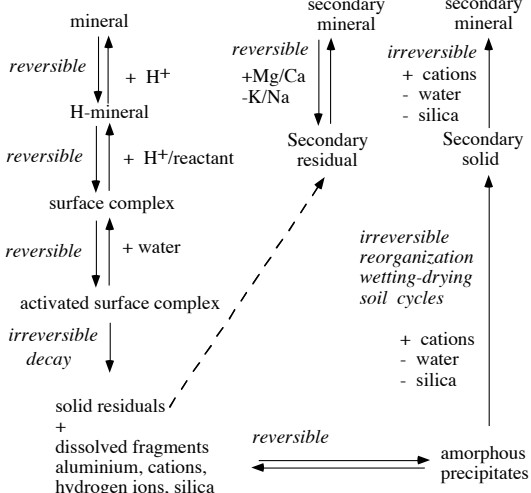

*Figure 5. The reaction pathway through the H⁺ reaction passes over several reversible steps that change the*
*surface sites and create an unstable surface complex; the Transition State Surface Complex that will decay*
*irreversibly. Note that the process is irreversible, and thus cannot go backwards. The mineral may dissolve*
*completely, be altered to a secondary mineral or form precipitates that slowly recrystalize to secondary solid*
*phases.*

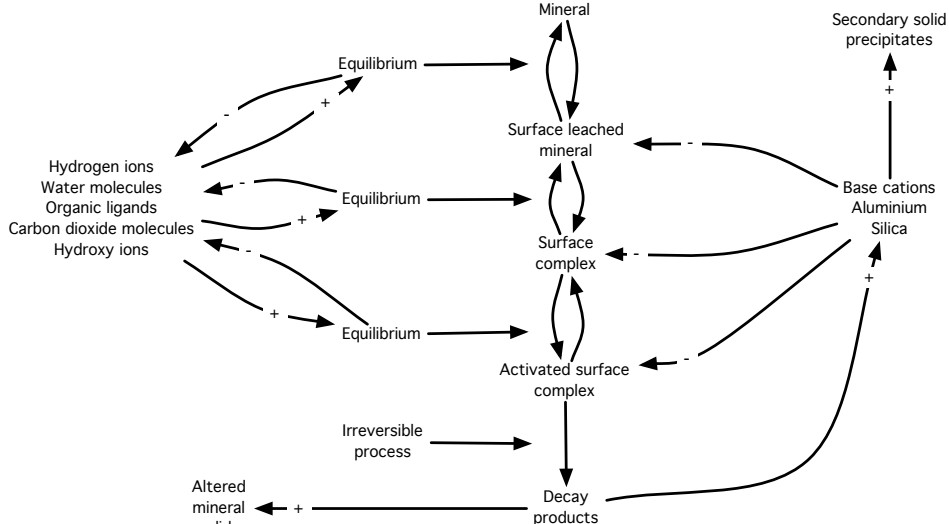

*Figure 6. Reaction pathway for silicate minerals in soils according to Transition State Theory as implemented*
*by the authors (See Sverdrup 1990, Sverdrup and Warfvinge 1995 for a full explanation).*



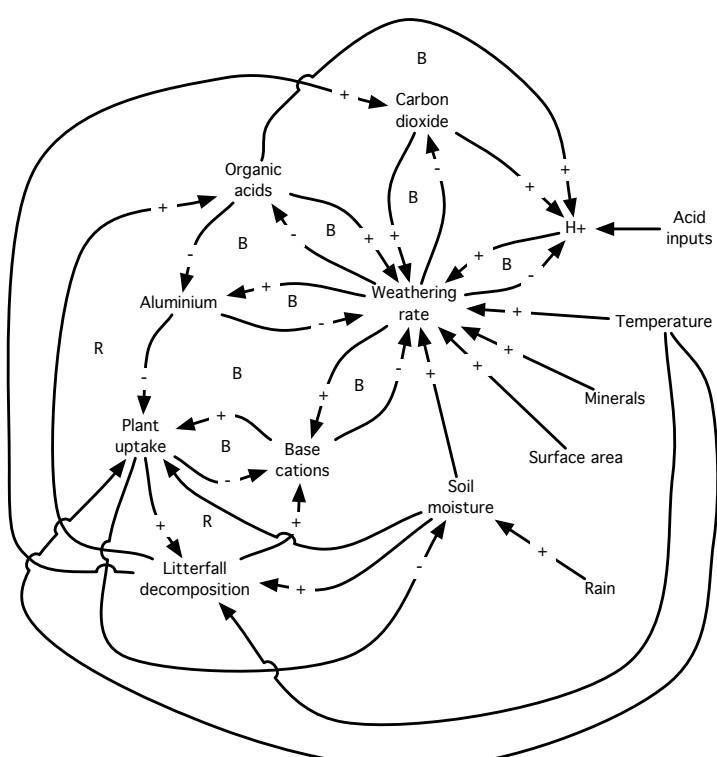

*Figure 7. The partial causal loop diagram for the weathering of a soil. See Sverdrup et al. (2018) for a full*
*explanation of causal loop diagrams and their use in modelling.*
Fluorides form soluble complexes in water with aluminium and silicates. The reaction of the mineral
surface with fluoride anions forms a strong reactions, but this occurs very rarely as the fluoride concentrations
are very low. The fluoride reaction has been ignored in this approach for most soils in natural terrestrial
ecosystems, as this would cause an unnecessary complication of the aluminium and silicate chemistry. The
dissolution rate per surface area of a mineral considering the first of the four above reactions is thus consistent
with (Sverdrup and Warfvinge 1988, 1992):
$$r_{Total} = r_{H^+} + r_{H_2O} + r_{CO_2} + r_R \qquad (4)$$

The mineral dissolution kinetic equation for the 4 individual reactions applied in the original PROFILE
model was the simplified version of the full kinetic expression based on the Transition State Theory applied
to silicate chemical weathering (see Sverdrup 1990, Sverdrup and Warfvinge 1995):
$$r = k_H * \frac{[H^+]^{n_H}}{f_H} \;+\; \frac{k_{H_2O}}{f_{H_2O}} \;+\; k_{CO_2} * P_{CO_2}^{n_{CO_2}} * \frac{1}{f_{CO_2}} \;+\; k_R * \frac{[R]^{n_R}}{1 + K_{Org} * [R]^{n_R}} * \frac{1}{f_R} \quad (5)$$

where the different $n$ designate reaction orders. The different $k_H$, $k_{H2O}$, $k_{CO2}$, $k_R$ stand for rate coefficients.
Constitutents within brackets [c] are concentrations, and R refers to organic ligands. The different $f_{H^+}$, $f_{H2O}$,
$f_{CO2}$, $f_R$, $f_{OH}$ signify retarding or 'brake' functions defined by (Sverdrup 1990, Sverdrup and Warfvinge 1992,
Warfvinge and Sverdrup 1993, Sverdrup and Warfvinge 1995):
$$f_{H^+} = \left(1 + \frac{[BC]}{C_{BC,H}}\right)^{x_H} * \left(1 + \frac{[Al^{3+}]}{C_{Al,H}}\right)^{y_H} \qquad (6)$$




$$f_{H_2O} = \left(1 + \frac{[BC]}{C_{BC,H_2O}}\right)^{x_{H_2O}} * \left(1 + \frac{[Al^{3+}]}{C_{Al,H_2O}}\right)^{y_{H_2O}} \quad (7)$$

$$f_{CO_2} = \left(1 + \frac{[BC]}{C_{BC,CO_2}}\right)^{x_{CO_2}} * \left(1 + \frac{[Al^{3+}]}{C_{Al,CO_2}}\right)^{y_{CO_2}} \quad (8)$$

$$f_R = \left(1 + \frac{[BC]}{C_{BC,R}}\right)^{x_R} * \left(1 + \frac{[Al^{3+}]}{C_{Al,R}}\right)^{y_R} \quad (9)$$

$$f_{OH^-} = \left(1 + \frac{[BC]}{C_{BC,OH}}\right)^{x_{OH}} * \left(1 + \frac{[Al^{3+}]}{C_{Al,OH}}\right)^{y_{OH}} \quad (10)$$

Note that the retardation or 'braking' functions represent molecular mechanisms that slow the reaction by forming fewer active surface complexes (Sverdrup 1990, Sverdrup and Warfvinge 1995). $Al^{3+}$ is the concentration of positive aluminium species in the aqueous solution, and not necessarily equal to the total aluminium concentration (Sverdrup 1990 – see also section 4.8); this concentration can be calculated using aqueous speciation estimates as described below. The subscript BC,OH represents a term related to base cations (BC) in the $OH^-$ reaction, Note this slowing of the rates with increasing fluid concentration is not due to the approach to a mineral-water equilibrium state. The dissolution of many primary silicate minerals is not reversible under normal soil conditions as the fluids do not attain close to equilibrium conditions. Instead, there will be a steady-state between the reaction at the surface and the removal of ions by solute transport and precipitation into secondary phases. This may look like an equilibrium condition, but does not behave like one. A few minerals are exceptions such as calcite, a few other carbonates, hydroxides and quartz. Even with these the attainment of equilibrium is kinetically limited. For calcite in soils we have observed this to take several days or weeks (Warfinge et al., 1987). All other minerals (feldspars, pyroxenes, amphiboles, etc.) do not precipitate from solution, some amorphous aluminosilicate clay precursors only precipitate very slowly.

### 3.5. The updated kinetics equation

The original 4 mineral dissolution reactions have been enlarged to include $OH^-$-reaction in the present study. The complete equation is consistent with

$$r_{Total} = r_{H^+} + r_{H_2O} + r_{CO_2} + r_{R+} + r_{OH^-} \quad (11)$$

The full kinetic equation for all 5 reactions is (Sverdrup 1990, Sverdrup and Warfvinge 1995):

$$r = k_H * \frac{[H^+]^{n_H}}{f_H} + \frac{k_{H_2O}}{f_{H_2O}} + k_{CO_2} * \frac{P_{CO_2}^{n_{CO_2}}}{1 + K_{CO_2} * P_{CO_2}^{n_{CO_2}}} * \frac{1}{f_{CO_2}}$$

$$+ k_R * \frac{[R]^{n_R}}{1 + K_{Org} * [R]^{n_R}} * \frac{1}{f_R} + k_{OH} * \frac{[OH^-]^{n_{OH}}}{f_{OH}} \quad (12)$$

For most minerals, the strongest effect of the brake functions is that of aluminium at pH < 7, followed by silica and base cations. At pH > 8, the strongest effect is from silica and base cations, and less pronounced for aluminium (Sverdrup 1990). Before applying Equation (12) a number of new adaptions have been carried out as described below.

### 3.6. Retardation of mineral dissolution rates by organic ligands

The original formula for the slowing of mineral dissolution rates with increasing organic ligand concentration was (Sverdrup 1990, Sverdrup and Warfvinge 1995):

$$r_{Org} = k_R * \frac{[R]^{n_R}}{1 + [R]^{n_R}} * \frac{1}{f_R} \quad (13)$$

this has been reformulated to:






$$r_{Org} = k_R * \left( \frac{[R]}{1 + \; [R \; + \; [R]_{limit}]} \right)^{n_R} * \frac{1}{f_R} \qquad (14)$$

The difference in these equations is that the latter contains one additional parameter $[R]_{Limit}$ in $f_R$ that has the
effect to set a lower concentration, below which the organic acids have no effect. This equation has been
parameterized and used in the final expression provided below. This limit was incorporated into the organic
acid ligand retardation function $f_R$ (Smits and Wallander 2016, Smits et al., 2014, Sverdrup 1990, 2009).
**3.7. Retardation of mineral dissolution rates by aqueous $CO_2$**
The main effect of the presence of $CO_2$ on mineral dissolution rates is to change the pH of the solution.
This effect is accounted for by the chemical solution equilibria, and dealt with in the $H^+$ reaction. The dedicated
$CO_2$ term takes into account the effect of a reaction between the $CO_2$ and the mineral surface. The effect of the
presence of aqueous organic species decreases at higher concentrations of organic acids as the surface sites
have become saturated with organic acid ligands. We hypothesize that $CO_2$ exhibits the same behaviour. Some
data show that $CO_2$ also reacts with mineral surface sites as some type of carbonate ligand (a bicarbonate
coordinated towards a cation in the lattice) adsorbed to the surface, setting up a transitional surface complex
may decay. The mechanism by which $CO_2$ effects silicate dissolution rates appears to follow the sequence
(Sverdrup 1990, Sverdrup and Warfvinge 1995, Brady and Carrol 1994, Golubev et al., 2005, Navarre-Sitchler
and Thyne 2007, Berg and Banwart 2000):
1.  The $CO_2$ molecule attaches to the mineral surface
2.  The $CO_2$ molecule forms a bicarbonate-water-metal complex with the mineral surface on singly
coordinated metal cations. Indications are that it may be the $CO_3^{2-}$ ligand that is forming a surface
complex.
3.  A cation is lifted into the complex (K, Na, Mg, Ca, Fe, etc..)
4.  A small fraction of the surface complexes detach from the surface and the mineral dissolves.
Thus there should potentially be an upper concentration limit where additional aqueous $CO_2$ will have no
further effect on mineral dissolution rates. This seems to occur between 10 and 50 atmospheres of $CO_2$ partial
pressure for mica and chlorites (Drever et al., 1996, Mast and Drever 1987, Hausrath et al., 2009). Observations
on some other minerals indicate of a similar behaviour, but this limit remains elusive due to lack of data. In
addition the dissolution rates of some minerals exhibit no detectable effect of the presence of aqueous $CO_2$,
and some are only slightly inhibited by this species. Lagache (1965, 1976), Busenberg and Clemency (1976),
Berg and Banwart (2000) and Golubev et al., (2005) reported experiments performed at different $CO_2$ partial
pressures between 0 and 26.3 $CO_2$ atmospheres and temperatures between 0 °C and 200 °C. The original
equation used by Sverdrup (1990) and Sverdrup and Warfvinge (1995) to describe these data was

$$r_{CO_2} = k_{CO_2} * \frac{P_{CO_2}^{n_{CO_2}}}{1 + K_{CO_2} * P_{CO_2}^{n_{CO_2}}} * \frac{1}{f_{CO_2}} \qquad (15)$$

In this study we use a variation of this equation of the form:

$$r_{CO_2} = k_{CO_2} * \left( \frac{P_{CO_2}}{1 + \; K_{CO_2} * (P_{CO_2} + P_{CO_2\,Limit})} \right)^{n_{CO_2}} * \frac{1}{f_{CO_2}} \qquad (16)$$

Evidence suggests that the value of $P_{Limit\,CO2}$ is in the range of 5 to 10 atmospheres and $K_{CO2}=0.05$ and $n_{CO2}$
=0.6 for albite (Sverdrup 1990). Navarre-Sitchler and Thyne (2007) suggest $n_{CO2}=0.45$, which is for practical
purposes the same. Berg and Banwart (2000) suggested $n_{CO2}=0.25$ at low pressures of $CO_2$. As mentioned
above, a similar behaviour was observed for mica, biotite and chlorites. Indications are that something similar
takes place on the surface of montmorillonite, diaspore, gibbsite, goethite and lepicrocite. There almost no
experimental data available allowing the retrieval of the parameters in Equation (16) for other minerals. The
effect of increasing aqueous $CO_2$ has been overlooked in most experimental studies.





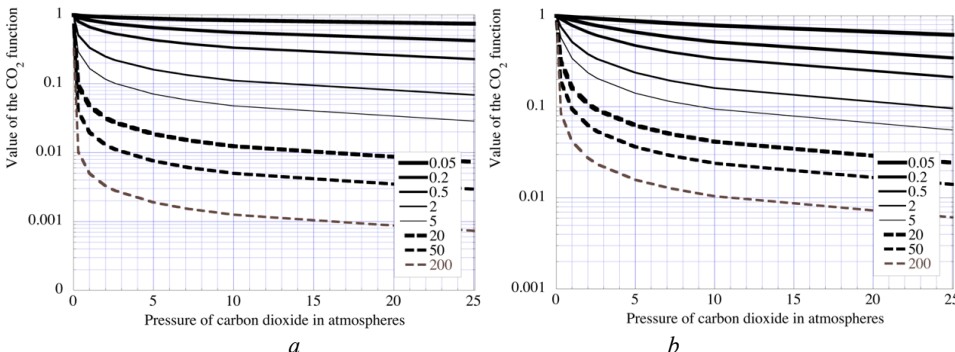

a                                                    b
*Figure 8. The calculated effect of aqueous carbon dioxide on mineral dissolution reactions using Equation 15*
*in (a) and Equation 16 in (b). See Table 2 for values for different minerals.*

| # | Silica brake response group | z-values suggested by the mineral reactions | | | | |
|---|---|---|---|---|---|---|
| | | $H^+$ | $H_2O$ | $CO_2$ | Organic acids | $OH^-$ |
| 1 | K-Feldspar and sericite | 6 | 2 | 2 | 2 | 1 |
| | Muscovite group and illites | 7 | 3 | 3 | 3 | 2 |
| 2 | Albite | 8 | 4 | 4 | 4 | 3 |
| | Na-rich Plagioclase | 7 | 4 | 4 | 4 | 3 |
| | Ca-rich Plagioclase | 10 | 6 | 6 | 6 | 4 |
| 3 | Biotite group Chlorite group Serpentinite Aluminum-nesosilicates Aluminium pyroxenes Tourmaline group | 16 | 6 | 6 | 6 | 4 |
| 4 | Amphibole group Pyroxene group Epidote group Nesosilicate | 20 32 32 32 | 16 | 16 | 16 | 8 |
| 5 | All other silicates | 32 | 16 | 16 | 16 | 8 |
| 6 | Carbonates | n.a | n.a | n.a | n.a | n.a |

*Table 1. Retrieved values of the parameter z in the silica brake function describing the dissolution rates of various silicate minerals (see equations 24-28).*

Values calculated of the effect of aqueous $CO_2$ on silicate dissolution rates are illustrated in Figure 8. These
calculations suggests that there is a significant saturation of the surface with $CO_2$ at approximately 5 to 10
atmospheres partial pressure of $CO_2$. See Table 1 for the z-values suggested for different minerals. Note that
the values of this parameter are based on minimal supporting experimental data - the available experimental
data are few and somewhat incomplete (See Golubev et al., 2005 for a limited but useful assessment). Overall,
the effect of $CO_2$ at normal soil conditions is limited. Nevertheless, these results provide a range for model
parameter adjustment. The effect of dissolved $CO_2$ on rates may become significant for deep aquifers,
subsurface $CO_2$ storage and in industrial high-pressure situations (Sverdrup 1990).

**3.8 The silica retarding or 'brake' function**
An illustrative plot of the effect of aqueous silica on silicate mineral dissolution rates is provided in Figure 9.
The equation used to describe the retardation effect of dissolved Si on mineral dissolution rates was:

$$\frac{1}{f_{Si}} = \frac{1}{1 + K_{Si,i} * \left(\frac{[Si]}{C_{Si}}\right)^{z_{Si}}} \qquad (17)$$

The value for the silica brake coefficient $K_{Si,i}$ =100 was chosen, and causes a gradual reduction in the
dissolution rate of minerals down to a minimum of approximately 0.9% of the rate unaffected by silica at very
high silica concentrations (see Table 1). Figure 9 shows values of the silica brake function calculated using



Equation 17, using the surface constant value, $K_{Si}$=100, and the saturation concentration $C_{Si}$=900 mmol per $m^3$
in Equation 17 together with the coefficients in Table 3. Exponents from $z_{Si}$ = 0.5 to 32 in Equation (17) of the
silica rate brake are shown in Figure 9.

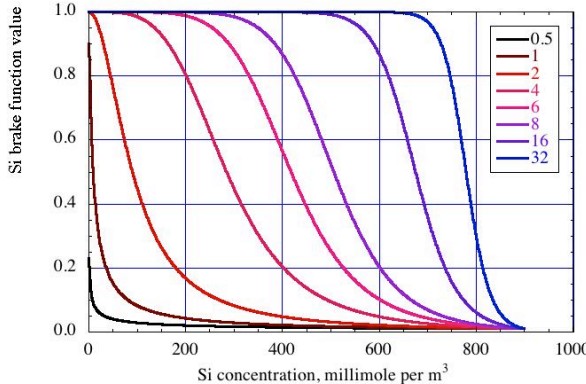

*Figure 9. Calculated effect of dissolved Si on silicate dissolution rates generated using Equation (17) together*
*with  $K_{Si}$=100, and the saturation concentration, $C_{Si}$=900 mmol per $m^3$ and the coefficients in listed in Table*
*1.*

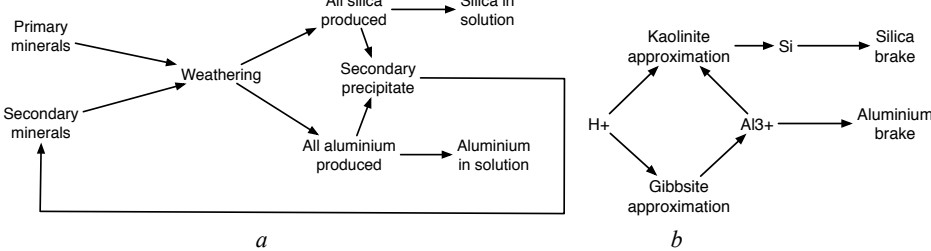

*a*                                                    *b*
*Figure 10. a) Plot illustrating the fate of silica during the mineral dissolution process. b) Diagram showing*
*how the aluminium and silica concentrations are estimated in the model. The $H^+$ concentration is used with*
*the equation called the "Gibbsite" equation (Eq. 19) to estimate the $Al^{3+}$ concentration in the soil solution.*
*These $H^+$ and $Al^{3+}$ concentrations are used in Equation 21 to estimate the silica concentration that is used to*
*quantify the silica 'brake' on the mineral weathering reactions.*

Figure 10a shows a plot illustrating the fate of silica in the dissolution process. Only a small part of the aqueous
aluminium and aqueous silica produced by the dissolution of minerals remain in solution. Most precipitates
out as secondary phases. Figure 10b shows how the aluminium and silica concentrations are estimated in the
model. To estimate $[Al^{3+}]$ and $[SiO_2]$ in the above equations we assume the systems are close to equilibrium
with a gibbsite-like and a kaolinite-like phase. Thus we assume that aluminium precipitates out from the
solution, controlled by something that appears to be gibbsite-like; it is likely something amorphous of unknown
composition, see Alveteg et al. (1995). The "Gibbsite" reaction is:

$Al^{3+} + 3\ OH^- = Al(OH)_3$                                                      (18)

Leading to the "Gibbsite" expression:

$[Al^{3+}] = K_G * [H^+]^Y$                                                          (19)

where the exponent Y has a value of 2.4 to 3. $K_G$ is the Gibbsite coefficient and defined in the critical loads
mapping manual (Sverdrup et al., 1990). An expression analogous to the Gibbsite approximation is used to
calculate the $SiO_2$ concentration (Equation 22b, below). We assume that the Si will be present as $H_4Si(OH)_4$
in the fluid phase, not upsetting any charge balance constraints. We assume that silica precipitates out,
controlled by what that appears to be kaolinite. As such, there is a similar expression can be used for
approximating the silica concentration:





$$2\ Al^{3+} + 2\ SiO_2 + 6\ OH^- = Al_2Si_2O_5(OH)_4 + H_2O \qquad (20)$$
which gives the apparent equilibrium expressions:
$$[Al^{3+}]^2 * [OH^-]^6 * [SiO_2]^2 = K_{Kaolinite} \qquad (21)$$
And this can be re-arranged to:
$$[SiO_2]^2 = K_{Kaolinite}^2 * \frac{[H^+]^6}{[Al^{3+}]^2} \qquad (22a)$$
which leads to the "kaolinite" expression:
$$[SiO_2] = K_{Kaolinite} * \frac{[H^+]^3}{[Al^{3+}]} \qquad (22b)$$
where $K_{Kaolinite}$ is the equilibrium coefficient being used. Note that the "equilibrium" equations assumed above,
are not true equilibrium, and that kaolinite and gibbsite minerals dissolve very slowly under normal conditions.
Both the "gibbsite" and "kaolinite" mentioned above are crude simplifications, possibly representing an
amorphous precipitate combined with precipitation kinetics and ion exchange (see Alveteg et al., 1995, Rietz
1995, Warfvinge et al., 1996 for more information).
**3.9. The full kinetic expression**
The equations and approximations summarized above leads to the full revised mineral dissolution rate
equations given by
$$r = k_H * \frac{[H^+]^{n_H}}{f_H} + \frac{k_{H_2O}}{f_{H_2O}} + k_{CO_2} * P_{CO_2}^{n_{CO_2}} * \frac{1}{f_{CO_2}}$$
$$+ k_R * [R]^{n_R} * \frac{1}{f_R} + k_{OH} * \frac{[OH^-]^{n_{OH}}}{f_{OH}} \qquad (23)$$
where the retarding or 'brake' functions are given by:
$$f_{H^+} = \left(1 + \frac{[BC]}{C_{BC,H}}\right)^{x_H} * \left(1 + \frac{[Al^{3+}]}{C_{Al,H}}\right)^{y_H} * \left(\left(1 + K_{Si,H} * \left(\frac{[Si]}{C_{Si,H^+}}\right)^{z_H}\right)\right) \qquad (24)$$
$$f_{H_2O} = \left(1 + \frac{[BC]}{C_{BC,H_2O}}\right)^{x_{H_2O}} * \left(1 + \frac{[Al^{3+}]}{C_{Al,H_2O}}\right)^{y_{H_2O}} * \left(\left(1 + K_{Si,H_2O} * \left(\frac{[Si]}{C_{Si,H_2O}}\right)^{z_{H_2O}}\right)\right) \qquad (25)$$
$$f_{CO_2} = \left(1 + K_{CO_2} * \frac{P_{CO_2}}{P_{CO_2\ Limit}}\right)^{n_{CO_2}} * \left(1 + \frac{[BC]}{C_{BC,CO_2}}\right)^{x_{CO_2}} * \left(1 + \frac{[Al^{3+}]}{C_{Al,CO_2}}\right)^{y_{CO_2}}$$
$$* \left(1 + K_{Si,CO_2} * \left(\frac{[Si]}{C_{Si,CO_2}}\right)^{z_{CO_2}}\right) \qquad (26)$$
$$f_R = \left(1 + \frac{[R]}{[R]_{Limit}}\right)^{n_R} * \left(1 + \frac{[BC]}{C_{BC,R}}\right)^{x_R} * \left(1 + \frac{[Al^{3+}]}{C_{Al,R}}\right)^{y_R} * \left(\left(1 + K_{Si,R} * \left(\frac{[Si]}{C_{Si,R}}\right)^{z_R}\right)\right) \qquad (27)$$
$$f_{OH^-} = \left(1 + \frac{[BC]}{C_{BC,OH}}\right)^{x_{OH}} * \left(1 + \frac{[Al^{3+}]}{C_{Al,OH}}\right)^{y_{OH}} * \left(\left(1 + K_{Si,OH} * \left(\frac{[Si]}{C_{Si,OH^-}}\right)^{z_{OH}}\right)\right) \qquad (28)$$
where:
$C_{BC,i}$ is the lower limiting base cation concentration in reaction i,
$C_{Al,i}$ is the lower limiting aluminium concentration in reaction i,



$C_{Si,i}$ is the lower limiting silica concentration in reaction i,
$P_{CO2limit}$ is the lower limiting carbon dioxide partial pressure in reaction i,
$[R]_{limit}$ is the lower limiting organic acid concentration in reaction i as concentration of DOC,
$x_i$ is the base cation brake reaction order for i,
$y_i$ is the aluminium brake reaction order for i
$z_i$ is the silica brake reaction order of i.
$K_{CO2}$ is the $CO_2$ brake coefficient and set to 20.
$K_{Si,i}$ is the silica brake constant for reaction i, set to 100.

Table 2. Alteration series from muscovite, biotite and feldspars to clays, corresponding to Figure 11.

| # | Mineral | Interlayer | Octahedral | Tetrahedral |
|---|---------|-----------|------------|-------------|
| | | Muscovite pathway | | |
| 1 | Muscovite | K | $Al_2$ | $Al_{1.0}Si_{3.0}O_{10}(OH)_2$ |
| 2 | Illite 1 | $K_{0.5}Mg_{0.01}Ca_{0.01}Al_{0.05}$ | $Al_{1.6}Fe_{0.25}Mg_{0.1}Ti_{0.04}$ | $Al_{0.6}Si_{3.4}O_{10}(OH)_2$ |
| 3 | Illite 2 | $K_{0.44}Mg_{0.01}Ca_{0.01}Al_{0.07}$ | $Al_{1.6}Fe_{0.25}Mg_{0.1}Ti_{0.04}$ | $Al_{0.6}Si_{3.4}O_{10}(OH)_2$ |
| 4 | Illite 3 | $K_{0.39}Mg_{0.013}Ca_{0.013}Al_{0.06}$ | $Al_{1.5}Fe_{0.32}Mg_{0.1}Ti_{0.08}$ | $Al_{0.6}Si_{3.4}O_{10}(OH)_2$ |
| 5 | Illitic vermiculite | $K_{0.35}Mg_{0.03}Ca_{0.03}Al_{0.06}$ | $Al_{1.63}Fe_{0.32}Mg_{0.08}Ti_{0.07}$ | $Al_{0.6}Si_{3.4}O_{10}(OH)_2$ |
| 6 | Kaolinite | | | $Al_{2.0}Si_2O_5(OH)_4$ |
| | | Chlorite pathway | | |
| 1 | Chlorite | $Ca_{0.5}Mg_{1.5}$ | $Al_{1.0}Fe_{0.5}\,Mg_{1.5}$ | $Al_{1.0}Si_{3.0}O_{10}(OH)_2$ |
| 2 | Vermiculite 1 | $K_{0.32}Mg_{0.07}Ca_{0.09}Al_{0.05}$ | $Al_{1.52}Fe_{0.35}Mg_{0.1}$ | $Al_{0.6}Si_{3.4}O_{10}(OH)_2$ |
| 3 | Vermiculite 2 | $K_{0.30}Mg_{0.05}Ca_{0.05}Al_{0.05}$ | $Al_{1.55}Fe_{0.32}Mg_{0.05}Ti_{0.06}$ | $Al_{0.6}Si_{3.4}O_{10}(OH)_2$ |
| 4 | Vermiculite 3 | $K_{0.25}Mg_{0.04}Ca_{0.04}Al_{0.08}$ | $Al_{1.55}Fe_{0.32}Mg_{0.05}Ti_{0.06}$ | $Al_{0.6}Si_{3.4}O_{10}(OH)_2$ |
| 5 | Al/OH interlayered vermiculite | $K_{0.11}Mg_{0.04}Ca_{0.04}Al_{0.1}$ | $Al_{1.52}Fe_{0.4}Mg_{0.05}Ti_{0.08}$ | $Al_{0.5}Si_{3.5}O_{10}(OH)_2$ |
| 6 | Kaolinite | | | $Al_{2.0}Si_2O_5(OH)_4$ |
| | | Biotite pathway | | |
| 1 | Biotite | $K_{1.0}Mg_{2.0}$ | $Al_{0.5}Fe_{0.5}Mg_{1.0}$ | $Al_{1.0}Si_{3.0}O_{10}(OH)_2$ |
| 2 | Vermiculite 1 | $K_{0.32}Mg_{0.07}Ca_{0.09}Al_{0.05}$ | $Al_{1.52}Fe_{0.35}Mg_{0.1}$ | $Al_{0.6}Si_{3.4}O_{10}(OH)_2$ |
| 3 | Vermiculite 2 | $K_{0.30}Mg_{0.05}Ca_{0.05}Al_{0.05}$ | $Al_{1.55}Fe_{0.32}Mg_{0.05}Ti_{0.06}$ | $Al_{0.6}Si_{3.4}O_{10}(OH)_2$ |
| 4 | Vermiculite 3 | $K_{0.25}Mg_{0.04}Ca_{0.04}Al_{0.08}$ | $Al_{1.55}Fe_{0.32}Mg_{0.05}Ti_{0.06}$ | $Al_{0.6}Si_{3.4}O_{10}(OH)_2$ |
| 5 | Al/OH interlayered vermiculite | $K_{0.1}Mg_{0.04}Ca_{0.04}Al_{0.1}$ | $Al_{1.52}Fe_{0.4}Mg_{0.05}Ti_{0.08}$ | $Al_{0.5}Si_{3.5}O_{10}(OH)_2$ |
| 6 | Kaolinite | | | $Al_{2.0}Si_2O_5(OH)_4$ |
| | | Feldspar pathway | | |
| 1 | Feldspar | K, Na, Ca | | $Al_1Si_3O_8$ |
| 2 | Sericite | $Na_{0.1}K_{0.75}$ | $Al_{1.9}Mg_{0.1}$ | $Al_{0.84}Si_{3.16}O_{10}(OH)_2$ |
| 3 | Sericitic vermiculite 1 | $K_{0.3}\,Mg_{0.02}Ca_{0.05}$ | $Al_{0.02}$ | $Al_{1.0}Si_3O_{10}(OH)_2$ |
| 4 | Sericitic vermiculite 2 | $K_{0.1}\,Mg_{0.05}Ca_{0.02}$ | $Al_{0.05}$ | $Al_{1.0}Si_3O_{10}(OH)_2$ |
| 5 | Al/OH interlayered vermiculite | $K_{0.1}Mg_{0.04}Ca_{0.04}Al_{0.1}$ | $Al_{1.52}Fe_{0.4}Mg_{0.05}Ti_{0.08}$ | $Al_{0.5}Si_{3.5}O_{10}(OH)_2$ |
| 6 | Kaolinite | | | $Al_{2.0}Si_2O_5(OH)_4$ |


## 3.10. Secondary phases in the soil

A significant fraction of primary minerals dissolve incongruently to alteration minerals often referred to as
secondary minerals and clays. Both terms are inconsistently used in the literature, and thus we define them as
follows: We define clay minerals by their composition (kaolinite, gibbsite, quartz) and as listed in Table 3.
This approach is thus not based on particle size, but on the molecular crystalline structure. Secondary minerals
are formed in either two ways; a mineral that has been altered significantly in situ as is described in Table 2,
for example when muscovite is altered through a series of illite and vermiculite phases and finally to kaolinite
as the end product. Vermiculite, illite, montmorillonite are minerals of variable composition that are often
called clays, In the soil, amorphous phases are composed of aluminium, silicate and soil organic substances.
These amorphous phases slowly change composition as the organic matter decomposes and a more solid
structure emerges. The alteration series from muscovite, biotite and feldspars to clays, are illustrated
schematically in Figure 11 and listed in Table 2. The concept behind Table 2 is that as these minerals go
through incongruent dissolution (alteration), they become depleted in certain ions (like Ca, Mg, K or Na, and
depending on pH, in aluminium (at low pH) or silica (at high pH), but the crystal structure remains constant.
Thus the crystal lattice destruction rate remains, but the base cation content of this structure becomes poorer,
yielding less cations and less acidity neutralization. We have simplified this process down to 4 pathways, the
muscovite pathway, the chlorite pathway, the biotite pathway and the feldspar pathway – see Table 2.
Muscovite changes through a series of alteration reactions to illite and finally to kaolinite. Chlorite alters to
vermiculites and finally to kaolinite. Biotite goes through a series of alterations to vermiculite and kaolinite.





Feldspars go through alterations, K-Feldspars through sericites and plagioclases to vermiculites (Holmqvist
2004, Holmqvist 2002, 2003).

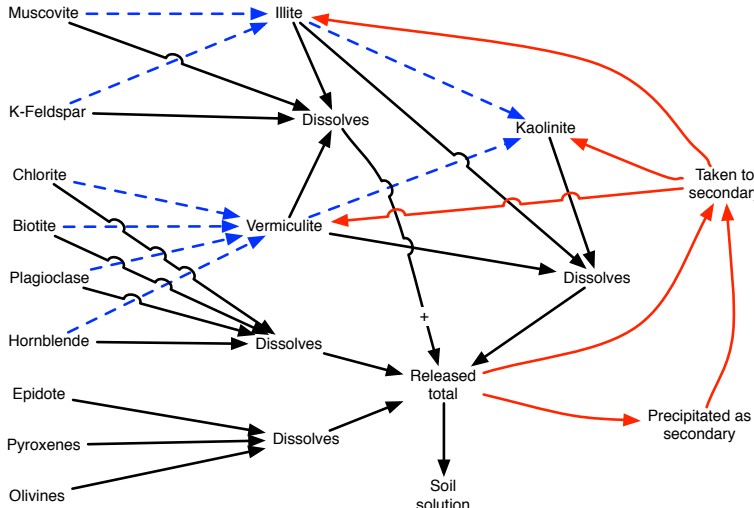

*Figure 11. The alteration sequence developed for primary mineral towards alteration minerals, of which some*
*are clay minerals. All minerals that dissolve contribute to the precipitation of secondary minerals.*

**3.11. The parameterization of the kinetic rate equations**
The original PROFILE database had kinetic data for 59 different minerals, including about 25 different
carbonates and some artificial silicates. New data from our own experiments (Sverdrup 1998, 1996, Sverdrup
and Alveteg 1998, Holmqvist et al., 2002, 2003; Sverdrup and Holmqvist 2004) and from the literature[2] have

[2]Examples are the following list of articles and studies we have used, but not limited to: Ajemba and Onokwuli 2012, Alekseyev 2007, Alexeyev et al., 1997, Amram and Ganor 2005, Amrhein and Suare 1992, Anbeek 1992a,b, Anbeek et al., 1994, Aradottir et al., 2013, Bandstra et al., 1998, Beig and Lüttge 2006, Bengtsson and Sjöberg 2009, Berg and Banwart 1994, 2000, Bibi et al., 2010, Bickmore et al., 2006, Blake and Walther 1996, Blum and Stillings 1995, Blum and Lasaga 1988, 1991, Blum 1994, Brady and Walther 1992, Bray et al., 2015, Brandt et al., 2005, Brantley 2003, 2008a,b, Brantley and Stillings 1994, 1996, Brantley and Chen 1995, Brantley and Conrad 2008, Brady and Walther 1992, Braun et al., 2016, Bray 2015, Cama et al., 2000, Carrol and Knauss 2005, Carrol and Walther 1990, Carrol and Smith 2013, Casetou-Gustafsson et al., 2018, Casey et al., 1991, Casey and Sposito 1992, Casey and Westrich 1992, Chaïrat et al., 2007, Chen and Brantley 1997, 1998, 2000, Chin and Mills 1991, Critelli et al., 2015, 2014, Cotton 2008, Crundwell 2013, 2014a,b,c,d, 2015a,b, 2017, Daval et al., 2010a,b, 2013, Devidal et al., 1997, Diedrich et al., 2014, Dixit and Carrol 2007, Dove and Crerar 1990, Dorozhkin 2012, Dresel 1989, Drever et al., 1994, 1996, Drewer and Clow 1995, Drewer and Zobrist 1992, Drever and Stillings 1997, Dorozin 2012, Duckworth and Martins 2003a,b, Fernandez-Bastero et al., 2008, Fischer and Liebscher 2014, Finlay et al., 2010, Fouda et al., 1996a,b, Frogner and Schweda 1998, Fumuto et al., 2001, Gahrke et al., 2005, Ganor et al., 2005, Gautier et al., 1994, Gislasson and Hans, 1987, Gislasson and Oelkers 2003, Gislasson et al., 1996, Godderis et al., 2006, Glover et al., 2003, Godderis et al., 2006, Golubev et al., 2004, 2005, Goyne et al., 2006, Gudbrandsson et al., 2011, 2014, Gustafsson and Puigdomenech 2003, Hamilton et al., 2000, 2001, Hangx and Spiers 2009, Harouiya et al., 2007, Harouiya and Oelkers 2004, Haug et al., 2010, Hausrath et al., 2009, Hayashi and Yamada 1990, Helgeson et al., 1984, Hellmann 2007, 2006, 2010, Hilley et al., 2010, Holmqvist and Sverdrup 2001, Holmqvist et al., 1999, 2002, 2003, 2004, Hodson 2006a,b, Hodson and Langan 1999, Hodson et al., 1996, 1997, Hänchen et al., 2006, Huertas et al., 1999, 2001, Jin et al., 2011, Johnsson et al., 1992, Johnson et al., 2014, Jonckbloedt 1998, Jönsson et al., 1995, Kalinowski 1997, Kalinowsli and Schweda 1995, Kalinowski et al., 1998, Knauss et al., 1993, Køhler et al., 2003, 2005, Kuwahara 206a,b, 2008, Labat and Viville 2006, Lagache 1965, Langan et al., 1996a, b, Lartigue 1994, Lasaga 1995, 1998, Lowson et al., 2005, 2007, Lazaro et al., 2015, Lu et al., 2013, 2015, Ludwig et al., 2013, Maher 2010, Malmstrøm and Banwart 1997, Malmström et al., 1996, Maurice et al., 2002, Mazer and Walther 1994, McCourt and Hendershot 1992, Metz et al., 2005, Meyer 2014, Mongeon et al., 2007, Murakami et al., 1998, Murphy and Helgesson 1987, Murphy et al., 1992, 1996, Nagy 1995, Nagy and Lasaga 1992, Nagy et al., 1991, Navarre-Sitchler and Thyne 2007, Nesbitt et al., 1991, Nyström-Claesson and Andersson 1996, Numan and Weaver 1969, Oelkers 2001a,b,, Oelkers and Schott 1995a,b, 1998, 2001, Oelkers et al., 1994, 2008, Oelkers and Gislasson 2001, Olsen 2007, 2008, Olsson 2007, Opolot and Finke 2015, Oxburgh 1991, Oxburgh et al., 1994, Paces 1983, Palandri and Kharka 2004, Pokrowsky and Schott 2000a,b, 2002, Pokorowsky et al., 2004, Poulson et al., 1997, Prajapati et al., 2014, Price et al., 2005, Pigiobbe et al., 2009, Ragnarsdottir 1993, Ragnarsdottir and Graham 1996, Raschmann and Fedorockova 2008, Rietz 1995, Rimstidt et al., 2012, Ross 1969, Rosso and Rimstidt 1999, Rozalen et al., 2014, Running and Gower 1991, Saldi et al., 2007, Sanemasa and Katura 1973, Schnoor 1990, Schofield et al., 2015, Schott et al., 2009, 2012, Smith et al., 2013, Smits and



been considered to upgrade this database for this study. Care of these new data we have obtained rate
parameters for about 107 different silicate or aluminium minerals and 6 generic carbonates. Of these minerals,
the regression of ~20 have yet to be published. In due time, these will get their own proper publications, it is
beyond the scope of this study to do them in detail. Data and records from unpublished experiments and
experiment evaluations by Sverdrup and Holmqvist are available on paper records held in a large number of
binders at the Inland University of Applied Sciences, at Hamar, Norway. These data are no longer available in
digital form due to computer system changes and data filing format changes that have occurred during the last
20 years. This documentation could be available in 1-2 years time, provided that funding for the redigitalization
work can be obtained. Rather some selected examples are presented below. The estimation of rate parameters
was performed using the complete rate equation 1 and equations 23-28. As such, for a successful regression
of experimental data, the rate must be known, along with the concentrations of all reactants at the conditions
that rate was observed including [H$^+$], pCO$_2$, [R], [OH$^-$], as well as the reaction products in solution potentially
contributing to retarding the dissolution reaction; [Ca$^{2+}$], [Mg$^{2+}$], [K$^+$], [Na$^+$], [Al$^{3+}$], [Al(OH)$_4^-$], [H$_4$SiO$_4$]
(Sverdrup 1990, Sverdrup and Warfvinge 1995). The experiments must have been performed over sufficient
reaction conditions for the parameters in Equation 29 to be estimated. In some cases, the data from different
experimental studies were combined to determine rate parameters or a reaction orders. During the regression
process, experimental studies with insufficient data or documentation were omitted, unless the gap could be
bridged with reasonable assumptions. Data regression was performed by rearranging equation (22) to:
$$k_H * \frac{[H^+]^{n_H}}{f_H} = r_{Observed} - (\frac{k_{H_2O}}{f_{H_2O}} + k_{CO_2} * \frac{P_{CO_2}^{n_{CO_2}}}{1 + K_{CO_2} * P_{CO_2}^{n_{CO_2}}} * \frac{1}{f_{CO_2}}$$

$$+ k_R * \frac{[R]^{n_R}}{1 + K_R * [R]^{n_R}} * \frac{1}{f_R} + k_{OH} * \frac{[OH^-]^{n_{OH}}}{f_{OH}}) \qquad (29)$$

In the acid to neutral pH range, such as pH < 7, this equation can be simplified in most instances by removing
the OH-reaction to get (Sverdrup 1990):
$$k_H * \frac{[H^+]^{n_H}}{f_H} = r_{Observed} - (\frac{k_{H_2O}}{f_{H_2O}} + k_{CO_2} * \frac{P_{CO_2}^{n_{CO_2}}}{1 + K_{CO_2} * P_{CO_2}^{n_{CO_2}}} * \frac{1}{f_{CO_2}}$$

$$+ k_R * \frac{[R]^{n_R}}{1 + K_R * [R]^{n_R}} * \frac{1}{f_R}) \qquad (30)$$

and the in the acid pH range (pH<4), this may be reduced to:
$$k_H * \frac{[H^+]^{n_H}}{f_H} = r_{Observed} \qquad\qquad\qquad (31)$$

By entering the concentrations of H$^+$, base cations, aluminium and silica into these equations, we can determine
the rate coefficient, $k_H$, and $f_{H+}$. When the experiment was performed in the absence of organic acids, as is
often the case, Equation (29) reduces to:
$$k_H * \frac{[H^+]^{n_H}}{f_H} = r_{Observed} - (\frac{k_{H_2O}}{f_{H_2O}} + k_{CO_2} * \frac{P_{CO_2}^{n_{CO_2}}}{1 + K_{CO_2} * P_{CO_2}^{n_{CO_2}}} * \frac{1}{f_{CO_2}}) \qquad (32)$$


Wallander 2016, Smits et al., 2014, Soler et al., 2008, Stephens and Hering 2003, Stillings and Brantley 1995, Stillings et al., 1996,
Stockmann et al., 2008, Stumm and Wollast 1990, Stumm and Wieland 1990, Sverdrup 1990, 1996a,b, 1998, 2009, Sverdrup and
Bjerle 1982, Sverdrup and Alveteg 1998, Sverdrup and Holmqvist 2016, Sverdrup and Warfvinge 1992a,b, 1995, Sverdrup et al., 1986,
1987, 1995,a,b, 1998, 2002, 2006, 2008, 2010, Traven et al., 2005, Swoboda-Collberg and Drever 1993, Taylor et al., 1999, 2000,
Taylor and Blum 1995, Taylor et al., 2017, Techer  et al., 2007, Teir et al., 2007, Terry 1983a,b,c, Terry and Monhemius 1983, Thom
et al., 2013, Valsami-Jones et al., 1998, Turpault and Trotignon 1994, Valsami-Jones et al., 1998, Voltini et al., 2012, Wang and
Giammar 2012, Wang et al., 2017, Warfvinge and Sverdrup 1992,a,b,c,d, 1993, 1995, Warfvinge et al., 1987, 1992, 1993, 1996, 2000,
Weissbart and Rimstidt 2000, Welch and Ullman 1993, 1996, 2000, Westrich et al., 1993, White and Brantley 1995, 2003, White and
Blum 1995, White et al., 1999, Whitfield et al., 2009, 2010, Wogelius and Walther 1991, 1992, Wolff-Boenisch et al., 2004a,b, 2011,
Wood et al., 1999, Xie and Walter 1994, Yadaw and Chakrapani 2006, Yadaw et al., 2000, Yang and Steefel 2008, Yoo et al., 2009,
Yu et al., 2016, 2017, Zabowski et al., 2007, Zhang and Bloom 1999a,b, Zhang et al., 1996, 2015, Zhang et al., 2013, Zhang and Lüttge
2017, 2009a,b, Zhu et al., 2010, Zassi 2009, Zavodsky et al., 1995, Zysset and Schindler 1996).



Some experiments were conducted at very low or with no dissolved $CO_2$ present and with organic ligands
absent. In such cases, Equation (29) reduces to (Sverdrup 1990, Chin et al., 1991):

$$r_H \;=\; k_H * \frac{[H^+]^{n_H}}{f_H} \;=\; r_{Observed} - \frac{k_{H_2O}}{f_{H_2O}} \qquad (33)$$

In this latter case, two reactions influence mineral dissolution rates: 1) the $H^+$ reaction, and 2) the water
reaction. The variation of rates as a function of pH at such conditions consists of a 'flat part' where rates are
controlled by the water reaction (Figure 12). At these conditions, by entering the concentrations of retarding
base cations, aluminium and silica, the rate coefficients can be determined. In the semi-neutral region (pH 6-
8), the expression may be a flat line and the rate expression is reduced to:

$$r_{Observed} = \frac{k_{H_2O}}{f_{H_2O}} \;+\; k_{CO_2} * \frac{P_{CO_2}^{n_{CO_2}}}{1 + K_{CO_2} * P_{CO_2}^{n_{CO_2}}} * \frac{1}{f_{CO_2}} \;+\; k_R * \frac{[R]^{n_R}}{1 + K_R * [R]^{n_R}} * \frac{1}{f_R} \qquad (34)$$
When neither organic ligands nor $CO_2$ is present, and in the pH range of 6-8, this is reduced to:

$$r_{Observed} = \frac{k_{H_2O}}{f_{H_2O}} \qquad (35)$$
With only organic acid ligands but no $CO_2$ present, and in the pH range of 6-8, the rate expression becomes:

$$r_{Observed} = \frac{k_{H_2O}}{f_{H_2O}} \;+\; k_R * \frac{[R]^{n_R}}{1 + K_R * [R]^{n_R}} * \frac{1}{f_R} \qquad (36)$$

In the far alkaline region (pH 10-14), where we may assume that the OH- reaction will be dominant, the rate
expression reduces to:

$$k_{OH} * \frac{[OH^-]^{n_{OH}}}{f_{OH}} = r_{Observed} \qquad (33)$$

By entering the concentrations of base cations, aluminium and silica, $f_{OH}$ can be determined and the rate
coefficient, $k_{OH}$, and reaction order, $n_{OH}$ be determined. The reaction order $n_H$ and the coupled $n_{OH}$ for the $H^+$
and the $OH^-$ reaction is derived from plots of the rate versus the solution pH

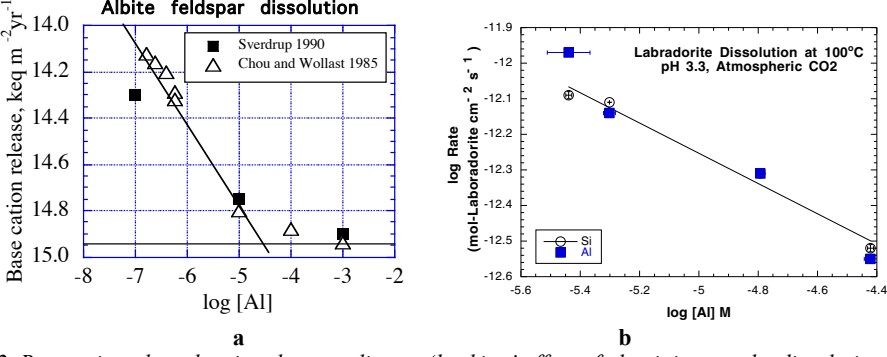

a                                    b
*Figure 12. Regression plots showing the retarding or 'braking' effect of aluminium on the dissolution rate of*
*albite. The figures were adapted from Sverdrup (1990). The decrease of rates as a function of aqueous*
*aluminium concentration (the aluminium brake) is very prominent in the range of log $[Al^{3+}]$ from -7 to -4.5.*
*Aluminium concentrations are in kmol $m^{-3}$. The figures were adapted from (a) Sverdrup et al. (1990) and from*
*(b) Carroll and Knauss (2001). For further information, see Sverdrup (1990) and Sverdrup and Warfvinge*
*(1995).*

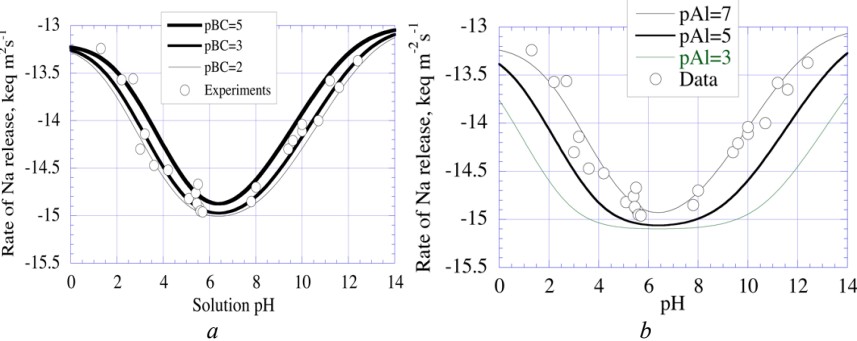

a                                                                      b
*Figure 13. The effect on the base cation (a) and the aluminium concentration (b) on the dissolution rate of*
*albite. (Sverdrup 1990). The circles represent the data from experiments, the solid lines the model simulations.*

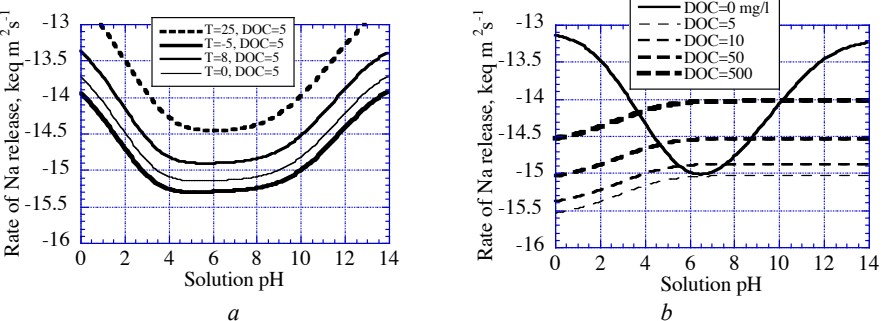

a                                                                      b
*Figure 14. The effect on the base cation (a) and the aluminium concentration (b) on the dissolution rate of*
*albite. The solid line is the reaction rate without $CO_2$ or organic acid ligands.*
Figure 13 shows diagrams used to quantify the retarding effect of aluminium on the dissolution rate
of albite feldspar. The figures were adapted from Sverdrup (1990) and the work prepared for Sverdrup and
Warfvinge (1995) and Sverdrup et al., (2009).  Similar results for aluminium were found by Oelkers (2001),
Oelkers and Gislasson (2001), Oelkers and Schott (2001, 1995a,b), Oelkers et al., (1999) for several minerals.
The aluminium brake is very prominent in the range of log [Al] from -7 to -4.5. For further information, see
Sverdrup (1990) and Sverdrup and Warfvinge (1995).
The reaction order for the organic acid reaction is derived from experiments where only the
concentration of organic ligand, [R], has been varied. This was found to be $n_R$=0.5 on most experiments and
this exponent value was universally adopted, suggesting a divalent ligand being the reactive agent (Sverdrup
1990, Sverdrup and Warfvinge 1995, Oelkers and Schott 1998).
The reaction order $n_{CO2}$ for the reaction with $CO_2$ is difficult to constrain, as very few experiments that
allow it to be determined are available (Daval et al., 2013, Berg and Banwart 2000, Golubev et al., 2005,
Fernandez-Bastero et al., 2008, Hangx and Spiers 2009, Lagache 1965, Wogelius and Walther 1991, Wolff-
Boenisch et al., 2011, Stephens and Hering 2004, Sverdrup 1990).  The few experiments that are available
often gives conflicting results. Moreover, many experiments dealing with the effect of $CO_2$ on weathering do
not have the required resolution to allow data regression. For the minerals where the $CO_2$ has little or no effect,
this is fine, but for some it is. It was found to be $n_{CO2}$=0.6 and was universally adopted. Sometimes these
parameterizations can be determined by making single factor plots, but more often, the whole model must be
used to recreate the experiments, taking many factors into account simultaneously. Figure 13 shows the effect
on the base cation (a) and the aluminium concentration (b) on the dissolution rate of albite. Various plots were
used to help data interpretation. Figure 14-15 illustrates how the model was used to plot different combinations
of conditions, to investigate how distinct factors affect the weathering rates. The experimental data were
overlaid in such diagrams (Figures 16-20) to help interpretation towards generating kinetic parameters (rate
coefficients and reaction orders), for example the combination of different organic acid ligand concentrations
and aluminium concentrations.  The last diagram, on the lower right of Figure 15, shows the combination of
different combinations of organic acid ligand concentrations and $CO_2$ pressures in atmospheres.  Figure 16



shows the effect on rates of the base cation (a) and the aluminium concentration (b) on the dissolution rate for
albite. The circles represent the data from experiments.
A further example of parameterization efforts is shown in Figure 16 for the case of hornblende
dissolution rate data reported by from Holmqvist and Sverdrup (2004) and Holmqvist et al. (2002, 2003).
Figure 16a and 16b shows these data as a function of pH. The figures were adapted from Holmqvist et al.,
2003). Figure 16c shows the retarding effect of aluminium on the dissolution rate of hornblende, adapted from
Holmqvist et al. (2003). Figure 16d shows a three-dimensional plot for the dissolution rate of hornblende, as
a function of solution pH and aluminium concentration (Sverdrup, 1990).
In total, the dissolution rate of hornblende is defined by at least 8 and perhaps 9 different chemical
factors including pH, Ca+Mg, K, Na, Al, DOC, $CO_2$, Si and sometimes Fe concentrations, and in addition to
mineral surface area, soil wetting degree and temperature. For example changes in the aluminium
concentration, can change the weathering rate by several orders of magnitude. Additional examples are
presented in Figs. 17-21.

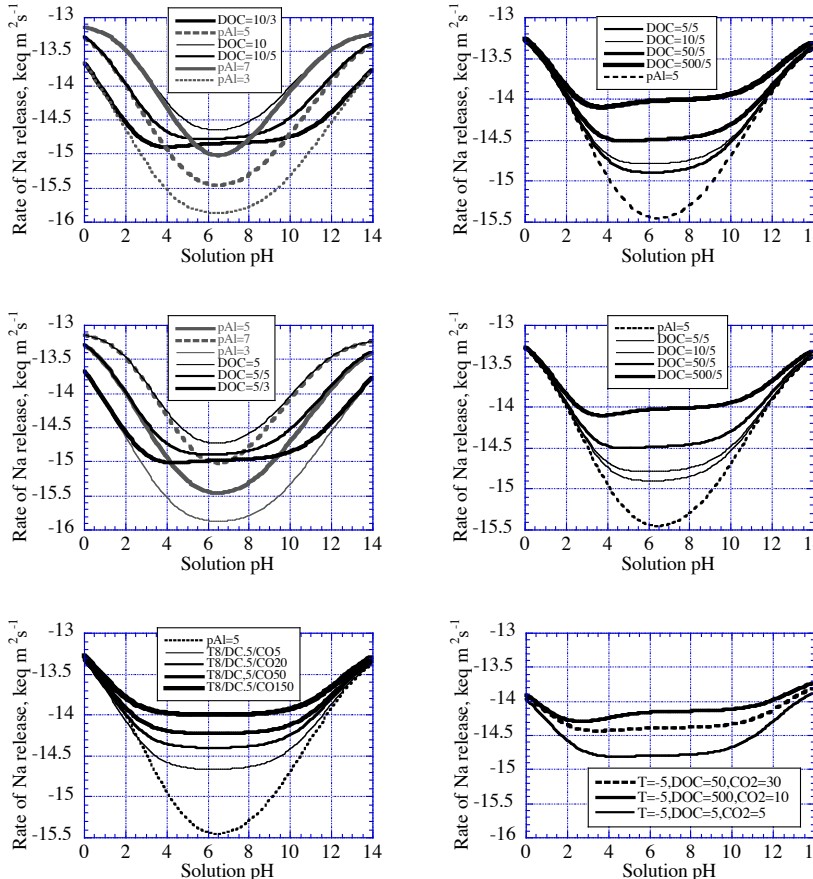

*Figure 15. The weathering rate model was used to plot different combinations of conditions, to investigate the*
*different shapes the weathering rate dependency can change (See Figure 7 and 9 for how the principle works).*
*The experimental data were overlaid in such diagrams, to help retrieve kinetic parameters (e.g. rate*
*coefficients and reaction orders). The last diagram, lower right, shows the combination of different*
*combinations of organic acid ligand concentrations and $CO_2$ pressures in atmospheres.*
Figure 17 shows a typical example of data generated for different minerals during the 1996-2002 field
seasons using a continuous, flow through, fluidized bed, with constant concentration feed solutions. Figure 18
shows the experimentally measured dissolution rates of epidote, after Holmqvist et al. (2003), as a function of
pH according to a number of weathering experiments.





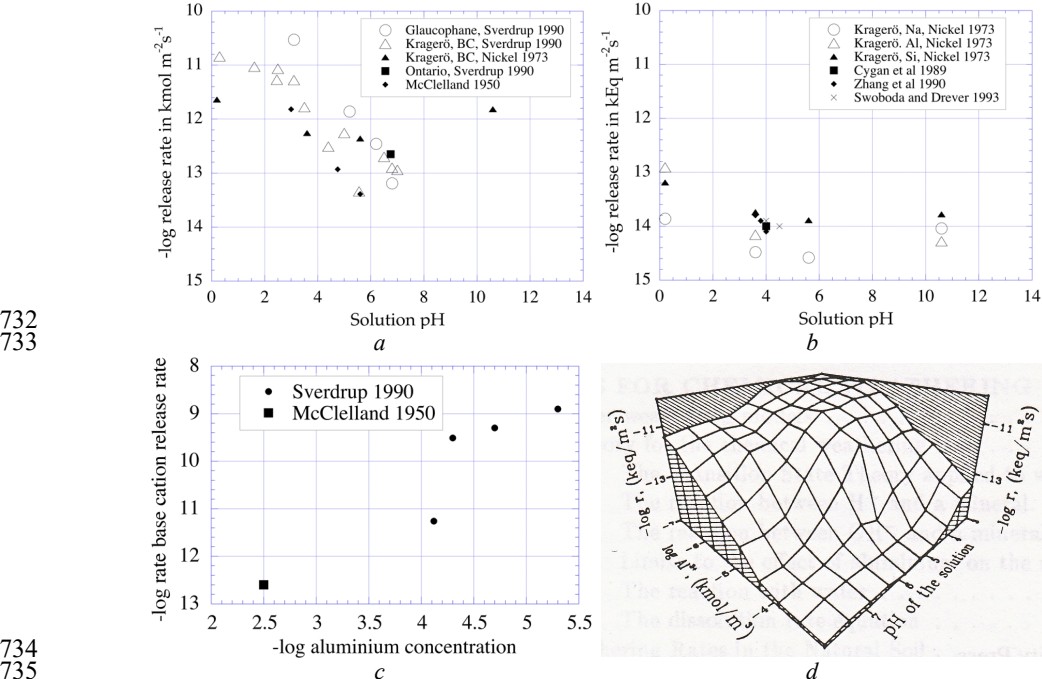

Figure 16. Diagram (a) shows the dissolution rate of minerals presented as base cation release rates as a function of pH and (b) shows the dissolution rate for hornblende as a function of solution pH, but under different experimental conditions (Adapted from Sverdrup, 1990). Diagram (c) shows the retarding effect of aluminium on the dissolution rate of hornblende. (Adapted from Holmqvist et al., 2003). Diagram (d) shows a three-dimensional plot for the dissolution rate of hornblende, as a function of solution pH and aluminium concentration (Adapted from Sverdrup, 1990).

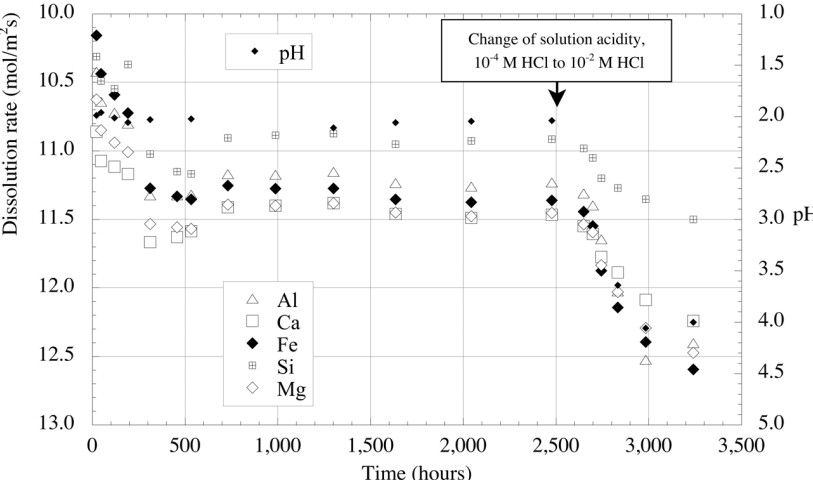

Figure 17. Typical example of dissolution rate data generated for epidote during 1996-2002 using a continuous, flow through, fluidized bed, with constant concentration feed solutions (Holmqvist 2002, 2003). All relevant constituents of the mineral were monitored in the aqueous solution in the experiment.


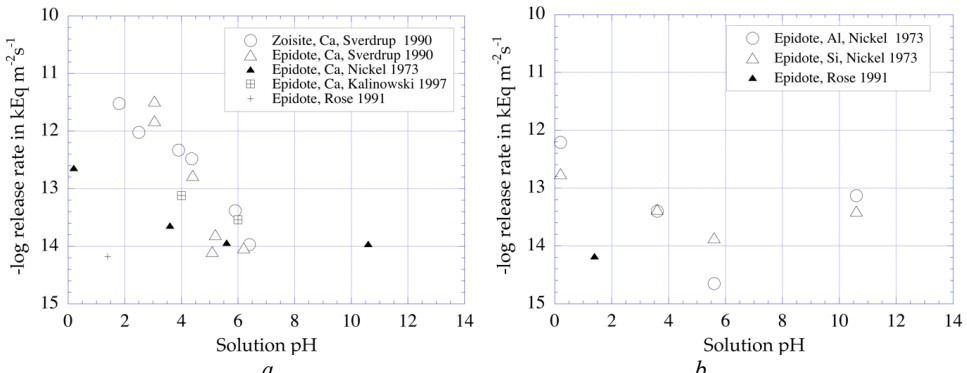

*Figure 18. Epidote dissolution rate versus pH according to experiments reported by Holmqvist and Sverdrup and other literature sources data.*

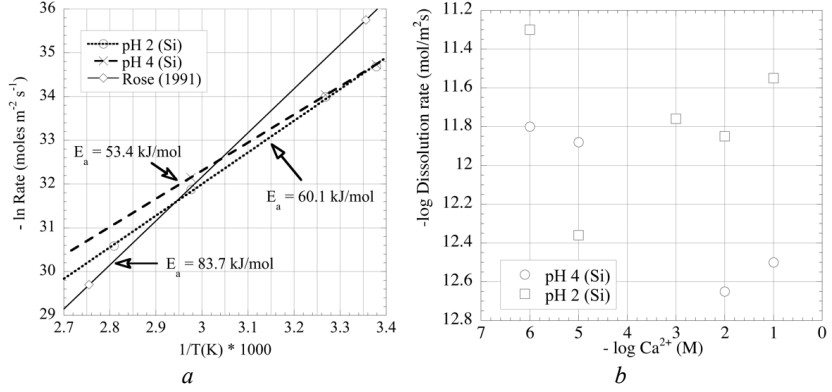

*Figure 19. a) Estimates of the energy of activation for the dissolution of epidote. (b) the dependence of the rate of epidote on the calcium concentration at pH 2 and pH 4 (From one series of experiments by the authors).*

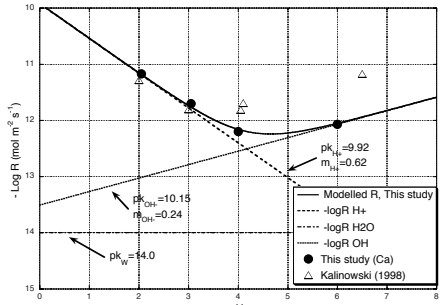

*Figure 20. Hornblende dissolution rate data from Holmqvist and Sverdrup (2004) and Holmqvist et al, (2002, 2003) suggests that an arithmetic addition gives a good fit to the data.*

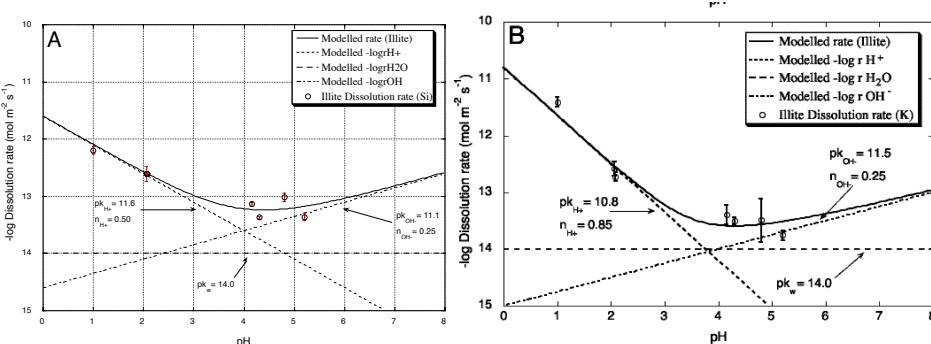

*Figure 21. Diagram A show regression results from hornblende dissolution rates, diagram (B) shows regression results from a natural illite mineral dissolution extracted from an agricultural soil sample taken at the agricultural research site at Lanna, Uppsala, Sweden. Data from Holmqvist and Sverdrup (2004) and Holmqvist et al. (2002, 2003)*

The release of all relevant ions was monitored by frequent sampling during the experiments. Figure 19a shows the activation energy for the dissolution of epidote. The dependence of the dissolution rate of epidote on the calcium concentration at pH 2 and pH 4 is shown in Figure 19b. Figure 20 and 21 shows data from Holmqvist and Sverdrup (2004) and Holmqvist et al. (2002, 2003) confirming that an arithmetic addition of the various rate contributions gives the best fit of the data, consistent with the principle shown in Figure 7. Figure 21 shows results from hornblende, the bottom diagrams (A, B) shows results from a natural illite mineral extracted from an agricultural soil sample taken at the agricultural research site at Lanna, Swedish Agricultural University, Uppsala, Sweden. Model lines were fitted to the data points to set the rate coefficients and reaction orders. Note that a complete set of kinetic parameters could not be directly generated for all minerals due to incomplete experimental data sets.

Estimates for some of the rate coefficients in Table 3 were based on mineral crystal structure analogies (Sverdrup 1990, Holmqvist 2003, Sverdrup and Stiernquist 2002, Crundwell 2014a,b, 2016), crystal bond energies (Sverdrup 1990, Velbel 1999, Crundwell 2014b, 2016) and comparison with analogue minerals. For many of the minerals, the dissolution kinetics patterns are very consistent. The dissolution rate curve shapes of feldspars, garnets, olivines, zoisites allow for this, but also muscovite to illite alteration series, K-feldspar to sericite alteration series.

For example, for the feldspars, we have sufficient data to parameterize the $H^+$ reaction for 5 different plagioclases, the mixed composition plagioclases from albite to anorthite. A plagioclase with a different composition will be interpolated between these as shown in Figure 22. We have the same situation for K-feldspars with increasing contents of Na and Ca, giving a systematic shift in parameter values. The pattern is very consistent as can be seen from the diagrams shown in Sverdrup (1990). However, for the $OH^-$ reaction we have less information. The $OH^-$ rate equation is theoretically linked to the $H^+$ reaction, but more sensitive to the concentration of the same base cation as in the mineral (Na, K, Ca). With the available data and the theoretical link, we can estimate the missing parameters for some of the feldspars. There is a similar situation for the $H_2O$ reaction. We have the experiments that allow it to be constrained for most of the feldspars, and the shifts between the feldspars are systematic and consistent.

For the reaction with organic acid ligands, the situation is more complex. Many of the dissolution experiments run with organic acids were poorly documented, and getting accurate parameterization from them is not possible. For some minerals like feldspars and olivine, some experimental results are available (Stillings et al., 1996 is one example for feldspar) that allow for kinetic parameter estimation. They found $n_R = 0.75$ in the range pH 3-7. For other minerals, we have only single experiments, scattered among some few minerals. Few experiments are available, and for only a few types of minerals. These provide suggestions on what the parameter values probably would be. The situation is similar for the reaction between the mineral surface and $CO_2$. The reaction seems to be weak, and only play a role at elevated pressures. For example, Wang (2013), based on the experimental results of Hänchen et al. (2006) concluded there was no effect of the $CO_2$ reaction on olivine dissolution rates beyond the effect caused by $CO_2$ on pH.

Retrieved kinetic parameters are provided in Table 3. Parameters that are derived directly from of one or more set of experimental data are given in **bold** font. The kinetic parameters that were estimated are shown in roman font. The minerals in this table are divided into 11 groups of basic crystalline structures. Some of the




minerals inside each group have large commonalities with respect to how they dissolve, and this was of great
help in parameter estimation table.
For feldspars, nesosilicates and phyllosilicates, the amount of experimental data available makes the
retrieved parameters robust. If three different compositions of basically the same type of mineral, A, B and C,
are known to have relative rates A>B>C, and we have the kinetic parameters for A and C, then we can be
fairly certain that the values for the kinetic parameters for B are constrained between A and C (see Figure 22).
If they are close, then we would be able to set parameters for B fairly accurately, even with sparse experimental
data for B. This is the case for many minerals (In particular feldspars, nesosilicates, phyllosilicates), and is a
way to get more parameterization from limited experimental data sets.  For the pyroxenes and amphiboles, the
experiments indicate that the minerals behave with some variety depending on their composition, making the
estimates less accurate. But, many pyroxenes are mixtures of definable end members and this was utilized to
interpolate and estimate missing parameters.

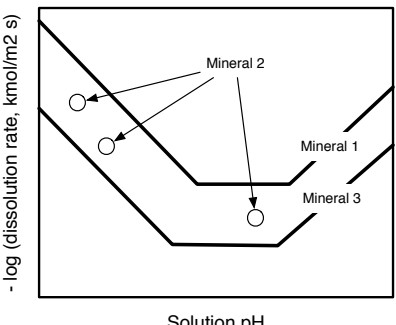
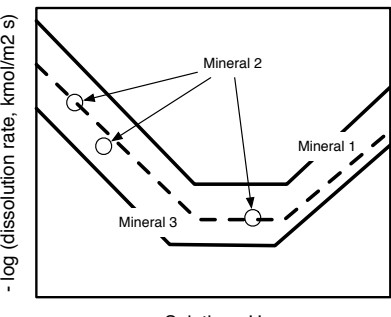

*a: Data points drawn in*                         *b: Interpolate line*
*Figure 22. Some mineral groups have very similar dissolution rate behaviours. Such similarities can be used*
*to interpolate between them (b) when we have intermediate minerals with only a few data points available (a).*
Nevertheless all parameters in Table 3 together with their kinetic expressions should be further validated as
additional experimental data become available. The ultimate test of the kinetics equations and parameters are
how well they describe both laboratory experiments and field data where independent estimates of the
weathering rate are available. Such tests have been generally successful (see the publications referred to earlier,
and Erlandsson Lampa et al., 2016, 2019), suggesting that the combined methodology (experiments,
analogues, interpolations, estimates based on theoretical rescaling, predictions made based on crystal bond
energies) have captured the kinetics sufficiently well. More on this will be forthcoming in future publications.

## 4. Results
### 4.1. Kinetics and parameterization

The tabulated kinetic coefficients are the major result of this report and they are provided in the Tables
1, 3 and 4. In total the dissolution kinetics parameterization for 112 minerals are provided. Erlandsson-Lampa
et al. (this volume) tested the application of these values using the parameters on the Svartberget research site
as a field evaluation.
The parameters in Table 3 are for a temperature of 8°C and standard atmospheric pressure.. The
following default approximations were adopted due to the lack of data; $C_{Al}$ for the $H^+$-reaction is taken to be
equal to $^1/_3$ of the $C_{Al}$ for the $OH^-$-reaction. $C_{BC}$ for the $H^+$-reaction is taken to be $^1/_3$ of the $C_{BC}$ for the $OH^-$-
reaction. The retarding reaction orders for base cations (x), aluminium (y) and silicate (z) have been extracted
from separate datasets and experiments where it was possible to separate out the effect of silicate alone, having
subtracted the effect of base cations and aluminium first. Default values were computed and scaled with
Madelung crystal lattice site energy (See Sverdrup 1990 and Velbel 1999 for how a-priori weathering rate
coefficient estimates are made from crystal properties). Irreversible dissolution implies that the mineral cannot
be formed from solution under soil conditions, and that there is no saturation concentration or back reaction.
Pokrovsky and Schott (2000) and Rosso and Rimstidt (2000) reports a reaction order of $n_{H+}=0.5$ for forsterite,
but others report $n_{H+}=1.0$ (Grandstaff 1986, Blum and Lasaga 1988, Siegel and Pfannkuch 1984, Sverdrup
1990). $n_{H+}=1.0$ seems to be a property of the nesosilicate group, but there is a possibility that presence of
impurities such as pyroxenes or feldspars in the nesosilicate may give it a different crystal structure and thus
a different $n_{H+}$. Others, Berg and Banwart (2000), report $n_{H+}$ in the range 0.5 to 1, depending on pH.



Table 4 shows the temperature dependencies of the dissolution rates. All variations of rates on
temperature are computed using a modified Arrhenius equation (Sverdrup 1990, 1998, Sverdrup and
Warfvinge 1988, 1992, 1995). Parameters for this equation generated from experimentally measured rates are
shown in bold. Where experimental data were not available, estimates were computed and scaled with
Madelung crystal lattice site energy from garnet (Sverdrup 1990, Velbel 1999). Values in normal font were
estimated from the lattice energies and the properties of the mineral surface. Table 5 shows the stoichiometry
of the minerals considered in this study.

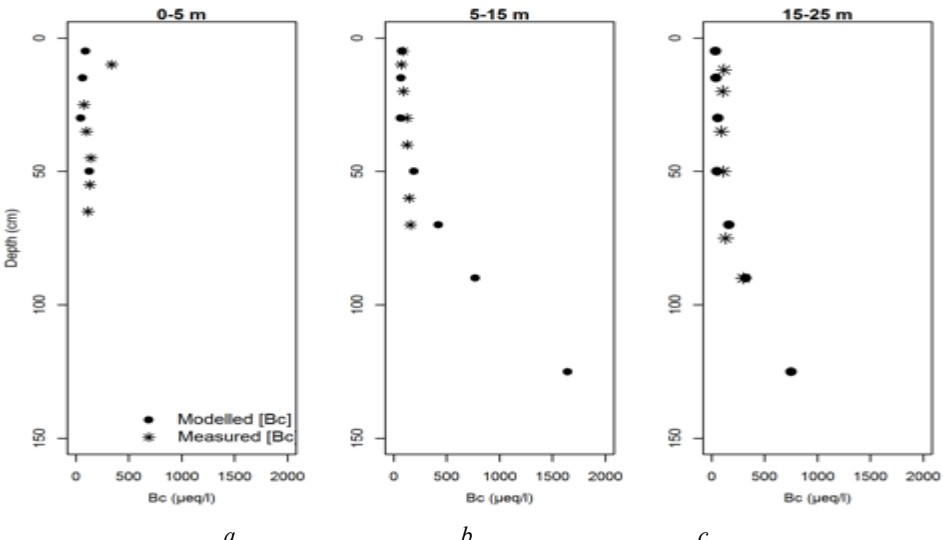

*a*                    *b*                    *c*
*Figure 23. Comparison of calculated with measured base cation concentrations at the Svartberget field site,*
*(Zanchi et al., 2016). Note the base cation concentrations (*[Bc]*) refer to the sum of the concentrations of Na,*
*H, Ca, and Mg in units of microequivalents per litre.*

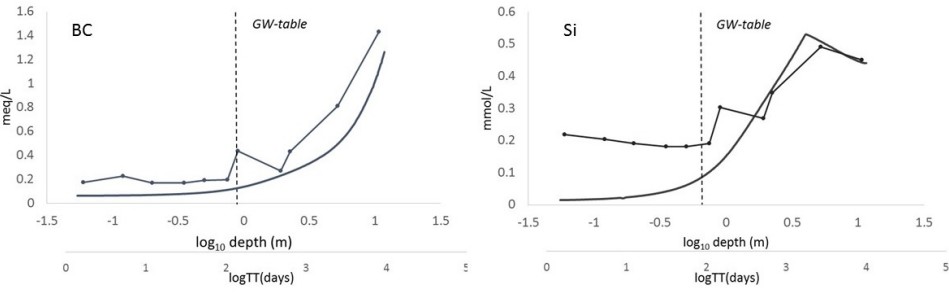

*a: Base cations*                        *b: Silica*
*Figure 24. Modelled base cation (a) and Si (b) concentrations plotted against log$_{10}$ of water transit time*
*(smooth lines) at the Svartberget field site (See Erlandsson-Lampa et al., 2016, 2019 for a full description of*
*the field test of the model). Overlain are the observed base cation and Si-concentrations from the soil profile,*
*plotted against log$_{10}$ of soil depth (straight lines with symbols).*

### 4.2. Testing the kinetic model

The most recent comparison between the kinetic weathering model results and field observations
follows in the article by Erlandsson-Lampa et al. (This issue). The research catchment where many of the
model applications have been tested is located in Northern Sweden. A few examples are shown in Figure 23
and 24. Figure 23 shows a comparison between calculated and observed base cation concentrations at the





Svartberget research site. The model results reproduce the observed concentration pattern (Zanchi et al., 2016).
Figure 23a shows the modelled base cation (Bc)[3] concentration and Figure 23b shows the Si concentrations,
plotted against $\log_{10}$ of water transit time (smooth lines). Overlaid are the observed Bc and Si-concentrations
from the soil profile, plotted against $\log_{10}$ of soil depth (solid lines with markers) in Figure 23c. The weathering
model considers all soil processes including ion exchange, vegetation interactions, decomposition of organic
matter, water transport in the catchment in both the horizontal and vertical directions (Belyazid et al., 2004,
2011a,b, 2010a,b, 2015, 2019, Erlandsson-Lampa et al., 2019, Sverdrup et al., 1995, 2002). The model
reproduces the observed field observations as a function of depth (Zanchi et al., 2016). The close
correspondence between the calculated dissolved metal concentrations and the field observation are notable
considering that we employed a silicate dissolution rate model based on laboratory measurements to determine
the composition of the aqueous phase in the soil.
**4.3. Discussion**
The detailed comparisons between laboratory measured and field determined weathering rates
generated using the kinetic models described above coupled to soil processes performed using PROFILE and
ForSAFE stand out in stark contrast to the traditional geochemical models, which give results that are several
orders of magnitude different from field observations (Erlandsson-Lampa et al., 2019). It was discovered that
past efforts to describe field weathering rates using laboratory measured dissolution rates without consideration
of the coupling of rates to the major soil processes yielded inaccurate results – see Erlandsson Lampa et al.
(2016) and Nyström-Claesson and Andersson, (1996). Such observations demonstrate a need to take into
account the complete set of processes occurring in the soil. Note that the mineral dissolution 'brake functions'
used in this approach act differently on the weathering rates that the equilibrium expressions used in earlier
models (Aagaard and Helgeson 1982, Murphy et al., 1987, Alekseyev et al., 1997, 2004, 2007, Oelkers, 2001,
Oelkers et al., 1994, 2001, 2008). The preference for using the brakes rather than the traditional rate expression
based on a slowing of rates as equilibrium is approached between the surface and the liquid is that equilibrium
is not approached for many primary silicate minerals and thus the weathering process is irreversible.
**7. Conclusions**
The complex nature of weathering in the field is nearly impossible to interpret without a
comprehensive model for the whole process. A first step to such interpretations can be the quantitative
description of the dissolution rates of the major rock forming minerals. Even the dissolution rates of an
individual mineral can involve several simultaneous reactions. Thus, experimentally measured rates results
can only be accurately interpreted when a full system model is used. Under field conditions, mineral dissolution
is coupled to other soil processes, and thus a full ecosystem system model is needed for their interpretation.
The apparent difference between field and laboratory dissolution rates arise from the coupling of these
processes, and disappear once a full model is employed. Use of a fully coupled model shows these differences
to be negligible (Keegan and Laskow-Lehey 2014).
Taking account the vast literature reporting experimentally measured mineral dissolution rates, it was
possible to create a fully parameterized kinetic database for about 100 minerals. About 40% of the kinetic
parameters were determined directly from experiment interpretations, and the rest were determined from inter-
mineral interpolations and using of analogues.
The adjustment of the aluminium 'brake function' and the introduction of a silica "brake function" as
described in this work were necessary to improve the description of weathering rates in the lower part of the
soil, below 1 meter depth. The test at the Svartberget catchment suggests that this revised mineral dissolution
model works adequately as can be seen from Figures 24-25.

**8. Acknowledgements**
This work is based upon that of Prof. Dr. Harald Sverdrup and Prof. Dr. Per Warfvinge who initiated the new
model approaches in the 1980's., Since then major contributions have been made by Dr. Matthias Alveteg (Uncertainty,
programming the code), Prof. Per Warfvinge (Programming the code), Dr. Cecilia Akselsson (Regional
parameterizations, geostatistics), Dr. Salim Belyazid (Programming the code, applying the model) and Daniel Kurz
(Adaption of the mineral stoichiometry, adaptation to Switzerland). Dr. Johan Holmqvist and Harald Sverdrup were
instrumental in taking up a second long campaign in weathering experiments, generating more kinetic data during 1997-
2004. Dr. Johan Holmqvist carefully worked out the geostatistics of landscape sampling and robustness of regional
parameterizations and creating geostatistically sound regional weathering rate maps), Dr. Salim Belyazid is the present
head code editor of the PROFILE and ForSAFE models.



This study was a part of the QWARTZ Project, coordinated by Prof. Kevin Bishop, Uppsala University, Sweden.
Dr. Salim Beliazid, Natural Geography and Quaternary Geology, Stockholm University, Sweden, Dr. Martin Erlandsson
Lampa, Institute of Hydrology, University of Uppsala, Sweden, Dr. Cecilia Akselsson, Earth Sciences, Lund University,
Lund, Sweden, Daniel Kurz, EKG Geoscience, Bern, Switzerland, Dr. Max Posch, CCE, RIVM, Bilthoven, Netherlands,
Dr Julian Aherne, Ecology, University of Trent, Canada, Dr. Jennifer Phelan, RTI Inc, Triangle Park, North Carolina,
United States of America and Professor Harald Sverdrup, Industrial Engineering, University of Iceland, Reykjavik,
Iceland (Earlier at Lund University) took part in the parameterization workshops, with the aim to have this updated
kinetics database completed.
Professor Dr. Eric Oelkers was external advisor to the project, which turned out to be a good choice. He
participated very willingly, eagerly and with excellent advice to the research process and in writing this paper.

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

Table 3. Dissolution kinetics parameterization for the 113 minerals from 12 major mineral structural groups that can be used by the current versions of the PROFILE and ForSAFE models for estimating the field soil weathering rates at 8°C. Many of the minerals can be grouped into closely related crystallographic groups where many analogues are possible. C is the limiting concentration for retarders given in units of $10^{-6}$ mol/m$^3$. Numbers in **bold** are parameters based on the direct fitting of existing experimental data. Data and records from unpublished experiments by Sverdrup and Holmqvist's experimental studies are available on paper records held at the Inland University of Applied Sciences, at Hamar, Norway. The digitized part of these data are no longer available in digital form due to computer system changes and data filing format changes that have occurred during the last 30 years. All other parameters not experimentally measured were estimated from interpolation or analogues. All concentrations are expressed in kmol m$^{-3}$. All exponents n, y, x are dimensionless. The rate coefficients k have the units as to make the weathering rate in kEq m$^{-2}$s$^{-1}$, where the area is mineral surface area (Equation 23). The weathering rate as focused to on release of cations. The release rates of Si and Al are found by stoichiometric adjustment, in subsequent steps.

| Mineral | | Fundamental chemical weathering reaction coefficients, reaction orders, and feedback effect threshold concentrations | | | | | | | | | | | | | | | | | | | | | | | | | | |
|---|---|---|---|---|---|---|---|---|---|---|---|---|---|---|---|---|---|---|---|---|---|---|---|---|---|---|---|---|
| | | H⁺-reaction | | | | | H₂O-reaction | | | | | | | CO₂-reaction[4] | | Organic acids | | | OH⁻-reaction | | | | | | | | |
| | | $pk_H$ | $n_H$ | $y_{Al}$ | $C_{Al}$ | $x_{BC}$ | $C_{BC}$ | $pk_{H2O}$ | $y_{Al}$ | $C_{Al}$ | $x_{BC}$ | $C_{BC}$ | $z_{Si}$ | $C_{Si}$ | $pk_{CO2}$ | $n_{CO2}$ | $pk_{Org}$ | $n_{Org}$ | $C_{Org}$ | $pk_{OH}$ | $w_{OH}$ | $y_{Al}$ | $C_{Al}$ | $x_{BC}$ | $C_{BC}$ | $z_{Si}$ | $C_{Si}$ |

**1a. Feldspars; tectosilicates**

| | Mineral | $pk_H$ | $n_H$ | $y_{Al}$ | $C_{Al}$ | $x_{BC}$ | $C_{BC}$ | $pk_{H2O}$ | $y_{Al}$ | $C_{Al}$ | $x_{BC}$ | $C_{BC}$ | $z_{Si}$ | $C_{Si}$ | $pk_{CO2}$ | $n_{CO2}$ | $pk_{Org}$ | $n_{Org}$ | $C_{Org}$ | $pk_{OH}$ | $w_{OH}$ | $y_{Al}$ | $C_{Al}$ | $x_{BC}$ | $C_{BC}$ | $z_{Si}$ | $C_{Si}$ |
|---|---|---|---|---|---|---|---|---|---|---|---|---|---|---|---|---|---|---|---|---|---|---|---|---|---|---|---|
| 1.1 | K-Feldspar, generic | 14.7 | 0.5 | 0.4 | 0.4 | 0.4 | 0.5 | 17.5 | 0.14 | 4 | 0.15 | 300 | 3 | 900 | **16.95** | **0.6** | **15.0** | **0.5** | 5 | 15.2 | 0.3 | 0.1 | 12 | 0.5 | 5 | 1 | 900 |
| 1.2 | K-Feldspar I | 14.8 | 0.5 | 0.4 | 0.4 | 0.4 | 0.5 | 17.8 | 0.14 | 4 | 0.15 | 300 | 3 | 900 | **17.05** | **0.6** | **15.1** | **0.5** | 5 | 15.4 | 0.3 | 0.1 | 12 | 0.5 | 5 | 1 | 900 |
| 1.3 | K-Feldspar II | 14.7 | 0.5 | 0.4 | 0.4 | 0.4 | 0.5 | 17.4 | 0.15 | 4 | 0.15 | 300 | 4 | 900 | **16.85** | **0.6** | **13.9** | **0.5** | 5 | 15.3 | 0.3 | 0.1 | 12 | 0.5 | 5 | 1 | 900 |
| 1.4 | K-Feldspar III | 14.7 | 0.5 | 0.4 | 0.4 | 0.4 | 0.5 | 17.4 | 0.15 | 4 | 0.15 | 300 | 4 | 900 | **16.80** | **0.6** | **13.9** | **0.5** | 5 | 15.2 | 0.3 | 0.1 | 12 | 0.5 | 5 | 1 | 900 |
| 1.5 | Anorthoclase | 13.6 | 0.6 | 0.4 | 0.5 | 0.4 | 0.5 | 17.2 | 0.15 | 5 | 0.15 | 200 | 3 | 900 | **16.65** | **0.6** | **13.7** | **0.5** | 5 | 14.2 | 0.3 | 0.1 | 15 | 0.5 | 5 | 2 | 900 |
| 1.6 | Albite (Ab) | 14.6 | 0.5 | 0.4 | 0.4 | 0.4 | 0.5 | 16.8 | 0.15 | 4 | 0.15 | 200 | 3 | 900 | **16.05** | **0.6** | **14.7** | **0.5** | 5 | 15.4 | 0.3 | 0.1 | 12 | 0.5 | 5 | 3 | 900 |
| 1.7 | Oligoclase | 14.6 | 0.5 | 0.4 | 0.4 | 0.4 | 1 | 16.8 | 0.15 | 4 | 0.15 | 250 | 4 | 900 | **16.05** | **0.6** | **14.7** | **0.5** | 5 | 15.4 | 0.3 | 0.1 | 12 | 0.5 | 4 | 3 | 900 |
| 1.8 | Labradorite | 13.9 | 0.5 | 0.3 | 0.5 | 0.4 | 2 | 16.8 | 0.15 | 5 | 0.15 | 300 | 5 | 900 | **16.05** | **0.6** | **14.7** | **0.5** | 5 | 14.5 | 0.3 | 0.1 | 15 | 0.5 | 3 | 3 | 900 |
| 1.9 | Bytownite | 13.8 | 0.6 | 0.3 | 0.6 | 0.4 | 3 | 16.7 | 0.15 | 6 | 0.15 | 300 | 6 | 900 | **15.95** | **0.6** | **14.6** | **0.5** | 5 | 14.4 | 0.3 | 0.1 | 18 | 0.5 | 3 | 3 | 900 |
| 1.10 | Other plagioclase | 14.6 | 0.5 | 0.4 | 0.4 | 0.4 | 1 | 16.8 | 0.15 | 4 | 0.15 | 250 | 4 | 900 | **16.05** | **0.6** | **14.7** | **0.5** | 5 | 15.4 | 0.3 | 0.1 | 12 | 0.5 | 4 | 3 | 900 |

**1b. Zeolites; tectosilicates**

| Mineral | | $pk_H$ | $n_H$ | $y_{Al}$ | $C_{Al}$ | $x_{BC}$ | $C_{BC}$ | $pk_{H2O}$ | $y_{Al}$ | $C_{Al}$ | $x_{BC}$ | $C_{BC}$ | $z_{Si}$ | $C_{Si}$ | $pk_{CO2}$ | $n_{CO2}$ | $pk_{Org}$ | $n_{Org}$ | $C_{Org}$ | $pk_{OH}$ | $w_{OH}$ | $y_{Al}$ | $C_{Al}$ | $x_{BC}$ | $C_{BC}$ | $z_{Si}$ | $C_{Si}$ |
|---|---|---|---|---|---|---|---|---|---|---|---|---|---|---|---|---|---|---|---|---|---|---|---|---|---|---|---|
| 1.11 | Helulandite | 11.9 | 0.73 | 0.2 | 30 | 0.2 | 20 | 16.8 | 0.15 | 4 | 0.15 | 250 | 3 | 900 | **16.05** | **0.6** | **14.7** | **0.5** | 5 | **14.8** | **0.3** | 0.1 | 12 | 0.5 | 4 | 2 | 900 |
| 1.12 | Analcime | 14.5 | 0.5 | 0.2 | 30 | 0.2 | 20 | 16.5 | 0.15 | 4 | 0.15 | 250 | 3 | 900 | **16.05** | **0.6** | **14.7** | **0.5** | 5 | **12.4** | **0.4** | 0.1 | 12 | 0.5 | 4 | 2 | 900 |
| 1.13 | Clinoptilolite | 14.5 | 0.3 | 0.2 | 30 | 0.2 | 20 | 16.5 | 0.15 | 4 | 0.15 | 250 | 3 | 900 | **16.05** | **0.6** | **14.7** | **0.5** | 5 | **14.8** | **0.3** | 0.1 | 12 | 0.5 | 4 | 2 | 900 |
| 1.14 | Stilbite | 14.5 | 0.3 | 0.2 | 30 | 0.2 | 20 | 16.2 | 0.15 | 4 | 0.15 | 250 | 3 | 900 | **16.05** | **0.6** | **14.7** | **0.5** | 5 | **14.7** | **0.3** | 0.1 | 12 | 0.5 | 4 | 2 | 900 |

**2. Nesosilicates**

| Mineral | | $pk_H$ | $n_H$ | $y_{Al}$ | $C_{Al}$ | $x_{BC}$ | $C_{BC}$ | $pk_{H2O}$ | $y_{Al}$ | $C_{Al}$ | $x_{BC}$ | $C_{BC}$ | $z_{Si}$ | $C_{Si}$ | $pk_{CO2}$ | $n_{CO2}$ | $pk_{Org}$ | $n_{Org}$ | $C_{Org}$ | $pk_{OH}$ | $w_{OH}$ | $y_{Al}$ | $C_{Al}$ | $x_{BC}$ | $C_{BC}$ | $z_{Si}$ | $C_{Si}$ |
|---|---|---|---|---|---|---|---|---|---|---|---|---|---|---|---|---|---|---|---|---|---|---|---|---|---|---|---|
| 2.1 | Monticellite | 7.7 | 0.55[8] | 0.1 | 100 | 0.3 | 50 | >16.4 | 0 | 100 | 0.2 | 50 | 16 | 900 | 15.4 | 0.6 | 13.9 | 0.5 | 5 | 13.3 | 0.6 | 0.1 | 100 | 0.2 | 60 | 14 | 900 |
| 2.2 | Tephroite | 9.3 | 0.56[8] | 0.1 | 100 | 0.3 | 50 | >17.0 | 0 | 100 | 0.2 | 50 | 16 | 900 | 15.4 | 0.6 | 13.9 | 0.5 | 5 | 13.3 | 0.6 | 0.1 | 100 | 0.2 | 60 | 14 | 900 |
| 2.3 | Nepheline[5] | 9.5 | 1.0 | 0.4 | 10 | 0.4 | 10 | **14.4** | 0.2 | 10 | 0.2 | 200 | 6 | 900 | 14.8 | 0.6 | 14.4 | 0.5 | 5 | 13.0 | 0.5 | 0.1 | 30 | 0.2 | 30 | 4 | 900 |

[4]There seems to be some type of CO₂ saturation of the surface between 10 and 50 atm CO₂ for mica and chlorites, beyond where the rate is no more affected. Some other minerals have indications of similar behaviour, but it remains elusive in terms of parameterization. Some minerals appear to have no detectable reaction with CO₂, some are slightly inhibited.
[5]Nepheline is classified as a feldspatoid in the mineralogical literature. However, when dissolving, the pre-dissolution complexing process at the mineral water interface create an activated surface complex with a nesosilicate structure. Thus, a nesosilicate classification here.

| | Mineral | $pk_H$ | $n_H$ | $y_{Al}$ | $C_{Al}$ | $x_{BC}$ | $C_{BC}$ | $pk_{H2O}$ | $y_{Al}$ | $C_{Al}$ | $x_{BC}$ | $C_{BC}$ | $z_{Si}$ | $C_{Si}$ | $pk_{CO2}$ | $n_{CO2}$ | $pk_{Org}$ | $n_{Org}$ | $C_{Org}$ | $pk_{OH}$ | $w_{OH}$ | $y_{Al}$ | $C_{Al}$ | $x_{BC}$ | $C_{BC}$ | $z_{Si}$ | $C_{Si}$ |
|---|---|---|---|---|---|---|---|---|---|---|---|---|---|---|---|---|---|---|---|---|---|---|---|---|---|---|---|
| 2.4 | Anorthite[6] (An) | 10.3 | 1.0 | 0.4 | 100 | 0.2 | 3 | **15.8** | 0.15 | 100 | 0.2 | 200 | 6 | 900 | **16.4** | **0.6** | **14.7** | **0.5** | 5 | 13.7 | 0.25 | 0.1 | 30 | 0.2 | 30 | 4 | 900 |
| 2.5 | Forsterite (Fo) | 10.2 | 1.0[8] | 0.1 | 1000 | 0.3 | 10 | 16.4 | 0 | 5000 | 0.2 | 5 | 16 | 900 | 15.4[7] | 0.6 | >13.9[8] | 0.5 | 5 | 13.3 | 0.6 | 0.1 | 100 | 0.2 | 60 | 14 | 900 |
| 2.6 | Olivine (Fo₅₀Fa₅₀) | 12.0 | 1.0[8] | 0.3 | 30 | 0.3 | 30 | >18.0 | 0.1 | 30 | 0.2 | 5 | 16 | 900 | 15.9[8] | 0.6 | **14.7** | 0.5 | 5 | 15.4 | 0.6 | 0.1 | 100 | 0.2 | 60 | 14 | 900 |
| 2.7 | Fayalite (Fa) | 10.2 | 1.0[8] | 0.1 | 1000 | 0.3 | 50 | 16.4 | 0 | 5000 | 0.2 | 5 | 16 | 900 | 15.4 | 0.6 | 13.9 | 0.5 | 5 | 13.3 | 0.6 | 0.1 | 100 | 0.2 | 60 | 14 | 900 |
| 2.8 | Al₁₄Py₈₄Gr₁₂ | | | | | | | | | | | | | | | | | | | | | | | | | | |
| 2.9 | Al₈₅Py₇₅ | | | | | | | | | | | | | | | | | | | | | | | | | | |
| 2.10 | Ad₈₀Gr₂₀ | 12.4 | 1.0 | 0.4 | 300 | 0.2 | 50 | **16.9** | 0.2 | 300 | 0.2 | 500 | 8 | 900 | 15.8 | 0.6 | 14.7 | 0.5 | 5 | **14.9** | **0.2** | 0.12 | 100 | 0.2 | 100 | 6 | 900 |
| 2.11 | Al₆₀Py₉₀Gr₁₀ | | | | | | | | | | | | | | | | | | | | | | | | | | |
| 2.12 | Gr₈₅Py₉Ad₆ | | | | | | | | | | | | | | | | | | | | | | | | | | |
| 2.13 | Grossular, (Gr) | | | | | | | | | | | | | | | | | | | | | | | | | | |
| 2.14 | Andradite (Ad) | | | | | | | | | | | | | | | | | | | | | | | | | | |
| 2.15 | Pyrope (Py) | 12.4 | 1.0 | 0.4 | 200 | 0.2 | 40 | **16.9** | 0.2 | 200 | 0.2 | 300 | 8 | 900 | 15.8 | 0.6 | 14.7 | 0.5 | 5 | 14.9 | 0.2 | 0.12 | 60 | 0.2 | 60 | 6 | 900 |
| 2.16 | Almandine (Al) | | | | | | | | | | | | | | | | | | | | | | | | | | |
| 2.17 | Uvarovite (Uv) | | | | | | | | | | | | | | | | | | | | | | | | | | |
| 2.18 | Spessarite (Sp) | | | | | | | | | | | | | | | | | | | | | | | | | | |
| 2.19 | Staurolite | 14.7 | 1.0 | 0.4 | 200 | 0.2 | 20 | 17.4 | 0.2 | 200 | 0.3 | 5 | 16 | 900 | 15.2 | 0.6 | 14.4 | 0.5 | 5 | **17.1** | **0.3** | 0.12 | 60 | 0.2 | 60 | 14 | 900 |
| 2.20 | Disthene | 15.5 | 1.0 | **0.33** | 10 | 0 | 500 | 17.0 | **0.33** | 10 | 0 | 500 | 4 | 900 | 16.5 | 0.5 | 15.6 | 0.5 | 5 | 15.8 | 0.4 | 0.1 | 400 | 0.3 | 60 | 3 | 900 |
| 2.21 | Kyanite | | | | | | | | | | | | | | | | | | | | | | | | | | |

**3. Pyroxenes[9] or single chain inosilicates.**

| Mineral | | $pk_H$ | $n_H$ | $y_{Al}$ | $C_{Al}$ | $x_{BC}$ | $C_{BC}$ | $pk_{H2O}$ | $y_{Al}$ | $C_{Al}$ | $x_{BC}$ | $C_{BC}$ | $z_{Si}$ | $C_{Si}$ | $pk_{CO2}$ | $n_{CO2}$ | $pk_{Org}$ | $n_{Org}$ | $C_{Org}$ | $pk_{OH}$ | $w_{OH}$ | $y_{Al}$ | $C_{Al}$ | $x_{BC}$ | $C_{BC}$ | $z_{Si}$ | $C_{Si}$ |
|---|---|---|---|---|---|---|---|---|---|---|---|---|---|---|---|---|---|---|---|---|---|---|---|---|---|---|---|
| 3.1 | Alite | 9.6 | 0.67 | 0.2 | 1000 | 0.3 | 200 | **7.85** | 0.1 | 400 | 0.3 | 5 | 16 | 900 | n.d | n.d | n.d | n.d | n.d | n.d | n.d | n.d | n.d | n.d | n.d | n.d |
| 3.2 | Wollastonite | 9.6 | 0.7 | 0 | 5000 | 0.3 | 100 | 15.1 | 0.1 | 5000 | 0.3 | 5 | 16 | 900 | **15.2** | **0.6** | 13.5 | 0.5 | 5 | 11.6 | 0.6 | 0 | 5000 | 0.5 | 5 | 8 | 900 |
| 3.3 | Spodumene | 9.6 | 0.7 | 0.2 | 400 | 0.3 | 200 | 17.2 | 0.1 | 400 | 0.3 | 5 | 16 | 900 | 15.8 | 0.6 | 14.2 | 0.5 | 5 | 14.6 | 0.6 | 0.1 | 400 | 0.5 | 5 | 8 | 900 |
| 3.4 | Diopside | 11.1 | 0.67 | 0.2 | 400 | 0.35 | 150 | **14.9** | 0.1 | 400 | 0.3 | 5 | 16 | 900 | >14.8[8] | 0.6 | 16.4 | 0.5 | 5 | 13.2 | 0.6 | 0 | 400 | 0.5 | 5 | 8 | 900 |
| 3.5 | Jadeite | 11.2 | 0.7 | 0.2 | 400 | 0.35 | 150 | **14.5** | 0.1 | 400 | 0.3 | 5 | 16 | 900 | 15.8 | 0.6 | 14.0 | 0.5 | 5 | 12.9 | 0.6 | 0 | 400 | 0.5 | 5 | 8 | 900 |
| 3.6 | Leucite | 11.1 | 0.4 | 0.2 | 400 | 0.35 | 150 | 14.5 | 0.1 | 400 | 0.3 | 5 | 16 | 900 | 14.4 | 0.6 | 14.0 | 0.5 | 5 | 12.9 | 0.6 | 0 | 400 | 0.5 | 5 | 8 | 900 |
| 3.7 | Augite I | 12.3 | 0.7 | 0.2 | 500 | 0.3 | 200 | **17.5** | 0.1 | 500 | 0.3 | 5 | 16 | 900 | 15.8 | 0.6 | 14.4 | 0.5 | 5 | 14.8 | 0.6 | 0.1 | 500 | 0.5 | 5 | 8 | 900 |
| 3.8 | Augite II | 12.3 | 0.7 | 0.2 | 500 | 0.3 | 200 | 17.5 | 0.1 | 500 | 0.3 | 5 | 16 | 900 | 15.8 | 0.6 | 14.4 | 0.5 | 5 | 14.8 | 0.6 | 0.1 | 500 | 0.5 | 5 | 8 | 900 |
| 3.9 | Hedenbergite | 12.8 | 0.7 | **0.25** | 500 | 0.2 | 200 | **17.5** | **0.16** | 500 | 0.3 | 5 | 16 | 900 | 15.8 | 0.6 | 14.4 | 0.5 | 5 | 14.8 | 0.6 | 0.1 | 500 | 0.5 | 5 | 8 | 900 |
| 3.10 | Augite II | 13.8 | 0.7 | 0.2 | 400 | 0.3 | 200 | 17.5 | 0.1 | 400 | 0.3 | 5 | 16 | 900 | 15.8[8] | 0.6 | **14.4** | 0.5 | 5 | 14.8 | 0.6 | 0.1 | 400 | 0.5 | 5 | 8 | 900 |
| 3.11 | Enstatite | 13.0 | 0.7 | 0.2 | 400 | 0.2 | 100 | 17.5 | 0.1 | 400 | 0.3 | 5 | 16 | 900 | 15.8 | 0.6 | **14.5** | 0.5 | 5 | **15.0** | **0.6** | 0.1 | 400 | 0.5 | 5 | 8 | 900 |
| 3.12 | Hypersthene | 13.2 | 0.7 | 0.2 | 400 | 0.2 | 100 | 17.6 | 0.1 | 400 | 0.3 | 5 | 16 | 900 | 15.8 | 0.6 | | 0.5 | 5 | | | 0.1 | 400 | 0.5 | 5 | 8 | 900 |
| | | | | | | | | | | | | | | | | | 14.5 | | | 15.0 | 0.6 | | | | | | |
| 3.13 | Ferrosilite | | | 0.2 | 400 | 0.3 | 200 | | 0.1 | 400 | 0.3 | 5 | 16 | 900 | 15.8[8] | 0.6 | 14.4 | 0.5 | 5 | 14.8 | 0.6 | 0.1 | 400 | 0.5 | 5 | 8 | 900 |
| | | 14.0 | 0.7 | | | | | 17.7 | | | | | | | | | | | | | | | | | | | |
| 3.14 | Bronzite | 14.4 | 0.7 | 0.2 | 400 | 0.2 | 200 | 17.5 | 0.1 | 400 | 0.3 | 5 | 16 | 900 | 15.8 | 0.6 | 14.8 | 0.5 | 5 | 14.8 | 0.6 | 0.1 | 400 | 0.5 | 5 | 8 | 900 |
| 3.15 | Pidgeonite | 13.8 | | 0.2 | 400 | 0.3 | 200 | **17.5** | 0.1 | 400 | 0.3 | 5 | 16 | 900 | 15.8[8] | 0.6 | 14.4 | 0.5 | 5 | 14.8 | 0.6 | 0.1 | 400 | 0.5 | 5 | 8 | 900 |
| | | | 0.7 | | | | | | | | | | | | | | | | | | | | | | | | |
| 3.16 | Other pyroxenes | 14.0 | 0.7 | 0.2 | 500 | 0.3 | 200 | 17.5 | 0.1 | 500 | 0.3 | 5 | 16 | 900 | 15.8 | 0.6 | 14.4 | 0.5 | 5 | 14.8 | 0.6 | 0.1 | 500 | 0.5 | 5 | 8 | 900 |

**4. Amphiboles or double chain inosilicates**

| Mineral | | H⁺-reaction | | H₂O-reaction | | CO₂-reaction | | Organic acids | | OH⁻-reaction | |
|---|---|---|---|---|---|---|---|---|---|---|---|

[6]Anorthite is classified as a feldspar in the mineralogical literature. However, when dissolving pure anorthite, the pre-dissolution complexing process at the mineral water interface create an activated surface complex with a nesosilicate structure. This applied only to pure anorthite with less than 2% other feldspars in the solution. That is why it is listed among the nesosilicates. See Sverdrup (1990) for further details. This may be thecase with Monticellite and Tephroite as well.
[7]According to Golubev et al., (2005) is the CO₂ reaction either very weak or absent, mostly from observations at high 5. For diopside and forsterite over the whole pH range (Golubev et al., 2005).
[8]A number of studies report this exponent to be 0.5. It was observed that all nesosilicates have reaction order n=1 in our own experiments, and in about half of all in the literature.
[9]He=Hedenbergite, En=Enstatite, Wo=Wollastonite, Di=Diopside, Au=Augite, Ja=Jadeite, Le=Leucite, Bz= Bronzite






| # | Mineral | $pk_H$ | $n_H$ | $y_{Al}$ | $C_{Al}$ | $x_{BC}$ | $C_{BC}$ | $pk_{H_2O}$ | $y_{Al}$ | $C_{Al}$ | $x_{BC}$ | $C_{BC}$ | $z_{Si}$ | $C_{Si}$ | $pk_{CO_2}$ | $n_{CO_2}$ | $pk_{Org}$ | $n_{Org}$ | $C_{Org}$ | $pk_{OH}$ | $w_{OH}$ | $y_{Al}$ | $C_{Al}$ | $x_{BC}$ | $C_{BC}$ | $z_{Si}$ | $C_{Si}$ |
|---|---|---|---|---|---|---|---|---|---|---|---|---|---|---|---|---|---|---|---|---|---|---|---|---|---|---|---|
| 4.1 | Glaucophane | 13.5 | 0.7 | 0.3 | 5 | 0.3 | 5 | 16.7 | 0.6 | 15 | 0.3 | 200 | 16 | 900 | 16.1 | 0.6 | 14.7 | 0.5 | 5 | >16.7 | 0.3 | 0.15 | 400 | 0.5 | 60 | 8 | 900 |
| 4.2 | Pargasite | | 0.3 | 5 | 0.3 | 5 | 16.6 | 0.6 | 15 | 0.3 | 200 | 16 | 900 | 16.1 | 0.6 | 14.7 | 0.5 | 5 | >16.7 | 0.3 | 0.15 | 400 | 0.5 | 60 | 8 | 900 |
| | | 13.8 | 0.7 | | | | | | | | | | | | | | | | | | | | | | | 8 | 900 |
| 4.3 | Hornblende I | 13.4 | 0.7 | 0.4 | 5 | 0.3 | 5 | 16.3 | 0.6 | 15 | 0.3 | 200 | 16 | 900 | 15.9[5] | 0.6 | 14.4 | 0.5 | 5 | 17.5 | 0.1 | 0.15 | 400 | 0.5 | 60 | 8 | 900 |
| 4.4 | Hornblende II | 14.8 | 0.6 | 0.3 | 5 | 0.3 | 5 | 16.5 | 0.6 | 15 | 0.3 | 200 | 16 | 900 | 16.1[5] | 0.6 | 14.5 | 0.5 | 5 | 18.2 | 0.1 | 0.15 | 400 | 0.5 | 60 | 8 | 900 |
| 4.5 | Tremolite | 15.2 | 0.2 | 0.2 | 5 | 0.3 | 5 | 16.8 | 0.6 | 15 | 0.3 | 200 | 16 | 900 | 16.2 | 0.6 | 14.8 | 0.4 | 5 | 16.1 | 0.3 | 0.15 | 400 | 0.5 | 60 | 8 | 900 |
| 4.6 | Riebeckite | 14.9 | 0.7 | 0.2 | 5 | 0.3 | 5 | 18.4 | 0.6 | 15 | 0.3 | 200 | 16 | 900 | 16.2 | 0.6 | 14.8 | 0.5 | 5 | 16.1 | 0.3 | 0.15 | 400 | 0.5 | 60 | 8 | 900 |
| 4.7 | Anthophyllite | 13.8 | 0.25 | 0.2 | 5 | 0.3 | 5 | 18.4 | 0.6 | 15 | 0.3 | 200 | 16 | 900 | 16.2 | 0.6 | 14.9 | 0.1 | 5 | 16.4 | 0.1 | 0.2 | 400 | 0.5 | 60 | 8 | 900 |
| 4.8 | Other amphiboles | 14.8 | 0.6 | 0.3 | 5 | 0.3 | 5 | 16.5 | 0.6 | 15 | 0.3 | 200 | 16 | 900 | 16.1 | 0.6 | 14.8 | 0.5 | 5 | 18.2 | 0.1 | 0.15 | 400 | 0.5 | 60 | 8 | 900 |

**5. Phyllosilicates or sheet silicates**

| Mineral | | H⁺-reaction | | | | | | H₂O-reaction | | | | | | | CO₂-reaction | | Organic acids | | | OH-reaction | | | | | | | |
|---|---|---|---|---|---|---|---|---|---|---|---|---|---|---|---|---|---|---|---|---|---|---|---|---|---|---|---|
| | | $pk_H$ | $n_H$ | $y_{Al}$ | $C_{Al}$ | $x_{BC}$ | $C_{BC}$ | $pk_{H_2O}$ | $y_{Al}$ | $C_{Al}$ | $x_{BC}$ | $C_{BC}$ | $z_{Si}$ | $C_{Si}$ | $pk_{CO_2}$ | $n_{CO_2}$ | $pk_{Org}$ | $n_{Org}$ | $C_{Org}$ | $pk_{OH}$ | $w_{OH}$ | $y_{Al}$ | $C_{Al}$ | $x_{BC}$ | $C_{BC}$ | $z_{Si}$ | $C_{Si}$ |
| 5.1 | Glauconite | 11.8 | 0.7 | 0.4 | 4 | 0.2 | 500 | 17.0 | 0.2 | 50 | 0.1 | 200 | 4 | 900 | 14.5 | 0.5 | 14.5 | 0.5 | 5 | 15.5 | 0.4 | 0.15 | 400 | 0.3 | 60 | 14 | 200 |
| 5.2 | Serpentinite, Antigotite Chrysotile | 12.7 | 0.8 | 0.2 | 50 | 0.2 | 200 | 17.5 | 0.1 | 50 | 0.1 | 200 | 16 | 900 | 14.8 | 0.5 | >14.1 | 0.5 | 5 | 17.8 | 0.6 | 0.15 | 400 | 0.3 | 60 | 14 | 200 |
| 5.3 | Talc | 13.3 | 0.7 | 0.2 | 50 | 0.2 | 200 | 16.7 | 0.1 | 50 | 0.1 | 200 | 16 | 900 | 14.5 | 0.5 | 14.5 | 0.5 | 5 | 15.5 | 0.4 | 0.15 | 400 | 0.3 | 60 | 14 | 200 |
| 5.4 | Nontronite | 14.8 | 0.3 | 0.2 | 30 | 0.2 | 20 | 16.5 | 0.15 | 4 | 0.15 | 250 | 3 | 900 | 16.05 | 0.6 | 14.5 | 0.5 | 5 | 15.4 | 0.3 | 0.1 | 12 | 0.5 | 4 | 2 | 900 |
| 5.5 | Phlogopite | 14.8 | 0.6 | 0.3 | 10 | 0.2 | 50 | 16.7 | 0.2 | 10 | 0.2 | 500 | 6 | 900 | 15.8 | 0.5 | 15.8 | 0.5 | 5 | 15.8 | 0.5 | 0.15 | 400 | 0.3 | 60 | 5 | 900 |
| 5.6 | Biotite[10] | 14.8 | 0.6 | 0.3 | 10 | 0.2 | 50 | 16.7 | 0.2 | 10 | 0.2 | 500 | 6 | 900 | 15.8 | 0.5 | 15.8[3] | 0.5 | 5 | 15.8[3] | 0.5 | 0.15 | 400 | 0.3 | 60 | 3 | 900 |
| 5.7 | Mg-Vermiculite 1[4] | 14.8 | 0.6 | 0.4 | 4 | 0.2 | 5 | 17.2 | 0.1 | 4 | 0.1 | 500 | 4 | 900 | 16.2 | 0.5 | 15.2 | 0.5 | 5 | 15.8 | 0.5 | 0.15 | 400 | 0.3 | 60 | 3 | 50 |
| 5.8 | Mg-Vermiculite 2[4] | 14.8 | 0.6 | 0.4 | 4 | 0.2 | 5 | 17.2 | 0.1 | 4 | 0.1 | 500 | 4 | 900 | 16.2 | 0.5 | 15.2 | 0.5 | 5 | 15.8 | 0.5 | 0.15 | 400 | 0.3 | 60 | 3 | 50 |
| 5.9 | Mg-Vermiculite 3[4] | 14.8 | 0.6 | 0.4 | 4 | 0.2 | 5 | 17.2 | 0.1 | 4 | 0.1 | 500 | 4 | 900 | 16.2 | 0.5 | 15.2 | 0.5 | 5 | 18.8 | 0.5 | 0.15 | 400 | 0.3 | 60 | 3 | 50 |
| 5.10 | Fe-vermiculite | 15.2 | 0.6 | 0.4 | 4 | 0.2 | 50 | 17.6 | 0.1 | 4 | 0.1 | 200 | 3 | 900 | 16.5 | 0.5 | 15.2 | 0.5 | 5 | 18.8 | 0.5 | 0.15 | 400 | 0.3 | 60 | 3 | 50 |
| 5.11 | Illitic vermiculite | 15.0 | 0.6 | 0.4 | 4 | 0.2 | 5 | 17.3 | 0.1 | 4 | 0.1 | 500 | 4 | 900 | 16.5 | 0.5 | 15.5 | 0.5 | 5 | 17.0 | 0.5 | 0.15 | 400 | 0.3 | 60 | 3 | 50 |
| 5.12 | Vermiculite Al-OH interlayer mineral | 15.2 | 0.5 | 0.4 | 4 | 0.1 | 5 | 17.5 | 0.2 | 4 | 0.1 | 500 | 6 | 900 | 16.5 | 0.5 | 15.6 | 0.5 | 5 | 17.2 | 0.4 | 0.15 | 400 | 0.3 | 60 | 5 | 100 |
| 5.13 | Fe-Chlorite | 14.8 | 0.7 | 0.2 | 50 | 0.2 | 5 | 17.0 | 0.1 | 50 | 0.1 | 200 | 4 | 900 | 16.2 | 0.5 | 15.0 | 0.5 | 5 | 18.3 | 0.4 | 0.15 | 400 | 0.3 | 60 | 3 | 50 |
| 5.14 | Chlorite | 14.8 | 0.5 | 0.2 | 50 | 0.2 | 5 | 17.0 | 0.1 | 50 | 0.1 | 200 | 4 | 900 | 16.2 | 0.5 | 12.6 | 0.5 | 5 | 18.0 | 0.4 | 0.15 | 400 | 0.3 | 60 | 3 | 50 |
| 5.15 | Mg-Chlorite | 14.3 | 0.7 | 0.2 | 50 | 0.2 | 200 | 16.7 | 0.1 | 50 | 0.1 | 200 | 4 | 900 | 15.8 | 0.5 | 15.8 | 0.5 | 5 | 18.0 | 0.4 | 0.15 | 400 | 0.3 | 60 | 3 | 50 |
| 5.16 | Smectites[11] | 14.9 | 0.5 | 0.4 | 4 | 0.1 | 5 | 17.6 | 0.2 | 4 | 0.1 | 50 | 4 | 900 | 16.5 | 0.5 | 15.6 | 0.5 | 5 | 17.5 | 0.5 | 0.1 | 400 | 0.3 | 60 | 3 | 50 |
| 5.17 | Muscovite[3] | 15.2 | 0.5 | 0.4 | 4 | 0.1 | 5 | 17.5 | 0.2 | 4 | 0.1 | 500 | 12 | 900 | 16.5 | 0.5 | 15.3 | 0.5 | 5 | 17.2 | 0.4 | 0.15 | 400 | 0.3 | 60 | 10 | 100 |
| 5.18 | Mixed muscovites | | | 0.4 | 4 | 0.1 | 5 | | 0.2 | 4 | 0.1 | 500 | 12 | 900 | 16.5 | 0.5 | 15.3 | 0.5 | 5 | | | 0.15 | 400 | 0.3 | 60 | 10 | 100 |
| | | 15.1 | 0.5 | | | | | 17.5 | | | | | | | | | | | | 17.2 | 0.4 | | | | | | |
| 5.19 | Illite 1[12] | 15.0 | 0.5 | 0.4 | 4 | 0.1 | 5 | 17.5 | 0.2 | 4 | 0.1 | 500 | 3 | 900 | 16.5 | 0.5 | 15.4 | 0.5 | 5 | 17.2 | 0.4 | 0.15 | 400 | 0.3 | 60 | 2 | 100 |
| 5.20 | Illite 2[3] | 15.2 | 0.5 | 0.4 | 4 | 0.1 | 5 | 17.5 | 0.2 | 4 | 0.1 | 500 | 3 | 900 | 16.5 | 0.5 | 15.6 | 0.5 | 5 | 17.2 | 0.4 | 0.15 | 400 | 0.3 | 60 | 2 | 100 |
| 5.21 | Illite 3[3] | 15.2 | 0.5 | 0.4 | 4 | 0.1 | 5 | 17.5 | 0.2 | 4 | 0.1 | 500 | 3 | 900 | 16.5 | 0.5 | 15.8 | 0.5 | 5 | 17.2 | 0.4 | 0.15 | 400 | 0.3 | 60 | 2 | 100 |
| 5.22 | Bentonite | 15.1 | 0.5 | 0.4 | 4 | 0.2 | 500 | 17.6 | 0.2 | 4 | 0.1 | 50 | 4 | 900 | 16.5 | 0.5 | 15.6 | 0.5 | 5 | 17.5 | 0.5 | 0.1 | 400 | 0.3 | 60 | 3 | 50 |
| 5.23 | Montmorillonite | 15.1 | 0.5 | 0.4 | 4 | 0.2 | 500 | 17.6 | 0.2 | 4 | 0.1 | 50 | 4 | 900 | 16.5 | 0.5 | 15.6 | 0.5 | 5 | 17.5 | 0.5 | 0.1 | 400 | 0.3 | 60 | 3 | 50 |
| 5.24 | Sericite | 15.2 | 0.5 | 0.4 | 4 | 0.1 | 5 | 17.5 | 0.2 | 4 | 0.1 | 500 | 3 | 900 | 16.5 | 0.5 | 15.6 | 0.5 | 5 | 17.2 | 0.4 | 0.15 | 400 | 0.3 | 60 | 2 | 100 |

**6. Cyclosilicates**

| Mineral | | H⁺-reaction | | | | | | H₂O-reaction | | | | | | | CO₂-reaction | | Organic acids | | | OH-reaction | | | | | | | |
|---|---|---|---|---|---|---|---|---|---|---|---|---|---|---|---|---|---|---|---|---|---|---|---|---|---|---|---|
| | | $pk_H$ | $n_H$ | $y_{Al}$ | $C_{Al}$ | $x_{BC}$ | $C_{BC}$ | $pk_{H_2O}$ | $y_{Al}$ | $C_{Al}$ | $x_{BC}$ | $C_{BC}$ | $z_{Si}$ | $C_{Si}$ | $pk_{CO_2}$ | $n_{CO_2}$ | $pk_{Org}$ | $n_{Org}$ | $C_{Org}$ | $pk_{OH}$ | $w_{OH}$ | $y_{Al}$ | $C_{Al}$ | $x_{BC}$ | $C_{BC}$ | $z_{Si}$ | $C_{Si}$ |

[10] All biotite and vermiculites have the same lattice breakdown rate (Sverdrup and Holmqvist 2004), the release rate results from the combination of lattice down kinetics and the mineral stoichiometry.

[11] All smectites, including montmorillonites and bentonites have the same lattice breakdown rate (Sverdrup and Holmqvist 2004), the release rate results from the combination of lattice breakdown kinetics and the mineral stoichiometry

[12] All muscovite and illites have the same lattice breakdown rate (Sverdrup and Holmqvist 2004), the release rate results from the combination of lattice breakdown kinetics and the mineral stoichiometry.


| # | Mineral | $pk_H$ | $n_H$ | $y_{Al}$ | $C_{Al}$ | $x_{BC}$ | $C_{BC}$ | $pk_{H_2O}$ | $y_{Al}$ | $C_{Al}$ | $x_{BC}$ | $C_{BC}$ | $z_{Si}$ | $C_{Si}$ | $pk_{CO_2}$ | $n_{CO_2}$ | $pk_{Org}$ | $n_{Org}$ | $C_{Org}$ | $pk_{OH}$ | $w_{OH}$ | $y_{Al}$ | $C_{Al}$ | $x_{BC}$ | $C_{BC}$ | $z_{Si}$ | $C_{Si}$ |
|---|---|---|---|---|---|---|---|---|---|---|---|---|---|---|---|---|---|---|---|---|---|---|---|---|---|---|---|
| 6.1 | Tourmaline | 13.2 | 1.0 | 0.3 | 200 | 0.2 | 200 | 15.4 | 0.2 | 200 | 0.3 | 100 | 8 | 900 | 14.8 | 0.6 | 14.4 | 0.5 | 5 | >17.0 | 0.5 | 0.15 | 400 | 0.3 | 60 | 8 | 30 |
| 6.2 | Cordierite | 15.4 | 1.0 | 0.3 | 200 | 0.2 | 200 | 16.5 | 0.2 | 200 | 0.3 | 100 | 8 | 900 | 15.9 | 0.6 | 15.5 | 0.5 | 5 | 17.4 | 0.5 | 0.15 | 400 | 0.3 | 60 | 8 | 30 |

**7. Sorosilicates**

| Mineral | | H⁺-reaction | | | | | | H₂O-reaction | | | | | | | CO₂-reaction | | Organic acids | | | OH-reaction | | | | | | | |
|---|---|---|---|---|---|---|---|---|---|---|---|---|---|---|---|---|---|---|---|---|---|---|---|---|---|---|---|
| | | $pk_H$ | $n_H$ | $y_{Al}$ | $C_{Al}$ | $x_{BC}$ | $C_{BC}$ | $pk_{H_2O}$ | $y_{Al}$ | $C_{Al}$ | $x_{BC}$ | $C_{BC}$ | $z_{Si}$ | $C_{Si}$ | $pk_{CO_2}$ | $n_{CO_2}$ | $pk_{Org}$ | $n_{Org}$ | $C_{Org}$ | $pk_{OH}$ | $w_{OH}$ | $y_{Al}$ | $C_{Al}$ | $x_{BC}$ | $C_{BC}$ | $z_{Si}$ | $C_{Si}$ |
| 7.1 | Epidote (Ep) | 14.0 | 0.8 | 0.3 | 50 | 0.2 | 5 | 17.7 | 0.2 | 50 | 0.2 | 20 | 32 | 900 | 16.2 | 0.5 | 14.4 | 0.5 | 5 | 18.4 | 0.2 | 0.15 | 400 | 0.3 | 60 | 32 | 200 |
| 7.2 | Zoisite (Zo) | 15.2 | 0.5 | 0.2 | 50 | 0.2 | 5 | 17.4 | 0.2 | 200 | 0.2 | 20 | 32 | 900 | 16.3 | 0.5 | 14.7 | 0.5 | 5 | 17.2 | 0.3 | 0.15 | 400 | 0.3 | 60 | 32 | 200 |
| 7.3 | Other zoisites | 15.2 | 0.5 | 0.2 | 50 | 0.2 | 5 | 17.4 | 0.2 | 200 | 0.2 | 20 | 32 | 900 | 16.3 | 0.5 | 14.7 | 0.5 | 5 | 17.2 | 0.3 | 0.15 | 400 | 0.3 | 60 | 32 | 200 |

**8. Aluminosilicates and quartz**

| Mineral | | H⁺-reaction | | | | | | H₂O-reaction | | | | | | | CO₂-reaction | | Organic acids | | | OH-reaction | | | | | | | |
|---|---|---|---|---|---|---|---|---|---|---|---|---|---|---|---|---|---|---|---|---|---|---|---|---|---|---|---|
| | | $pk_H$ | $n_H$ | $y_{Al}$ | $C_{Al}$ | $x_{BC}$ | $C_{BC}$ | $pk_{H_2O}$ | $y_{Al}$ | $C_{Al}$ | $x_{BC}$ | $C_{BC}$ | $z_{Si}$ | $C_{Si}$ | $pk_{CO_2}$ | $n_{CO_2}$ | $pk_{Org}$ | $n_{Org}$ | $C_{Org}$ | $pk_{OH}$ | $w_{OH}$ | $y_{Al}$ | $C_{Al}$ | $x_{BC}$ | $C_{BC}$ | $z_{Si}$ | $C_{Si}$ |
| 8.1 | Kaolinite | 15.1 | 0.7 | 0.4 | 4 | 0.4 | 5 | 17.6 | 0.2 | 5 | 0.4 | 50 | 2 | 900 | 16.5 | 0.5 | 19.5 | 0.5 | 5 | >15.1 | 0.6 | 0.15 | 400 | 0.3 | 60 | 1 | 900 |
| 8.2 | Gibbsite | 13.9 | 1.0 | 0.5 | 5 | 0 | 500 | 16.4 | 0.2 | 5 | 0.4 | n.a. | n.a. | n.a. | >18.0 | 0.5 | 16.3 | 0.5 | 5 | >13.4 | 1.0 | 0.15 | 5 | 0 | 5000 | n.a. | n.a. |
| 8.3 | Quartz | 18.4 | 0.3 | 0.5 | 5 | 0 | 500 | >17.8 | 0 | 5 | 0 | 5000 | 4 | 900 | >18.0 | 0.5 | 16.3 | 0.5 | 5 | 14.1 | 0.3 | 0.4 | 200 | 0 | 5000 | 1 | 900 |

**9. Volcanic glasses**

| Mineral | | H⁺-reaction | | | | | | H₂O-reaction | | | | | | | CO₂-reaction | | Organic acids | | | OH-reaction | | | | | | | |
|---|---|---|---|---|---|---|---|---|---|---|---|---|---|---|---|---|---|---|---|---|---|---|---|---|---|---|---|
| | | $pk_H$ | $n_H$ | $y_{Al}$ | $C_{Al}$ | $x_{BC}$ | $C_{BC}$ | $pk_{H_2O}$ | $y_{Al}$ | $C_{Al}$ | $x_{BC}$ | $C_{BC}$ | $z_{Si}$ | $C_{Si}$ | $pk_{CO_2}$ | $n_{CO_2}$ | $pk_{Org}$ | $n_{Org}$ | $C_{Org}$ | $pk_{OH}$ | $w_{OH}$ | $y_{Al}$ | $C_{Al}$ | $x_{BC}$ | $C_{BC}$ | $z_{Si}$ | $C_{Si}$ |
| 9.1 | Base cation poor volcanic glass | 15.2 | 0.5 | 0.4 | 5 | 0.1 | 300 | 18.2 | 0.1 | 5 | 0 | 50 | 2 | 900 | 17.9[5] | 0.5 | 15.7 | 0.5 | 5 | 15.7 | 0.25 | 0.25 | 5 | 0.3 | 60 | 2 | 900 |
| 9.2 | Base cation rich volcanic glass | 15.2 | 0.5 | 0.4 | 5 | 0.1 | 300 | 18.2 | 0.1 | 5 | 0 | 50 | 2 | 900 | 17.9[5] | 0.5 | 19.5 | 0.5 | 5 | 15.8 | 0.25 | 0.25 | 5 | 0.3 | 60 | 2 | 900 |
| 9.3 | Other glasses | 15.2 | 0.5 | 0.4 | 5 | 0.1 | 300 | 18.2 | 0.1 | 5 | 0 | 50 | 2 | 900 | 17.9 | 0.5 | 19.5 | 0.5 | 5 | 15.8 | 0.25 | 0.25 | 5 | 0.3 | 60 | 2 | 900 |

**10. Carbonates**

| Mineral | | H⁺-reaction | | | | | | H₂O-reaction | | | | | | | CO₂-reaction | | Organic acids | | | OH-reaction | | | | | | | |
|---|---|---|---|---|---|---|---|---|---|---|---|---|---|---|---|---|---|---|---|---|---|---|---|---|---|---|---|
| | | $pk_H$ | $n_H$ | $y_{Al}$ | $C_{Al}$ | $x_{BC}$ | $C_{BC}$ | $pk_{H_2O}$ | $y_{Al}$ | $C_{Al}$ | $x_{BC}$ | $C_{BC}$ | $z_{Si}$ | $C_{Si}$ | $pk_{CO_2}$ | $n_{CO_2}$ | $pk_{Org}$ | $n_{Org}$ | $C_{Org}$ | $pk_{OH}$ | $w_{OH}$ | $y_{Al}$ | $C_{Al}$ | $x_{BC}$ | $C_{BC}$ | $z_{Si}$ | $C_{Si}$ |
| 10.1 | Calcite[13] | 13.6 | 1.0 | 0 | 5000 | 0.4 | 5 | 14.2 | 0 | 5000 | 0.2 | 1000 | 16 | 900 | 13.2 | 0.5 | 13.2 | 0.5 | 5 | 0 | 0 | 0 | 5000 | 0 | 5000 | 16 | 900 |
| 10.2 | Aragonite | 13.6 | 1.0 | 0 | 5000 | 0.4 | 5 | 14.6 | 0 | 5000 | 0.2 | 1000 | 16 | 900 | 13.4 | 0.6 | 13.4 | 0.5 | 5 | 0 | 0 | 0 | 5000 | 0 | 5000 | 16 | 900 |
| 10.3 | Dolomite | 11.1 | 0.5 | 0 | 3000 | 0.4 | 5 | 17.5 | 0 | 3000 | 0.2 | 10 | 4 | 900 | 14.8 | 0.6 | 14.4 | 0.5 | 5 | 0 | 0 | 0 | 5000 | 0 | 5000 | 4 | 900 |
| 10.4 | Magnesite | 13.1 | 0.5 | 0 | 3000 | 0.4 | 5 | 17.6 | 0 | 3000 | 0.2 | 5 | 3 | 900 | 14.8 | 0.6 | 14.4 | 0.5 | 5 | 0 | 0 | 0 | 5000 | 0 | 5000 | 3 | 900 |
| 10.5 | Siderite[14] | 15.4 | 0.74 | 0 | 3000 | 0.4 | 5 | 18.8 | 0 | 3000 | 0.2 | 10 | 8 | 900 | 14.8 | 0.6 | 14.4 | 0.5 | 5 | 0 | 0 | 0 | 5000 | 0 | 5000 | 4 | 900 |
| 10.6 | Rhodochrosite[11] | 15.6 | 0.67 | 0 | 3000 | 0.4 | 5 | 18.6 | 0 | 3000 | 0.2 | 10 | 8 | 900 | 14.6 | 0.6 | 14.2 | 0.5 | 5 | 0 | 0 | 0 | 5000 | 0 | 5000 | 4 | 900 |

**11. Phosphates**

| Mineral | | H⁺-reaction | | | | | | H₂O-reaction | | | | | | | CO₂-reaction | | Organic acids | | | OH-reaction | | | | | | | |
|---|---|---|---|---|---|---|---|---|---|---|---|---|---|---|---|---|---|---|---|---|---|---|---|---|---|---|---|
| | | $pk_H$ | $n_H$ | $y_{Al}$ | $C_{Al}$ | $x_{BC}$ | $C_{BC}$ | $pk_{H_2O}$ | $y_{Al}$ | $C_{Al}$ | $x_{BC}$ | $C_{BC}$ | $z_{Si}$ | $C_{Si}$ | $pk_{CO_2}$ | $n_{CO_2}$ | $pk_{Org}$ | $n_{Org}$ | $C_{Org}$ | $pk_{OH}$ | $w_{OH}$ | $y_{Al}$ | $C_{Al}$ | $x_{BC}$ | $C_{BC}$ | $z_{Si}$ | $C_{Si}$ |
| 11.1 | Apatite[15] | 12.8 | 0.67 | 0 | - | 0.4 | 100 | 16.1 | 0.2 | 20 | 0.4 | 50 | n.a. | n.a. | 15.8 | 0.6 | 19.5 | 0.5 | 5 | 12.8 | 0.6 | 0.15 | 400 | 0.3 | 60 | n.a. | n.a. |
| 11.2 | Fluoroapatite | 12.8 | 0.7 | 0 | - | 0.4 | 100 | 15.9 | 0.2 | 20 | 0.4 | 50 | n.a. | n.a. | 15.8 | 0.5 | 19.5 | 0.5 | 5 | 12.8 | 0.5 | 0.15 | 400 | 0.3 | 60 | n.a. | n.a. |
| 11.3 | Other soil phosphorus solids | 12.8 | 0.7 | 0 | - | 0.4 | 100 | 15.8 | 0.2 | 20 | 0.4 | 50 | n.a. | n.a. | 15.8 | 0.5 | 19.5 | 0.5 | 5 | 12.8 | 0.5 | 0.15 | 400 | 0.3 | 60 | n.a. | n.a. |

[13] This is a general calcite. Accurate kinetic data are available for 8 different Swedish and 6 different American commercially available calcites, and 4 different Swedish, English, Finnish and Estonian dolomites (See Sverdrup and Bjerle 1983).

[14] Siderite and rhodocrosite have strong inhibition of the water reaction by dissolved oxygen in the solution.

[15] Apatite dissolution is retarded at all pH by oxalate concentrations and the presence of aluminium and iron. Silica seems to interfere less with the rate of dissolution.






Table 4. Temperature dependencies, measured are in bold. Default values were computed and scaled with Madelung crystal lattice site energy from different minerals (See Sverdrup 1990 for a detailed explanation). Normal font means we have estimated it from the lattice energies and the properties of the mineral surface. Based on the modified Arrhenius equation (Sverdrup 1990, 1998, Sverdrup and Warfvinge 1988, 1992, 1995). The units are °K⁻¹ used in the Arrhenius equation as defined in Sverdrup (1990).

| Mineral | | $H^+$ | $H_2O$ | $CO_2$ | Organic acids | $OH^-$ | Comments |
|---|---|---|---|---|---|---|---|
| | | | | 1. Feldspars | | | |
| 1.1-1.2 | K-Feldspar I; Orthoclase, Sanidine | **3500** | **1940** | 1700 | **1200** | **3200** | Irreversible dissolution |
| 1.3 | K-Feldspar II; Microcline | **3470** | **1820** | **1700** | **1200** | **3200** | Irreversible dissolution |
| 1.4 | K-Feldspar III; Orthoclase | **4090** | 2000 | 1700 | 1200 | 3500 | Irreversible dissolution |
| 1.5 | Anorthoclase | **3500** | **2000** | 1700 | 1200 | **3200** | Irreversible dissolution |
| 1.6 | Plagioclase; Albite | **3350** | **2500** | **1680** | 1200 | **3100** | Irreversible dissolution |
| 1.7 | Plagioclase; Oligoclase | **4200** | **2330** | 1700 | 1200 | **3600** | Irreversible dissolution |
| 1.8 | Plagioclase; Labradorite | **4200** | **2500** | 1700 | 2200 | 3500 | Irreversible dissolution |
| 1.9-1.10 | Plagioclase; Bytownite and near anorthite | **3500** | **2500** | 1700 | 1200 | 3100 | Irreversible dissolution |
| 1.11 | All other feldspars | 3685 | 2085 | 1700 | 1200 | 3100 | Irreversible dissolution |
| | | | | 1b. Zeolites | | | |
| 1.12 | Helulandite | 3500 | 2550 | 1700 | 1200 | 3450 | Irreversible dissolution |
| 1.13 | Analcime | 3500 | 2500 | 1700 | 1200 | 3400 | Reversible reaction |
| 1.14 | Clinoptilolite | 3500 | 2550 | 1700 | 1200 | 3600 | Irreversible dissolution |
| 1.15 | Stilbite | 3500 | 2500 | 1700 | 1200 | 3400 | Irreversible dissolution |
| | | | | 2. Nesosilicates | | | |
| 2.1 | Monticellite | **3480** | **4200** | 1700 | **1600** | **2200** | Irreversible dissolution |
| 2.2 | Tephroite | **2551** | **4400** | 1700 | **1534** | **1450** | Irreversible dissolution |
| 2.4 | Anorthite (An) | **1820** | **5670** | 1700 | **1800** | **1700** | Irreversible dissolution |
| 2.5 | Forsterite (Fo) | **3350** | **4510** | 1700 | 1800 | **2100** | Irreversible dissolution |
| 2.6 | Olivine | **2580** | **4510** | 1700 | 1800 | **2100** | Irreversible dissolution |
| 2.7 | Fayalite | **2550** | **4400** | 1700 | 1800 | 2200 | Irreversible dissolution |
| 2.13 | Nepheline | **3630** | **3130** | 1700 | 1800 | **2180** | Irreversible dissolution |
| 2.8-2.18 | Garnet mixes, all garnets | 2500 | 3500 | 1700 | 1800 | 2000 | Irreversible dissolution |
| 2.19 | Staurolite | 3100 | 3200 | 1700 | 1800 | **3100** | Irreversible dissolution |
| 2.20-2.21 | Disthene, Kyanite | **3918** | **2400** | 1700 | 1800 | **2200** | Irreversible dissolution |
| 2.22 | All other nesosilicates | 2676 | 4436 | 1700 | 1800 | 2180 | Irreversible dissolution |
| | | | | 4. Pyroxenes | | | |
| 3.2 | Wollastonite | **3100** | **3600** | 1700 | 2000 | **2100** | Irreversible dissolution |
| 3.4 | Diopside | **2610** | **3400** | 1700 | 2000 | 2000 | Irreversible dissolution |
| 3.9 | Hedenbergite | **2311** | **3500** | 1700 | 2000 | 2000 | Irreversible dissolution |
| 3.7-3.8, 3.10 | Augite | **2700** | **4100** | 1700 | 2000 | 2000 | Irreversible dissolution |
| 3.11 | Enstatite | **2550** | **5950** | 1700 | 2000 | 2000 | Irreversible dissolution |
| 3.16 | All other pyroxenes | 2700 | 4100 | 1700 | 2000 | 2000 | Irreversible dissolution |
| | | | | 4. Amphiboles | | | |
| 4.1 | Glaucophane | **4300** | **3800** | 1700 | 2000 | 3500 | Irreversible dissolution |
| 4.2 | Hornblende I | **4300** | **3800** | 1700 | 2000 | 3500 | Irreversible dissolution |
| 4.3 | Hornblende II | 4300 | 4000 | 1800 | 2200 | 3500 | Irreversible dissolution |
| 4.4 | Tremolite | **4500** | **3390** | 1700 | 2000 | 3600 | Irreversible dissolution |
| 4.5 | Antophyllite | 3800 | 3300 | 1700 | 2200 | 4500 | Irreversible dissolution |
| 4.6 | All other amphiboles | 4300 | 3390 | 1700 | 2000 | 3500 | Irreversible dissolution |
| | | | | 5. Phyllosilicates | | | |
| 5.1 | Glauconite | **4300** | **1950** | 1700 | 2000 | 3500 | Irreversible dissolution |
| 5.2 | Serpentinite, Chrysotile, Antigorite | **4282** | **3600** | 1700 | 2000 | 3500 | Irreversible dissolution |
| 5.3 | Talc | **4200** | **3700** | 1700 | 2000 | 3500 | Irreversible dissolution |
| 5.4 | Nontronite | 4500 | 3500 | 1700 | 1200 | 3400 | Irreversible dissolution |
| 5.6 | Biotite | **4500** | **3840** | 1700 | 2000 | 3500 | Irreversible dissolution |
| 5.5 | Phlogopite | **4500** | **3840** | 1700 | 2000 | 3500 | Irreversible dissolution |
| 5.7 | Vermiculite 1 | 4500 | 3840 | 1700 | 2000 | 3500 | Alteration mineral, irreversible dissolution |
| 5.8 | Vermiculite 2 | 4500 | 3840 | 1700 | 2000 | 3500 | Alteration mineral, irreversible dissolution |
| 5.9 | Vermiculite 3 | 4500 | 3840 | 1700 | 2000 | 3500 | Irreversible dissolution |
| 5.10 | Fe-Chlorite | 4500 | 3800 | 1700 | 2000 | 3500 | Irreversible dissolution |
| 5.14 | Fe-Mg-Chlorite | **4520** | 3500 | 1700 | 1800 | 3500 | Irreversible dissolution |
| 5.17 | Mg-Chlorite | **4500** | **1400** | 1700 | 1700 | 3500 | Irreversible dissolution |
| 5.19 | Muscovite | 3038 | 3800 | 1700 | 2000 | **4656** | Irreversible dissolution |
| 5.21 | Illite 1 | 4500 | 3800 | 1700 | 2000 | 3500 | Alteration mineral, irreversible dissolution |
| 5.22 | Illite 2 | 4500 | 3800 | 1700 | 2000 | 3500 | Alteration mineral, irreversible dissolution |
| 5.23 | Illite 3 | 4500 | 3800 | 1700 | 2000 | 3500 | Irreversible dissolution |
| 5.24 | Montmorillonite | 4300 | **3840** | 1700 | 2000 | 3500 | Alteration mineral, irreversible dissolution |
| 5.27 | All other phyllosilicates | 4410 | 3770 | 1700 | 2000 | 3500 | Irreversible dissolution |
| | | | | 7. Cyclosilicates | | | |
| 6.1 | Tourmaline | **3600** | **3100** | 1700 | 1800 | 2500 | Irreversible dissolution |
| 6.2 | Cordierite | **2600** | **5900** | 1700 | 2000 | 2000 | Irreversible dissolution |
| 6.3 | All other cyclosilicates | 3100 | 4500 | 1700 | 1900 | 2250 | Irreversible dissolution |
| | | | | 8. Sorosilicates | | | |
| 7.1 | Epidote | **5330** | **3800** | 1700 | 2000 | 2300 | Irreversible dissolution |
| 7.2 | Zoisite | **4400** | **3900** | 1800 | 2200 | 3300 | Irreversible dissolution |
| 73 | All other sorosilicates | 4375 | 3850 | 1750 | 2100 | 3300 | Irreversible dissolution |
| | | | | 10. Oxides and simple aluminosilicates | | | |
| 8.1 | Kaolinite | **5310** | **3580** | 1700 | 2000 | **4100** | Irreversible dissolution, gibbsite possible outcome |
| 8.2 | Gibbsite | **3400** | **3600** | 1700 | 2000 | **3170** | Alteration mineral, irreversible dissolution |
| 8.3 | Quartz | **3890** | n.a. | **2200** | 2000 | **3320** | Reversible reactions, back reaction, dissolution is kinetically limited |
| | | | | 11. Volcanic glasses | | | |
| 9.1 | Volcanic glass, base cation poor | **3890** | **3010** | 2400 | 2800 | 2700 | Irreversible dissolution |
| 9.2 | Volcanic glass, base cation rich | **4500** | **3310** | 2500 | 2800 | 3400 | Irreversible dissolution |
| 9.3 | All other volcanic glasses | 4200 | 3110 | 2450 | 2800 | 3050 | Irreversible dissolution |
| | | | | 10 Carbonates | | | |
| 10.1 | Calcite and limestones | **444** | **1180** | **2180** | **2200** | - | Reversible reaction, Back reaction important |
| 10.2 | Aragonite | **530** | **1210** | **2200** | **2400** | - | Reversible reaction, Back reaction important |
| 10.3 | Dolomite | **1880** | **2700** | **1800** | **2200** | - | Irreversible dissolution. Back reaction to calcite and magnesite |
| 10.5 | Siderite | **3300** | **3500** | 1700 | 2000 | 2500 | Irreversible dissolution |
| 10.6 | Rhodochrosite | **3300** | **3500** | 1700 | 2000 | 2500 | Irreversible dissolution |
| | | | | 11 Phosphates | | | |
| 11.1 | Apatite | **3500** | **4000** | 1700 | **1200** | 2500 | Irreversible dissolution, precipitates with oxalate and aluminium important |
| 11.2 | Fluoroapatite | **1110** | **4790** | 1700 | **1200** | 2500 | Irreversible dissolution, precipitates with oxalate and aluminium important |
| 11.3 | Immobilized inorganic phosphorus, all other phosphorus | **2350** | **4000** | 1700 | 1200 | 2200 | Possibly reversible reaction |







| Table 5. Stoichiometry of the minerals applied in Tables 3 and 4. | | |
|---|---|---|
| **1a. Feldspars** | | |
| | Mineral | Formula |
| 1.1 | K-Feldspar | $KAlSi_3O_8$ = Or |
| 1.2 | K-Feldspar I; Orthoclase, K-Feldspar I; Sanidine, 100-90% | $Or_{97}An_3$ |
| 1.3 | K-Feldspar II; 90%, Microcline | $Or_{97}Ab_2An_1$ |
| 1.4 | K-Feldspar II; 80%, Orthoclase | $Or_{80}Ab_{20}$ |
| 1.5 | Anorthoclase | $Or_{29}Ab_{62}An_{17}$ |
| 1.6 | Albite | $NaAlSi_3O_8$ = Ab |
| 1.7 | Plagioclase; Oligoclase | $Ab_{85}An0_{15}$ |
| 1.8 | Plagioclase; Labradorite | $Ab_{46}An_{54}$ |
| 1.9 | Plagioclase; Bytownite | $Ab_{22}An_{78}$ |
| 1.10 | Plagioclase; feldparic Anorthite | $Ab_6An_{94}$ |
| **1b. Zeolites with tectosilicate structure** | | |
| 1.12 | Helulandite | $(Ca,Na)_{0.45}Al_{0.89}Si_{3.1}O_8 \cdot 2.7\ H_2O$ |
| 1.13 | Analcime | $NaAlSi_2O_6 \cdot H_2O$ |
| 1.14 | Clinoptilolite | $(Na,K,Ca)_{2-3}Al_3(Al,Si)_2Si_{13}O_{36} \cdot 12H_2O$ |
| 1.15 | Stilbite | $Na_{0.09}Ca_{0.66}AlSi_3O_8 \cdot 3.1\ H_2O$ |
| **2. Nesosilicates** | | |
| 2.1 | Monticellite | $CaMgSiO_4$ |
| 2.2 | Tephoite | $Mn_2SiO_4$ |
| 2.3 | Nepheline | $(Na_{0.75}K_{0.25})AlSiO_4$ |
| 2.4 | Anorthite | $CaAl_2Si_2O_8$ = An |
| 2.5 | Forsterite | $Mg_2SiO_4$ |
| 2.6 | San Carlos, Arizona Forsterite | $Mg_{1.81}Fe_{0.19}SiO_4$ |
| | Salem, Tamil Nadu Indian olivine | $Mg_{1.84}Fe_{0.16}SiO_4$ |
| | Norwegian Olivine ($Fo_{65}Fa_{35}$) | $Mg_{1.5}Fe_{0.35}Al_{0.02}Si_{1.04}O_4$ |
| 2.7 | Fayalite | $Fe_2SiO_4$ |
| 2.8-2.12 | Generic garnet, continuous series | $Al_{44}Py_{44}Gr_{12}, Al_{65}Py_{35}, Ad_{80}Gr_{20}, Al_{50}Py_{40}Gr_{10}, Gr_{88}Py_6Ad_6$ |
| 2.13 | Grossular | $Ca_3Al_2(SiO_4)_3$ |
| 2.14 | Almandine =Al | $Fe_3Al_2(SiO_4)_3$ |
| 2.15 | Spessartine = Sp | $Mn_3Al_2(SiO_4)_3$ |
| 2.16 | Andradite = Ad | $Ca_3Fe_2(SiO_4)_3$ |
| 2.17 | Uvarovite = Uv | $Ca_3Cr_2(SiO_4)_3$ |
| 2.18 | Pyrope = Py | $Mg_3Al_2(SiO_4)_3$ |
| 2.19 | Staurolite | $Mg_{0.2}Fe_{1.2}Al_{7.4}Si_{4.3}O_{22}(OH)_2$ |
| 2.20 | Disthene | $Al_2SiO_5$ |
| 2.21 | Kyanite | $Al_2SiO_5$ |
| **2.223. Pyroxenes (End members are diopside, hedenbergite, enstatite, ferrosilite)** | | |
| 3.1 | Alite (T-slag, K-slag) | $Ca_3SiO_5$ or $(CaO)_3SiO_2$ |
| 3.2 | Wollastonite ($Ca_{22}Si_2O_6$) | $Ca_{1.7}Mg_{0.11}Si_{2.2}O_6$ |
| 3.3 | Spodumene ($LiAlSi_2O_6$) | $LiAl_{0.86}Fe_{0.3}Si_2O_6$ |
| 3.4 | Diopside ($CaMgSi_2O_6$) | $Ca_{1.04}Mg_{1.0}Al_{0.02}Fe_{0.01}Si_{2.03}O_6, Ca_{0.8}Mg_{0.8}Fe_{0.2}Al_{0.2}Si_2O_6$ |
| 3.5 | Jadeite ($NaAlSi_2O_6$) | $Na_{1.0}Ca_{0.2}Fe_{0.3}AlSi_2O_6$ |
| 3.6 | Leucite ($KAlSi_2O_6$) | $Na_{0.05}K_{1.09}Al_{1.15}Si_{2.3}O_6$ |
| 3.7 | Augite I | $He_{55}En_{45}$ |
| 3.8 | Augite II | $En_{51}Wo_{39}He_{10}$ |
| 3.9 | Hedebergite ($CaFeSi_2O_6$) | $Ca_{0.4}Mg_{0.7}Fe_{0.09}Al_{0.15}Si_{1.86}O_6$ |
| 3.10 | Augite III | $Ca_{0.86}Mg_{1.0}Fe_{0.02}Si_2O_6$ |
| 3.11 | Enstatite ($Mg_2Si_2O_6$) | $Mg_{1.7}Fe_{0.3}Si_2O_6$ |
| 3.12 | Hypersthene | $MgFeSi_2O_6$ ($En_{50}Fs_{50}$) |
| 3.13 | Ferrosilite | $Fe_2Si_2O_6$ |
| 3.14 | Bronzite (mixed) | $Mg_{1.54}Fe_{0.42}Ca_{0.2}Si_{1.9}O_6$ ($En_{70}He_{10}Fs_{20}$) |
| 3.15 | Pidgeonite | $Mg_{50}Ca_{15}Fe_{35}Si_2O_6$ |
| 3.16 | Mixed pyroxenes | $Ca_{0.8}Mg_{0.9}Fe_{0.3}Al_{0.04}Si_2O_6$ (DixEnvFszHew) |
| **4. Amphiboles** | | |
| 4.1 | Glaucophane | $Na_2MgFe_2Al_2Si_8O_{22}(OH)_2$ |
| 4.2 | Pargasite | $NaCa_2(Mg_4Al)(Si_6Al_2)O_{22}(OH)_2$. |
| 4.3 | Hornblende I (Norwegian) | $Ca_{2.1}Mg_{4.5}Na_{0.08}Al_{2.1}Si_7O_{22}(OH)_2(PO_4)_{0.01}$ |
| 4.4 | Hornblende II (Canadian) | $Ca_{2.0}Mg_{4.0}Na_{0.16}Al_{0.4}Si_{8.3}O_{22}(OH)_2$ |
| 4.5 | Tremolite | $Ca_2Mg_5Si_8O_{22}(OH)_2$ |
| 4.6 | Riebeckite | $Na_2Fe^{2+}{}_3Fe^{3+}{}_2Si_8O_{22}(OH)_2$ |
| 4.7 | Anthophyllite | $Mg_{5.7}FeAl_{0.1}Si_{7.8}O_{22}(OH)_2$ |
| 4.8 | Other amphiboles | Various compositions |
| **5. Phyllosilicates** | | |
| 5.1 | Glauconite | $(K,Na)(Fe^{3+},Al,Mg)_2(Si,Al)_4O_{10}(OH)_2$ |
| 5.2 | Serpentine, Antigorite, Chrysotile | $Mg_{4.1}Fe_{0.4}Al_{0.15}Si_{2.8}O_{10}(OH)_4, (Mg, Fe)_3Si_2O_5(OH)_4$ |
| 5.3 | Talc | $Mg_{2.8}Fe_{0.18}Si_4O_{10}(OH)_3$ |
| 5.4 | Nontronite | $Ca_{.5}(Si_7Al_{.8}Fe_{.2})(Fe_{3.5}Al_{.4}Mg_{.1})O_{20}(OH)_4$ |
| 5.5 | Phlogopite | $K_{1.0}Mg_3Al_{1.0}Si_3O_{10}(OH)_2$ |
| 5.6 | Biotite | $K_{0.9}Mg_{1.9}Fe_{1.1}Al_{1.0}Na_{0.1}Si_3O_{10}(OH)_2$ |



| 5.7 | Mg-Vermiculite I | $K_{0.5}Mg_{1.5}Fe_{1.1}Al_{1.7}Na_{0.05}Si_3O_{10}(OH)_2$ |
|---|---|---|
| 5.8 | Mg-Vermiculite II | $K_{0.3}Mg_1Fe_{1.1}Al_{1.5}Si_3O_{10}(OH)_2$ |
| 5.9 | Mg-Vermiculite III | $K_{0.1}Mg_{0.5}Fe_{1.1}Al_2Si_3O_{10}(OH)_2$ |
| 5.10 | Fe-Vermiculite | $(Mg,Fe^{+2},Fe^{+3})_3[(Al,Si)_4O_{10}](OH)_2 \cdot 4H_2O$ |
| 5.11 | Illitic vermiculite | $K_{0.35}Mg_{0.11}Ca_{0.03}Al_{2.13}Fe_{0.32}Ti_{0.07}Si_{3.4}O_{10}(OH)_2$ |
| 5.12 | Vermiculite Al-OH interlayer mineral | $(Mg, Al, Fe^{2+})_3 (Si,Al)_4 O_{10} (OH)_2 \cdot nH_2O$ |
| 5.13 | Fe-Chlorite V, Chamosite | $Fe_5Al_2Si_3O_{10}(OH)_8$ |
| 5.14 | Chlorite IV (mixed) | $Mg_{0.7}Fe_{2.7}Al_{2.3}Si_3O_{10}(OH)_8$ |
| 5.15 | Chlorite III (mixed) | $Mg_2Fe_3Al_2Si_3O_{10}(OH)_8$ |
| 5.16 | Chlorite II (mixed) | $Mg_{4.9}Fe_{0.6}Al_{1.4}Si_3O_{10}(OH)_8$ |
| 5.17 | Mg-Chlorite I, Clinochlore | $Mg_5Al_2Si_3O_{10}(OH)_8$ |
| 5.18 | Smectite | $Ca_{0.2}Mg_{1.6}Na_{0.13}Al_{1.0}Si_4O_{10}(OH)_2$ |
| 5.19 | Muscovite | $KAl_3Si_3O_{10}OH_2$ |
| 5.20 | Muscovite (mixed) | $K_{0.9}Na_{0.02}Mg_{0.3}Fe_{0.4}Al_{2.7}Si_{3.5}O_{10}(OH)_2$ |
| 5.21 | Illite I | $K_{0.1}Mg_{0.28}Fe_{0.3}Al_{2.6}Si_{3.3}O_{10}(OH)_2$ |
| 5.22 | Illite II | $K_{0.7}Mg_{0.26}Fe_{0.1}Al_{2.5}Si_{3.1}O_{10}(OH)_2$ |
| 5.23 | Illite III | $K_{0.6}Mg_{0.25}Al_{2.3}Si_3O_{10}(OH)_2$ |
| 5.24 | Montmorillonite | $Ca_{0.2}Mg_{1.6}Na_{0.13}Al_{1.0}Si_4O_{10}(OH)_2$ |
| 5.25 | Bentonite | See illite |
| 5.26 | Sericite | $KAl_2Si_3O_{10}(OH)_2$ |
| | **6. Cyclosilicate** | |
| 6.1 | Tourmaline | $Ca_{1.0}Fe_3MgAl_5Si_6O_{18}(BO_3)_3(OH)_4(PO_4)_{0.01}$ |
| 6.2 | Cordierite | $Ca_{3.5}Fe_{0.07}K_{0.09}Al_{3.3}Si_{4.6}O_{18}$ |
| | **7. Sorosilicates** | |
| 7.1 | Epidote | $Ca_{1.5} K_{0.46}Fe_{0.74}Al_{1.5}Si_{3.4}O_{12}(OH)$ |
| 7.2 | Zoisite (Clino-) | $Ca_{2.2}Fe_{0.13}Al_{1.5}Si_{3.2}O_{12}(OH)$ |
| | **8. Clay minerals** | |
| 8.1 | Kaolinite | $Al_2Si_2O_5(OH)_4$ |
| 8.2 | Gibbsite | $Al(OH)_3$ |
| 8.3 | Quartz | $SiO_2$ |
| | **9. Glasses** | |
| 9.1 | Volcanic glass, base cation poor | $Ca_{0.2}Mg_{0.2}K_{0.4}Na_{0.4}Al_{0.8}Si_3O_8$ |
| 9.2 | Volcanic glass, base cation rich | $Ca_{0.62}Mg_{0.53}K_{0.27}Na_{0.27}Al_{0.66}Si_{2.68}O_8$ |
| | **10. Carbonates** | |
| 10.1a | Calcite (Ca) | $(CaCO_3)_{99.9}(Ca_5(PO_4)_3(OH))_{0.1}$ |
| 10.1b | Köping limestone | $Ca_{97}Do_2Ma_1Ap_{0.1}$ |
| 10.1c | Red Öland limestone | $Ca_{97}Do_1Sd_2Ap_{0.1}$ |
| 10.1d | Ignaberga limestone | $Ca_{50}Ar_{45}Do_1Sd_2Ap_{0.5}$ |
| 10.2 | Aragonite (Ar) | $(CaCO_3)_{99.9}(Ca_5(PO_4)_3(OH))_{0.1}$ |
| 10.3 | Dolomite (Do) | $(CaMg(CO_3)_2)_{99.9}(Ca_5(PO_4)_3(OH))_{0.1}$ |
| 10.4 | Magnesite (Ma) | $MgCO_3$ |
| 10.6 | Rhodochrosite | $MnCO_3$ |
| 10.5 | Siderite (Sd) | $FeCO_3$ |
| | **11. Phosphorus minerals** | |
| 11.1 | Apatite (Ap) | $Ca_5(PO_4)_3(OH)$ |
| 11.2 | Fluoroapatite | $Ca_5(PO_4)_3(OH_{0.7}F_{0.3})$ |
| 11.3 | Immobilized inorganic phosphorus | Unknown, assume as semi-apatite $(Ca_3AlFe_{0.5})_5(PO_4)(F_{0.1}OH_{0.4}(CO_3)_{0.5})$ |





## Appendix. Overview of the PROFILE family of weathering rate modelling codes

A large number of computational weathering models are based on PROFILE approach. To clarify these models and their interconnections the following list is provided

1. **Steady-state weathering rate models**
   a. 1987-1995; Warfvinge P. and Sverdrup, H.; The single site version of the **PROFILE** model for the calculation and mapping of critical loads and rates of field chemical weathering was developed. It has been validated and used operationally in more than 50 countries worldwide. It uses laboratory generated kinetic models and coefficients to predict field weathering rates. The interface software for PROFILE became outdated, thus, this version is no longer available.
   b. 1992-present; Sverdrup, H., Warfvinge, P., Alveteg, M., Walse, C., Kurz, P., Posch, M., Belyazid, S.; The code **RegionalPROFILE** was developed. This code is a regionalized version of PROFILE, used for creating weathering rate maps for soils and catchments across regions and countries, as well as to estimate critical loads for forest soils. Updated versions of the code are available upon request from Sverdrup, Akselsson or Belyazid.
   c. 2000; Sverdrup, H. and Alveteg, M., The **CLAY-PROFILE** code was developed. This model was made for volcanic and clayey agricultural soils. This code is no longer operable. Archived, the code is available upon written request from Sverdrup or Belyazid.
2. **Dynamic weathering models**
   a. 1987-2008; Warfvinge P., Sverdrup, H., Alveteg, M., Walse, C., Martinsson, L.: The **SAFE** model and its helper routine **MakeDep** were created. **SAFE** is a generally applicable dynamic soil chemistry and acidification model. This tool is used worldwide for acidification research, forest sustainability assessments and for mapping critical loads.
   b. 1995-1996; Rietz, F., Sverdrup, H., Warfvinge, P.; The **SkogsSAFE** model was developed. This long-term dynamic model simulates soil genesis, mineralogy dynamics, soil chemistry and base cation release from chemical weathering in soils over time since the most recent glaciation (14,000 years ago to present) (Rietz 1995, Warfvinge et al., 1996). This code is written in FORTRAN. This code and its databases are available upon written request from Sverdrup.
   c. 1996-2004; Sverdrup, H., Wallman P., Belyazid, S., Alveteg, M., Walse, C., Martinsson, L.: These scientists developed **ForSAFE,** an integrated biogechemical forest ecosystem model for growth, nitrogen and carbon cycling. This code is written in FORTRAN code, and the code is available upon written request from Sverdrup or Belyazid.
3. **Regional mineralogy estimation**
   a. 1990; Sverdrup, H., Melkerud, P. A., Kurz, D.: The **UPPSALA** model was developed for the reconstruction of soil mineralogy from soil total analysis data. This model is run in a spreadsheet. It is available upon written request from Sverdrup.
   b. 1998; Sverdrup, H. and Erdogan, B. The **Turkey** mineral depletion model (TMD) was developed. This model estimates soil mineralogy from bedrock geology and estimates of soil age. This code is written in STELLA®. It is archived and available upon written request from Sverdrup.
   c. 2005-2010; Posch, M., Kurz, D., Alveteg, M., Akselsson, C., Eggenberger, U., Holmqvist, J; 2007 **A2M**, a model to quantify mineralogy from geochemical analyses was developed. This code is available on-line from doi:10.1016/j.cageo.2006.08.007, https://dl.acm.org/citation.cfm?id=1231715or from Kurz or Akselsson (Posch et al., 2006, 2007).

These models are not commercial products. They do not have ready-made handbooks (only the early single site PROFILE models had a good users interface and a user's manual). The models are available, but the best option to learn how to run these get training from the contact scientists in how to operate the models and how to set up the input data for a site or a region. The core code is written in FORTRAN