# Peer review of "Improved parameterization of the weathering kinetics module in the PROFILE and ForSAFE models"

_Biogeosciences, 2019_

## Referee Comment (RC1) · Anonymous Referee #1 · 12 Apr 2020

This is not a scholarly publishable paper for a variety of reasons. The paper reads like a monograph – in fact, an old fashioned monograph where the author asserts his/her opinions. I do not find the paper to be publishable in this peer reviewed journal. Some examples of why I dont find this paper publishable are shown below.

1. The main reason to publish this paper might be because PROFILE as a model (and its descendents and modifications) has been amply used in the European system to assess impacts of acid rain. Nonetheless, the model is based on an approach that has never been fully or adequately discussed in the literature. In particular, kinetic constants in the model are often described only in Sverdrup, H.U., 1990. The Kinetics

of Base Cation Release Due to Chemical Weathering. Lund University Press, Lund, Sweden, which is itself a monograph with many unsubstantiated assertions and questionable data sources. This paper does not address this limitation and only seems to make the problem worse. The authors should be encouraged to publish this paper as a monograph because this paper is impossible to peer review.

2. Apparently, one of the things that is new in this presentation is the addition of "braking functions". As far as I can tell, these are fudge factors that are not based on data nor theory. Some of the Al effect may be based on Oelkers' models but the Si function does not seem to even have that basis (Figure 9 shows no data). In fact, no reactive transport model can be used to simulate a natural field system accurately without fudge factors. Usually authors choose surface area as a fudge factor. As I see this paper, these authors are simply choosing braking factors as their fudge factors. I don't see why this is an advance. Especially when it is not adequately explained.

3. The abstract suggests that the paper treats 2D and 3D catchments but I do not see where that is described in the paper. What, in particular, is a 2D catchment? The assertions in the abstract are over reaching.

4. Most of the citations throughout are only to papers with Sverdrup as a co author. The many other papers that are cited are almost strictly cited as a footnote to one of the tables or they are briefly mentioned when reaction rate orders are chosen. But very little substantiation for choices is presented.

5. While the authors need to present specifics to build confidence in what they are doing, the paper almost entirely deals in generalities. For example, the first figure is a simple schematic of feedback loops. How does this help the paper? It has very little to contribute. Likewise, other figures are not helpful: for example, Figure 2 is ostensibly a comparison of field to model rates, but so little information is given that no interpretation of the figure can be made by the reader.

6. The first paragraph in Section 2. Methodology provides little in the way of useful

content.

7. One of the interesting points of this paper and the model is that Sverdrup et al. point out that modelling feldspar dissolution as if the feldspar precipitates is inaccurate...and this is how all the other models treat the system. As written in the paper, however, the authors simply make an assertion that re-precipitation of feldspar cannot happen and so the TST treatment is not appropriate. Interestingly, new data seem to substantiate the authors' assertion...but are not mentioned or cited. The authors should see Zhu et al. 2020 in GCA. This is another good example of how the authors are not really pinned to data or to the literature at all. This could become a big paradigm shift, if all models shifted how they treat feldspar dissolution (as well as other minerals). But this is not adequately defended nor substantiated.

8. The authors assert that the way the model treats surface area has been reviewed in detail. Nonetheless, my understanding of what the model does with surf area is that it assumes a particle size distribution that then sets the surface area. This is perhaps not as robust or theoretically defensible as the authors assert or imply.

9. Some descriptions are impossible to understand: line 230 , "if some reactions occupy the same active mineral surface sites, the expression given above would change to a quadratic sum." I could give many other examples of sentences that were impossible for me to understand.

10. The caption of figure 6 says that the reaction pathways are shown according to Transition State Theory...which sounds good...but what does this mean? I don't see how this figure is related to TST and the caption does not make it clear. There are many examples like this where big words and big assertions are used or made that make what is being done here sound better than it really is. (Although who knows? It is all a mystery)

11. The authors do not explain partial causal loop diagrams and yet they present one. This is from systems theory but is not well explained here. It is slightly explained

around lines 125 but the presentation is not adequate (if the reader is to understand the importance here).

12. I may have missed something but the authors seem to only include retardation of mineral reactions by organic molecules. Do they include acceleration by organic molecules (which is known to happen)?

13. The authors seem to think it is ok to define clay minerals to include non clays such as quartz. There are two definitions of clays...but the authors do not make this clear.

14. I did not understand the pathways in Table 2.

15. The authors cite a lot of experimental data papers in footnote 2. But they state that the data derive from these papers, but are not limited to these papers. This is not scholarly. How can we have faith in what they have done if we cannot figure out what they have done?

16. In the data table for kinetic constants, the authors write: " ...the regression of $\sim$20 have yet to be published. In due time, these will get their own proper publications, it is beyond the scope of this study to do them in detail. Data and records from unpublished experiments and experiment evaluations by Sverdrup and Holmqvist are available on paper records held in a large number of binders at the Inland University of Applied Sciences, at Hamar, Norway. These data are no longer available in digital form due to computer system changes and data filing format changes that have occurred during the last 20 years. This documentation could be available in 1-2 years time, provided that funding for the redigitalization work can be obtained." I have never seen anything published with that sort of apology. I think it is inappropriate. If we cannot see the data, then the authors are simply making an assertion. Which minerals fall into this category?

17. Why don't these authors use standard units? See Figure 12: kmol/m3. Are these standard units in some field? If so, fine. If not, please use standard units.

18. I could not figure out figures 13 nor 14...how did the authors parameterize the effect of Al and BC based on what I can see in the figures?

19. In some cases, I saw that the authors cited many more recent papers but I was left to wonder if they actually used those papers. Or were they just citing them to make it look like their compilation of kinetic constants is up to date? A few examples. On line 690 or so, they cite papers for the rate order with respect to CO2. Earlier, they cited a paper by Navarre-Sitchler and Thyne, but it is not cited here. Was that paper used here or not? Somehow, it seems like magic, they come up with a rate order of 0.6. It is unclear how they arrived at this value. Another mystery: "Estimates for some of the rate coefficients in Table 3 were based on mineral crystal structure analogies (citations)." Which minerals? And how did this analogy argument work? Likewise, I do not understand what the authors mean by Figure 22. "Data points drawn in". I am left with little confidence in the rate constants that the authors chose here.

20. Even when the authors seemed up to date in their citations, it was not clear that they actually used the new data to do anything. They cite a thesis and a paper about olivine reactions by A Olsen. Did they use those two references? What about the compilation by Palandri and Kharaka? How was that compilation used? P and K are cited, but was the compilation used? For example, in the specific discussion late in this paper about olivine reaction rate order, the Olsen and Rimstidt paper and the Olsen thesis were not cited. In fact, many many people have discussed the reaction rate orders for different minerals, but these discussions in the literature do not seem to have been considered here. Rather, it seems like ad hoc decisions have been made.

---

## Referee Comment (RC2) · Anonymous Referee #2 · 13 Apr 2020

This paper is useful in the sense that it provides a compilation of available data on mineral kinetics. Although empirical rather than mechanistical in approach, it still can have its place in applied science, but the suitability for Biogeosciences is in doubt. The manuscript is not properly executed and does not fit the format of a paper in academic journal. I noted 3 scientific problems that have to be resolved by the authors before the paper can be considered for further review. 1) Database issue. The manuscript aims to provide a kinetic database for executing some weathering models. This task is not fulfilled. A simple compilation of available kinetic data is not the same as critical evaluation of these data and expert recommendation of rate constants (regardless of the type of equation used to model the rates). Consider some analogy with thermodynamics. A database used in any thermodynamic model is a product of EXPERT evaluation of available thermodynamic information data, and recommendation of definite values based on expert judgement of various experimental (or modeling - if needed) works. The same should be clearly done for kinetic database, but this step is missing or not presented in the manuscript. The table of recommended kinetic constants should contain at least some pertinent references in the way it is organized for thermodynamic databases. Example is given in L598-605. This is totally unacceptable. Original data and their quality evaluation should be presented. Another example is L 921-923: the use of inter-mineral interpolations and of analogues is totally obscure.

2) Natural application issue. The dissolution of alumosilicates, at far from equilibrium conditions at least, is controlled by activity of free $Al^{3+}(aq)$ in solution (unequivocally demonstrated by group of Schott, Oelkers). The binding of $Al^{3+}$ to natural organic ligands decreases the activity of main inhibitor of alumosilicate dissolution and thus increases the rates. This Al complexation is not considered (or not described) in the model because simple organic ligands (carboxylates, aromatic) cannot be used to predict $Al^{3+}$ speciation in the presence of complex fulvic and humic acids, as well as organo-mineral (Al, Fe) colloids. The authors mention some acids (L306-307) but what are these acids? Some soils contain 5-10 cm of organic layer and then the mineral layer occurs. These minerals will interact with DOM. The equation 13 is not justified. Why mineral dissolution should be slowing with increasing organic ligand concentration (L395)? This contradicts to a large body of knowledge on mineral dissolution, notably of alumosilicate minerals. How the secondary mineral precipitation was modelled?

3) The quality of modeling is not demonstrated. The use of Na (Figs 13, 14, 15) without considering stoichiometry and Si release rate is not suitable. Original data allowing to see the quality to modeling should be presented. The model curves should be given for other plots (Figs 16 -18). Overall, the need of the empirical adjustments (braking functions) is not clear or not presented. There are robust equations of mineral dissolution rates, cited in L 904-905

Some technical issues (to be re-evaluated at the 2nd stage): Abstract is simply impossible to read by non-experts. The braking function should be defined. L568-569: quartz does not belong to clay minerals
* * *

---

## Author Comment (AC1) · 31 May 2020

First some important statements by the authors to the Journal and its editorial board:

The paper was received and published in 20. October 2018 by the Journal, reviewed and went through large revisions three times during 2018 and 2019, then the procedure broken was off by the journal without explanation. We were told the we needed to resubmit the paper again in February 2020, and discover that everything starts again back to zero. Thus we think the correct time statement for the article should be:

Received and published: 20. October 2018

[Figure]

The purpose of this paper was to present the changes made to the PROFILE-ForSAFE model system in recent years, and focused on the community of integrated ecosystems modelers. A number of earlier papers presenting the PROFILE-ForSAFE model system have hundreds of citations. Since the version of the model now in use incorporate a number of changes not present in the earlier, well-cited presentations of the model system, we felt this virtual special issue would be an excellent opportunity to update the presentation of the model system based on the research that has motivated the changes in the model system. (This VSI also includes several papers, now published, that use the updated system.) This paper is not focused on the geochemistry community working with experimental weathering kinetics, which has shown little or no interest in the PROFILE-ForSAFE model in the past. It is also not meant as a detailed step-by-step parameterization study, rather it documents the PROFILE-ForSAFE kinetic coefficients database that is being used by the research community, including several of the papers not accepted and published in this Virtual Special Issue.

It appears as if the reviewers for this new round of reviews have not seen the earlier reviewers comments and the revisions made to the original paper. The constructive tone of the critique in the earlier roudns of reviews (and the extensive changes made to address those reviewer comments), makes the very negative tone of the current reviews, all the more remarkable. We are prepared to address the concerns the reviewers have identified which we consider justified, and where we do not feel concerns are justified, we will do our best to explain our basis for protesting. In many cases this is because the current reviewers do not seem aware of the original publications stretching back over three decades that are the basis for this current article.

Anonymous Referee #1

This is not a scholarly publishable paper for a variety of reasons. The paper reads like a monograph – in fact, an old fashioned monograph where the author asserts his/her opinions. I do not find the paper to be publishable in this peer reviewed journal. Some examples of why I dont find this paper publishable are shown below.

We are surprised that the editors were willing to allow a statement like this to be published, given the journal's guidelines for reviewer comments. We document below the errors and lack of awareness of integrated systems modelling that lead to this very disturbing openting statement by the reviewer.

1. The main reason to publish this paper might be because PROFILE as a model (and its descendents and modifications) has been amply used in the European system to assess impacts of acid rain. Nonetheless, the model is based on an approach that has never been fully or adequately discussed in the literature.

This is a remarkable statement that summarily dismissed the value of dozens of papers over the course of decades. The model has been discussed extensively in the scientific literature, we do not understand why this reverwer chooses to ignore that and disregard advances made in Europe concerning modelling field weathering rates. We suggest the reviewer explain why it is that the judgment of so many reviewers and editors should be disregarded.

We would also like to point out that this model system has been used for mapping regional soil weathering rates and soil weathering rate at a large number of research sites, across Europe as well as in both Americas and Asia. We are not aware that any of the 25 countries that have used the system and subjected the model to field tests by independent research teams has deemed the performance to be unsatisfactory. The PROFILE and ForSAFE models have been used widely in European soil chemistry, environmental chemistry and found use in soil and water acidification, estimation of critical loads for air pollution related to sulfur and nitrogen, to assess nutrient (N, P, K, Mg, Ca) forest sustainability and agricultural sustainability assessments. It has also found use in geochemical assessment of weathering rates for nuclear waste repositories. The main reason for publishing the paper, is to make the kinetics coefficients used in the model available, and show some examples of how the values were estimated. In a companion paper by Erlandsson-Lampa et al., 2020. field testing of the new model configuration was tested. The purpose is not to show how every rate coefficient and

parameter was estimated, tracing it back to the primary data, step-by-step. It shows how the model has been updated, and examples of data used and examples of how it was tested in a new setting. Further testing of the model and kinetics has been shown in a companion paper by Erlandsson-Lampa et al. 2020.

In particular, kinetic constants in the model are often described only in Sverdrup, H.U., 1990. The Kinetics of Base Cation Release Due to Chemical Weathering. Lund University Press, Lund, Sweden, .......and Chartwell-Bratt, London, UK. which is itself a monograph with many unsubstantiated assertions and questionable data sources.

The model, including the kinetic constants has been described in the scientific literature in many publications. It would be helpful if the reviewer could be clear about which assertions are only described in the 1990 book and are unsubstantiated. It is only when confronted by specific examples (and we expect many such examples on the basis of how the reviewer has phased this) for it to be possible for us to even begin to reply to this sweeping charge.

This paper does not address this limitation and only seems to make the problem worse. The authors should be encouraged to publish this paper as a monograph because this paper is impossible to peer review.

We do not understand why characterizing something as a monograph means that it is not possible to peer review. If this reviewer does not feel that they can provide a peer review of this manuscript, why are they proceeding to make a review?

2. Apparently, one of the things that is new in this presentation is the addition of "braking functions". As far as I can tell, these are fudge factors that are not based on data nor theory.

We strongly disagree with this statement. The Erlandsson et al. paper in this VSI should be a conclusive refutation. But this statement also reveals that the reviewer has not gone to the effort to familiarize themselves with the original weathering book

(Sverdrup 1990), or the publication in the American Mineralogy Society (Sverdrup and Warfvinge 1995) or any of the other earlier papers. In these publications, the scientific basis of the retardation functions have been explained on basis of the TST. The braking functions were introduced already in our 1988 papers, and further described in detail with their theoretical basis in Sverdrup (1990), and again explained in publications in 1992, 1993 and 1995. And most recently, Erlandsson et al., in this VSI.

Some of the Al effect may be based on Oelkers' models but the Si function does not seem to even have that basis (Figure 9 shows no data). In fact, no reactive transport model can be used to simulate a natural field system accurately without fudge factors.

If the reviewer is of the opinion that any reactive transport model applied to natural systems "must use fudge factors", then they seem to be of the opinion that all published literature in this area from the European soil chemistry modelling arena during the last 20-30 years is relying on fudge factors. We feel it unfair that that viewpoint should be applied to this paper, as it should have apparently stopped all earlier publications in this area as well. .

Usually authors choose surface area as a fudge factor.

We will be please to refute this statement in a review by documenting the use of surface area measurements in calibration. .

As I see this paper, these authors are simply choosing braking factors as their fudge factors. I don't see why this is an advance. Especially when it is not adequately explained.

The model has been explained in great detail in the underlying published papers and books. And the braking functions are based in experimental lab data, as explained already 30 years ago. I take this as an admission that the reviewer did not read Sverdrup (1990) nor ever had any experience with the PROFILE or ForSAFE model. The most recent example of the exploration of "brake" functions is presented in the Erlandsson

et al. paper in this special issue.

3. The abstract suggests that the paper treats 2D and 3D catchments but I do not see where that is described in the paper. What, in particular, is a 2D catchment?

The original PROFILE model was a straight percolation down through a soil column, a one-dimensional model (1D). Lateral flow was introduced, there was movement vertically and laterally, a 2-dimensional model of a catchment (2D). Including flow in two lateral dimensions plus the vertical dimension, would be a 3-dimensional model (3D).

The assertions in the abstract are over reaching.

We will request the reviewer to substantiate that concern.

4. Most of the citations throughout are only to papers with Sverdrup as a co-author. The many other papers that are cited are almost strictly cited as a footnote to one of the tables or they are briefly mentioned when reaction rate orders are chosen. But very little substantiation for choices is presented.

This paper is a compilation and update of our work and not a detailed walkthrough review of all underlying primary data. The model is explained in our earlier work, and it is intentional that we give a list of the documentation of the development process behind our models. Complaining about that the model is not explained in detail and then complaining that we list the references that explain the model appears to be a self-contradiction.

5. While the authors need to present specifics to build confidence in what they are doing, the paper almost entirely deals in generalities. For example, the first figure is a simple schematic of feedback loops. How does this help the paper? It has very little to contribute.

It explains how to read the causal loop diagram used later in Figures 6 and 7. Causal loop diagrams are differential equations in graphical form, which is evident from Figure 1. The reviewer asks for equations and explanations, that are present in the text. We

can take away Figure 1, but Figures 6 and 7 are necessary.

Likewise, other figures are not helpful: for example, Figure 2 is ostensibly a comparison of field to model rates, but so little information is given that no interpretation of the figure can be made by the reader.

The comment is simply not true, only a poor attempt to discredit the work and a confirmation that the reviewer is not familiar with the earlier peer reviewed papers on PRO-FILE and ForSAFE. The diagram has been shown and explained in the following references (Sverdrup 1990, 1996, Sverdrup and Alveteg 1998, Sverdrup and Warfvinge 1988a,b, 1991, 1992a,b, 1993, 1995, Sverdrup et al., 1990, 1998).

6. The first paragraph in Section 2. Methodology provides little in the way of useful content.

We would appreciate more specifics about what the reviewer deems not to be useful.?

7. One of the interesting points of this paper and the model is that Sverdrup et al. point out that modelling feldspar dissolution as if the feldspar precipitates is inaccurate. and this is how all the other models treat the system. As written in the paper, however, the authors simply make an assertion that re-precipitation of feldspar cannot happen and so the TST treatment is not appropriate. Interestingly, new data seem to substantiate the authors' assertion, but are not mentioned or cited.

It has been known for over 60 years that one cannot precipitate feldspar (or most other soil primary silicate minerals) from solution. It has been discussed many times over the years. Garrels and other geochemists knew this and Stumm said so in his university lectures at the ETH already 40 years ago, and he says so clearly in his books (Stumm and Wieland 1990). The use of "thermodynamical data" to limit dissolution is a real fudge factor, since it assumes an assumption of reversible reactions that is well known NOT to exist. Thus, in traditional models, the saturation term is a simple forcing function used to turn the dissolution off. It does not reflect any underlying mechanism, and

calling the saturation coefficients "thermodynamic data is really false marketing.

The authors should see Zhu et al. 2020 in GCA. This is another good example of how the authors are not really pinned to data or to the literature at all. This could become a big paradigm shift, if all models shifted how they treat feldspar dissolution (as well as other minerals). But this is not adequately defended nor substantiated.

Our paper was submitted in 2018, and revisions were completed, successfully we believed, in 2019, which explains why this paper is not in the manuscript.

The evaluation of the data was written about in Sverdrup (1990) and Sverdrup and Warfvinge (1995) in detail, and was reiterated is different rehearsals and additions many times later. We are listing a long number of papers, and had to remove a lot for reasons of having to shorten the article. The mineral weathering rate model applied in PROFILE, ForSAFE and the podsolization and soil development simulating model SkogsSAFE uses the transition state theory concepts, but do not introduce the false assumption of solution equilibrium and solution saturation based on that. Thus, this is the appropriate TST treatment and it is described in Sverdrup (1990) the first time and repeated since.

8. The authors assert that the way the model treats surface area has been reviewed in detail. Nonetheless, my understanding of what the model does with surf area is that it assumes a particle size distribution that then sets the surface area. This is perhaps not as robust or theoretically defensible as the authors assert or imply.

This is a comment made without checking how the methodology works. The particle size distribution is not assumed, it is measured. For Sweden this was done for a total of 27,000 sites. In Switzerland in 660 sites. And so on through country after country. Translation functions to go from particle size measurements to surface area were determined for a number of regions. It is there in the publications that the reviewer complains about. This is described in detail in Warfvinge and Sverdrup (1993, 1995) and Sverdrup (1990), and a number of other publications (See Sverdrup and Stjernquist

2002).

9. Some descriptions are impossible to understand: line 230 , "if some reactions occupy the same active mineral surface sites, the expression given above would change to a quadratic sum." I could give many other examples of sentences that were impossible for me to understand.

These is a simple mathematical proof for this (Sverdrup 1990), but we thought that would be diving into too much detail to show it in full. And it has already been published. But we can take the sentence away.

10. The caption of figure 6 says that the reaction pathways are shown according to Transition State Theory. . .which sounds good. . .but what does this mean? I don't see how this figure is related to TST and the caption does not make it clear. There are many examples like this where big words and big assertions are used or made that make what is being done here sound better than it really is. (Although who knows? It is all a mystery)

This is explained in Sverdrup (1990) and Sverdrup and Warfvinge (1995). If the reviewer is not satisfied with the explanations there, the reason for this should be stated.

11. The authors do not explain partial causal loop diagrams and yet they present one. This is from systems theory but is not well explained here. It is slightly explained around lines 125 but the presentation is not adequate (if the reader is to understand the importance here).

We can expand the explanation of the causal loop diagrams in the text. Figure 1 explains how a CLD works, but the reviewer did not understand that (Said the reviewer earlier).

12. I may have missed something but the authors seem to only include retardation of mineral reactions by organic molecules. Do they include acceleration by organic molecules (which is known to happen)?

Yes, we do include that. In our full reply we will show where this is in the equations. When the surface adsorption sites get filled, then the reaction rate that run proportional to the surface adsorbed amount, does not increase if the solution concentration is further increased. Because the increased solution concentration cannot put more ligands at the surface because all sites are full. That is the internal retardation for organic molecules and $CO_2$. The retardation term also describes actions of the cations, Al and Si.

13. The authors seem to think it is ok to define clay minerals to include non clays such as quartz. There are two definitions of clays...but the authors do not make this clear.

We can make this clear with a sentence.

14. I did not understand the pathways in Table 2.

These are described in detail in Sverdrup (1990) and Sverdrup and Warfvinge (1995). We will add some clarifying sentences in the text. The pathways are also illustrated in Figure 11 which appears as fairly clear.

15. The authors cite a lot of experimental data papers in footnote 2. But they state that the data derive from these papers, but are not limited to these papers. This is not scholarly. How can we have faith in what they have done if we cannot figure out what they have done?

Some of the data come from the literature and substantial amounts from our own experiments over very many years. There has been many chances to get familiar with our work, and we published a lot of it every year. This is the only integrated soil chemistry model that really can do weathering rates and passes the field test, without fudge factors, so the reviewer ought to be familiar with it. Because of this, this is the United Nations Economic Commissions (UN/ECE-LRTAP) recommended model for mapping soil and catchment weathering rates across nations and large regions. The "standard Geochemical Models" based on" thermodynamic data" have failed all field tests

grandly, and need massive fudge-factors to get to something that looks presentable. And under full field conditions, they simply are orders of magnitude off and were rejected as dysfunctional for operational use in the UN/ECE LRTAP work both in Europe and in the US.

16. In the data table for kinetic constants, the authors write: " . . .the regression of âĹij20 have yet to be published. In due time, these will get their own proper publications, it is beyond the scope of this study to do them in detail. Data and records from unpublished experiments and experiment evaluations by Sverdrup and Holmqvist are available on paper records held in a large number of binders at the Inland University of Applied Sciences, at Hamar, Norway. These data are no longer available in digital form due to computer system changes and data filing format changes that have occurred during the last 20 years. This documentation could be available in 1-2 years time, provided that funding for the redigitalization work can be obtained." I have never seen anything published with that sort of apology. I think it is inappropriate. If we cannot see the data, then the authors are simply making an assertion. Which minerals fall into this category?

Whenever the truth is inappropriate, I will stand up and fight for it. The kinetic experiments will be published when we have the funding for it. You can see the data, all the binders and paper files are available. The experiments were done 1984-1991, and 2000-2004.

17. Why don't these authors use standard units? See Figure 12: kmol/m3. Are these standard units in some field? If so, fine. If not, please use standard units.

These are standard European and SI system units. Moles and meter are metric units. Used in chemistry and chemical engineering.

18. I could not figure out figures 13 nor 14...how did the authors parameterize the effect of Al and BC based on what I can see in the figures?

Please read the text, it is there. Then check Sverdrup (1990) and Sverdrup and War-fvinge (1990).

19. In some cases, I saw that the authors cited many more recent papers but I was left to wonder if they actually used those papers. Or were they just citing them to make it look like their compilation of kinetic constants is up to date? A few examples. On line 690 or so, they cite papers for the rate order with respect to CO2. Earlier, they cited a paper by Navarre-Sitchler and Thyne, but it is not cited here. Was that paper used here or not? Somehow, it seems like magic, they come up with a rate order of 0.6. It is unclear how they arrived at this value. Another mystery: "Estimates for some of the rate coefficients in Table 3 were based on mineral crystal structure analogies (citations)." Which minerals? And how did this analogy argument work?

We used some of the papers when there was enough information available for assisting parameterization. Do note that the parameterization has taken place gradually over 30 years, and have been steadily, but slowly updated. See Sverdrup (1990).

Likewise, I do not understand what the authors mean by Figure 22. "Data points drawn in". I am left with little confidence in the rate constants that the authors chose here.

20. Even when the authors seemed up to date in their citations, it was not clear that they actually used the new data to do anything. They cite a thesis and a paper about olivine reactions by A Olsen. Did they use those two references?

The data that pass the quality criteria are pooled with other data and used in the evaluation. In some of the minerals, the new data was studied and if it did not really bring anything new or change the kinetics in a significant way, it was only used to confirm what we already knew.

What about the compilation by Palandri and Kharaka? How was that compilation used?

The kinetic coefficients depend on the rate equations used for parameterizing them. Palandri and Kharka (2004) used a model type that does not have the capability of

estimating field weathering rates, since it has no reaction brake terms and rests on the old saturation term, which rests on a faulty assumption (The saturation term called "thermodynamic") as mentioned in the comments earlier. Thus, their coefficients based on a model unsuited for field conditions is not helpful for the PROFILE or ForSAFE models. We have to some extent used some of the same primary data as Palandri and Kharka (2004) did, but with a very different model, coming up with a different result. Thus, the compilation of was of limited value.

P and K are cited, but was the compilation used? For example, in the specific discussion late in this paper about olivine reaction rate order, the Olsen and Rimstidt paper and the Olsen thesis were not cited. In fact, many people have discussed the reaction rate orders for different minerals, but these discussions in the literature do not seem to have been considered here. Rather, it seems like ad hoc decisions have been made.

Reaction rates and a theory for that was worked out already in Sverdrup (1990). The reviewer can find what we did there. Stumm has a partial explanation of those mechanisms in one of his papers. The reaction order involves both stoichiometry as well as what mechanism is taking place at the surface. There it was also discussed why researchers came up with different reaction orders, and how sometimes their experimental design would distort the results and lead to misinterpretations Stumm, W. and Wieland, E., Dissolution of oxide and silicate minerals; rates depend on surface speciation. In: Stumm, W. Ed.., Aquatic Chemical Kinetics. Wiley-Interscience, New York. 1990. Considering that this particular paper was started 2017 at a workshop in Ystad, Sweden, and submitted in 2018 it will be up to date to that year. The references reflect what we have done over the last 30 years in the field, and is not all-out review of the very latest literature. For the purpose of this paper, a long discussion of reaction order of forsterite does not seem very important for the overall performance of the model, since natural soils very rarely have more than 0.2% olivine. We guess the reviewer thinks of Olsen, A.A., and Rimstidt, J.D. (2008), Oxalateâ Řpromoted forsterite dissolution at low pH, Geochim. Cosmochim. Acta, 72, 1758– 1766, and Rimstidt, J.D.,

Brantley, S.L. Olsen, A.A. 2012, Systematic review of forsterite dissolution rate data, Geochim. Cosmochim. Acta, 99, 159–178

---

## Author Comment (AC2) · 31 May 2020

This paper is useful in the sense that it provides a compilation of available data on mineral kinetics. Although empirical rather than mechanistical in approach, it still can have its place in applied science, but the suitability for Biogeosciences is in doubt.

We are of the opinion that the approach underlying the PROFILE-ForSAFE model system does have a complete theoretical foundation, which should be apparent from the published documentation of the model. If the reviewer questions the very presence

of a theoretical basis for this model system, then it will be difficult for us to satisfy the reviewer that there is a case for publishing this paper, despite the fact that it defines the scientific basis for improvements made in a widely using modeling system,.

The manuscript is not properly executed and does not fit the format of a paper in academic journal. I noted 3 scientific problems that have to be resolved by the authors before the paper can be considered for further review.

1 - Database issue. The manuscript aims to provide a kinetic database for executing some weathering models. This task is not fulfilled. A simple compilation of available kinetic data is not the same as critical evaluation of these data and expert recommendation of rate constants (regardless of the type of equation used to model the rates). Consider some analogy with thermodynamics. A database used in any thermodynamic model is a product of EXPERT evaluation of available thermodynamic information data, and recommendation of definite values based on expert judgement of various experimental (or modeling - if needed) works. The same should be clearly done for kinetic database, but this step is missing or not presented in the manuscript.

It our response we intend to point out that an expert evaluation is presented and the methodology is described over several pages. The manuscript describes how the available data was used for the rate coefficients, and then how the gaps in the data table wer filled using interpolation methods. For details, there are references to earlier works, going in detail and showing the path back to the primary data. We will recommend in particular Sverdrup (1990) and Sverdrup and Warfvinge (1995) carefully.

The table of recommended kinetic constants should contain at least some pertinent references in the way it is organized for thermodynamic databases. Example is given in L598-605. This is totally unacceptable.

In our reply we will argue that the models referred to as "thermodynamic" based on the use of a saturation factor which assumes an equilibrium that probably does NOT exist in soils for primary minerals. These models are neither "thermodynamic" nor not

pass any field test. The resulting weathering rates obtained with these models are from 3 to 4 orders off when tested on field, which means they are not useful. Only the use of massive adjustments make the results appear as publishable. The geochemical community sometimes failed to take this fact in, and failed to make necessary innovations. We will argue that the scientific discussion about weathering should have room for alterantives to the thermodynamic approach. But if this see seen as "totally unacceptable", then we believe that the reviewer would not have accepted any of the papers published on the PROFILE-ForSAFE system of models.

Original data and their quality evaluation should be presented. Another example is L 921-923: the use of inter-mineral interpolations and of analogues is totally obscure.

Sverdrup (1990) explains how this was done, and this has been elaborated in later publications.

2 - Natural application issue. The dissolution of alumosilicates, at far from equilibrium conditions at least, is controlled by activity of free Al3+(aq) in solution (unequivocally demonstrated by group of Schott, Oelkers). The binding of Al3+ to natural organic ligands decreases the activity of main inhibitor of alumosilicate dissolution and thus increases the rates. This Al complexation is not considered (or not described) in the model because simple organic ligands (carboxylates, aromatic) cannot be used to predict Al3+ speciation in the presence of complex fulvic and humic acids, as well as organo-mineral (Al, Fe) colloids.

We agree that the solution activity Al3+ is part of the braking mechanism, and there is more than just Al3+ being involved. This is all considered in the model (See the references to our work on this), and described in detail the referenced papers published by the authors on the subject.

The authors mention some acids (L306-307) but what are these acids? Some soils contain 5-10 cm of organic layer and then the mineral layer occurs. These minerals will interact with DOM. The equation 13 is not justified.

For how this was derived, we will back to Sverdrup (1990), Sverdrup and Warfvinge (1995) and further work by Sverdrup and Holmqvist in the years after. The organic acids were assembled into two groups, depending on reactivity with the surfaces (Sverdrup 1990). We disagree with the reviewer and have data to show the expression is valid and justified. That has been published several times, and we will call the reviewer's attention to this. Other researchers also support such a formulation.

Why mineral dissolution should be slowing with increasing organic ligand concentration (L395)? This contradicts to a large body of knowledge on mineral dissolution, notably of alumosilicate minerals.

We will disagree with this, by taking the reviewr through the relevant equations, and also with with references to Sverdrup (1990), Sverdrup and Warfvinge (1995). The rate of dissolution increases with the activity of the organic ligand at the surface. When all adsorption sites at the surface are occupied, then increasing solution concentration will have no further accelerating effect (Sverdrup 1990). This is evident from the rate equation.

How the secondary mineral precipitation was modelled?

This was modelled as kinetic precipitation, and is detailed in the description of the SkogSAFE model version (Rietz 1995). This was tested at the chronosequences and data from the integrated monitoring sites at the Gårdsjøn Research Catchment.

3 - The quality of modeling is not demonstrated. The use of Na (Figs 13, 14, 15) without considering stochiometry and Si release rate is not suitable.

The stochiometry is explained in Sverdrup (1990) and Sverdrup and Warfvinge (1995). We ill provide precise direction to where that was discussed, so that we can hopefully make clear that there is no need to republish that.

Original data allowing to see the quality to modeling should be presented. The model curves should be given for other plots (Figs 16 -18). Overall, the need of the empir-

ical adjustments (braking functions) is not clear or not presented. There are robust equations of mineral dissolution rates, cited in L 904-905

This section was limited because of lack of space (The original contribution was 63 pages), and after discussions with earlier reviewers, a number of graphs and figures were taken away. A part of the reference list was taken away as we were asked to reduce length substantially (From 63 to 49 pages means that stuff has to go. If we are being asked to put this back again, we could do that). The reference list can easily be made 10 pages longer, but that is not the purpose of the article. Thus discussing every reference is just out of the question. More of the model testing is described in the companion paper by Erlandsson-Lampa et al. 2020 in this VSI, which we do refer to already.

Some technical issues (to be re-evaluated at the 2nd stage): Abstract is simply impossible to read by non-experts. The braking function should be defined. L568-569: quartz does not belong to clay minerals

Quartz can be removed from the text. Abstracts can be rewritten, but it is not meant for the layman. The reader has to have some background about weathering. The braking functions are in the equations; but we can add some more explanation in the text.